# A Retention-Centric Framework for Continual Learning with Guaranteed Model Developmental Safety

## Abstract

In real-world applications, learning-enabled systems often undergo iterative model development to address challenging or emerging tasks. This continual model development process raises a significant issue that acquiring new or improving existing capabilities may inadvertently lose good capabilities of the old model, also known as catastrophic forgetting. While existing continual learning aims to mitigate catastrophic forgetting by trading off performance on previous tasks and new tasks to ensure good average performance, it often falls short in cost-sensitive applications, where failing to preserve essential established capabilities introduces unforeseen costs and risks and substantial expenses for re-improving these capabilities. To address this issue, we impose a requirement on learning systems to ensure that a new model strictly retains important capabilities of the old model while improving target-task performance, which we term model developmental safety. To ensure model developmental safety, we propose a retention-centric framework with data-dependent constraints, and study how to continually develop a pretrained CLIP model for acquiring new or improving existing capabilities of image classification. We propose an efficient constrained optimization algorithm with theoretical guarantee and use its insights to finetune a CLIP model with task-dependent heads for promoting the model developmental safety. Our experiments on improving vision perception capabilities on autonomous driving and scene recognition datasets demonstrate the efficacy of the proposed approach.

## 1 Introduction

Learning-enabled systems are rapidly transforming various sectors, with applications in autonomous vehicles, medical diagnosis, and financial prediction. These systems often rely on ML models that are trained on vast amounts of data. However, the inherent complexity of the environments in which these systems operate often presents critical challenges, e.g., dealing with corner cases and rare scenarios that deviate from the norm. Additionally, real-world scenarios continuously evolve, presenting new challenges and requiring the system to adapt. These necessitate an iterative development process where models are constantly refined and improved based on new data. Continuously updating the model has become a norm especially in the era of large foundation models, e.g., ChatGPT has experienced several cycles of development from GPT-3.5 to GPT-4 and GPT-4o and recent GPT-o1.

However, this iterative model development process raises a significant issue, i.e., the model development for improving the existing capabilities or acquiring new capabilities may inadvertently lose the previously acquired capabilities of the old model. This issue has been widely observed and documented as catastrophic forgetting when models are trained to learn a sequence of contents (McCloskey & Cohen, 1989). Tremendous studies have been conducted to mitigate the forgetting problem in continual learning literature (Zhou et al., 2022; Rolnick et al., 2019; Shin et al., 2017; Li & Hoiem, 2016; Kirkpatrick et al., 2017). However, these works primarily focus on mitigating the catastrophic forgetting problem, by trading off performance on previous tasks and new tasks to have good average performance (Wang et al., 2024), but do not strictly retain critical existing abilities (i.e., ensuring zero forgetting) while learning new tasks. Ensuring zero forgetting is crucial for many cost-sensitive applications, as failure to strictly preserve the model's essential capabilities not only introduces unforeseen costs and risks but also imposes substantial expenses in the re-improving of these measures, such as in domains like autonomous driving, where established capabilities are usually critical, and

re-validation and re-verification is complex and could cost billions of dollar (Rajabli et al., 2020; Koopman & Wagner, 2016; Company, 2023). This presents a significant challenge for iterative model development process.

To address this challenge, this paper formally introduces **model developmental safety (MDS)** as a requirement of a learning system that in the model development process the new model should strictly retain the existing important capabilities of the old model while improving its performance on target tasks. This concept subtly differs from trading off performance between previous tasks and new tasks to have good average performance of existing continual learning approaches. Moreover, MDS cannot be achieved by the naive weighting method that optimizes a weighted loss by combining protected and target tasks' losses and tuning the weight to preserve protected capabilities. This approach does not necessarily retain the old model's performance across **all** protected tasks even if the weight is large enough, observed in our experiments (Table 6), and will yield no improvement on target tasks if the weight is too large. A more effective algorithm is required to efficiently find a model that not only retains the performance on protected tasks but also improves the performance on target tasks. To the best of our knowledge, no such algorithm currently exists.

This paper aims to address this critical gap by introducing a novel retention-centric framework to ensure MDS. We propose to formulate the MDS as data-dependent constraints, which offers statistical guarantee for strict preservation of performance for all protected tasks. With this framework, we explore developing a pretrained CLIP model (Radford et al., 2021a) for acquiring new capabilities or improving existing ones in image classification. We propose an efficient constrained optimization algorithm with theoretical guarantee. With insights from theoretical analysis, we finetune the CLIP model with task-dependent heads to facilitate MDS. Finally, we demonstrate the efficacy of our approach on autonomous driving dataset and scene recognition dataset, highlighting the practical importance of MDS in real-world scenarios. Our contributions are summarized below:

- We introduce a retention-centric framework by formulating the MDS as data-dependent constraints, which offer statistical guarantee for strictly preserving performance for every protected task.

- We propose an efficient constrained optimization algorithm with theoretical guarantee to develop a pretrained CLIP model for acquiring new or improving existing capabilities of image classification.

- We conduct comprehensive experiments to study the proposed algorithm and compare it with existing baselines to demonstrate its effectiveness in improving vision-based perception capabilities in autonomous driving and scene recognition.

## 2 Related Work

**Continual learning.** This work is closely related to Continual learning (CL), also known as lifelong learning, yet it exhibits nuanced differences. Continual learning usually refers to learning a sequence of tasks one by one and accumulating knowledge like human instead of substituting knowledge (Wang et al., 2024; Qu et al., 2021). The core issue in CL is known as catastrophic forgetting (McCloskey & Cohen, 1989), i.e., the learning of the later tasks may **significantly** degrade the performance of the model for the earlier tasks. There is a vast literature of CL of deep neural networks (DNNs) (Aljundi et al., 2018; Lopez-Paz & Ranzato, 2017a; Farajtabar et al., 2019; Lee et al., 2017; Guo et al., 2020; Parisi et al., 2018). Different approaches have been investigated to mitigate catastrophic forgetting, including regularization-based approaches (Castro et al., 2018; Kirkpatrick et al., 2017; Zenke et al., 2017; Li & Hoiem, 2016), expansion-based approaches (Zhou et al., 2022; Yan et al., 2021; Li et al., 2019; Rusu et al., 2016), and memory-based approaches(Ibrahim et al., 2024) (Buzzega et al., 2020; Cha et al., 2021; Guo et al., 2020; Lopez-Paz & Ranzato, 2017a; Chaudhry et al., 2019). The framework proposed in this work is similar to conventional memory-based approaches in the sense that both use examples of existing tasks to regulate learning. However, the key difference is that most existing continual learning focuses on the trade-off between learning plasticity and memory stability and aims to find a proper balance between performance on previous tasks and new tasks (Wang et al., 2024). Hence, they do not provide a guarantee for MDS. A recent work (Peng et al., 2023) has proposed an ideal continual learner that never forgets by assuming that all tasks share the same optimal solution. However, it is not implementable for deep learning problems.

**Constrained Learning.** Our work is also related to constrained learning. While most traditional constrained optimization works focus on convex objectives or convex constraints, the research interest recently has been directed to non-convex optimization (Boob et al., 2023; Facchinei et al., 2021; Li et al., 2024; Chamon et al., 2022; Alacaoglu & Wright, 2024), due to its increasing importance in modern machine learning problems, such as in applications concerned with fairness (Cotter et al., 2019), robustness (Robey et al., 2021), and safety (Paternain et al., 2019b) problems. Nevertheless, none of existing algorithms can be directly applied to our large-scale deep learning problem (3), due to either prohibitive running cost or failure to handle biased stochastic gradients caused by compositional structure. We include more discussion in Appendix B.

## 3 Notations and Preliminaries

**Notations.** We consider developing a model $\mathbf{w}$ to improve its capabilities on a target task $\mathbb{T}_o$ (dataset $\mathcal{D}$) while preserving its performance on a set of $m$ protected tasks denoted by $\mathbb{T}_1, \ldots, \mathbb{T}_m$. A task can be as simple as predicting a class for multi-class classification. In the paper, we focus on classification using CLIP models and each task refers to one class. For example, we can consider tasks of detecting different weather conditions in autonomous driving, e.g., foggy, overcast, cloudy, clear, rainy, etc. We assume that each task is associated with a data distribution denoted by $\mathfrak{D}_k$. Let $(\mathbf{x}, y) \sim \mathfrak{D}_k$ denote random data of task $\mathbb{T}_k$ with input $\mathbf{x} \in \mathcal{X}$ (e.g., an image) and output $y \in \mathcal{Y}$ (e.g., its class label). We assume that each protected task has a set of examples denoted by $\mathcal{D}_k = \{(\mathbf{x}_i, y_i)\}_{i=1}^{n_k}$, sampled from $\mathfrak{D}_k$. Let $\ell_k(\mathbf{w}, \mathbf{x}, y) = \ell_k(s(\mathbf{x}; \mathbf{w}), y)$ denote a loss function that measures the loss of the model's prediction $s(\mathbf{x}; \mathbf{w})$ with respect to the groundtruth $y$ for task $k$. For classification, the loss could be zero-one loss $\ell_{0-1}$ that measures the classification error or the cross-entropy loss $\ell_{ce}$ that is differentiable for learning. We will define these losses shortly for using CLIP models. We denote by $\mathcal{L}_k(\mathbf{w}, \mathfrak{D}_k) = \mathbb{E}_{\mathbf{x}, y \sim \mathfrak{D}_k} \ell_k(\mathbf{w}, \mathbf{x}, y)$ as the expected loss, and by $\mathcal{L}(\mathbf{w}, \mathcal{D}_k) = \frac{1}{n_k} \sum_{(\mathbf{x}_i, y_i) \sim \mathcal{D}_k} \ell_k(\mathbf{w}, \mathbf{x}_i, y_i)$ as the empirical loss for task $k$.

**The CLIP model and Contrastive Loss.** We consider optimizing a two-way contrastive loss for each image-text pair $(\mathbf{x}_i, \mathbf{t}_i)$ following Yuan et al. (2022):

$$
\begin{aligned}
L_{\mathrm{ctr}}(\mathbf{w}; \mathbf{x}_i, \mathbf{t}_i, \mathcal{T}_i^-, \mathcal{I}_i^-) := \\
- \tau \log \frac{\exp(E_1(\mathbf{w}, \mathbf{x}_i)^\top E_2(\mathbf{w}, \mathbf{t}_i)/\tau)}{\sum_{\mathbf{t}_j \in \mathcal{T}_i^-} \exp(E_1(\mathbf{w}, \mathbf{x}_i)^\top E_2(\mathbf{w}, \mathbf{t}_j)/\tau)} \\
- \tau \log \frac{\exp(E_2(\mathbf{w}, \mathbf{t}_i)^\top E_1(\mathbf{w}, \mathbf{x}_i)/\tau)}{\sum_{\mathbf{x}_j \in \mathcal{I}_i^-} \exp(E_2(\mathbf{w}, \mathbf{t}_i)^\top E_1(\mathbf{w}, \mathbf{x}_j)/\tau)},
\end{aligned}
\tag{1}
$$

where $E_1(\mathbf{w}, \mathbf{x})$ and $E_2(\mathbf{w}, \mathbf{t})$ denotes a (normalized) encoded representation of a image $\mathbf{x}$, and a text $\mathbf{t}$, respectively, $\mathcal{T}_i^-$ denotes the set of all texts to be contrasted with respect to (w.r.t) $\mathbf{x}_i$ (including itself) and $\mathcal{I}_i^-$ denotes the set of all images to be contrasted w.r.t $\mathbf{t}_i$ (including itself).

To utilize a CLIP model for multi-class classification with classes $\mathcal{C} = \{c_1, \ldots, c_K\}$, we will convert a class $c_k$, e.g., "rainy", into a text description of $c_k$, denoted by $\hat{\mathbf{t}}_k$, e.g., "the weather is rainy", similar to the zero-shot classification scheme of the CLIP model. Hence, a prediction score (i.e., a logit) for an image $\mathbf{x}$ and a text description $\hat{\mathbf{t}}_k$ of class $c_k$ is calculated by $s_k(\mathbf{x}; \mathbf{w}) = E_1(\mathbf{w}, \mathbf{x})^\top E_2(\mathbf{w}, \hat{\mathbf{t}}_k)$. The predicted class label is given by $\hat{y} = \arg\max_{c_k \in \mathcal{C}} s_k(\mathbf{x}; \mathbf{w})$. Hence, given the true class $y \in \mathcal{C}$, the zero-one loss is given by $\ell_{0,1}(\mathbf{w}, \mathbf{x}, y) = \mathbb{I}(\hat{y} \neq y)$, and the cross-entropy loss is given by $\ell_{ce}(\mathbf{w}, \mathbf{x}, y) = -\log \frac{\exp(s_y(\mathbf{x}; \mathbf{w})/\tau_0)}{\sum_{\ell=1}^K \exp(s_l(\mathbf{x}; \mathbf{w})/\tau_0)}$, where $\tau_0 > 0$ is a temperature parameter that controls the balance between the approximation error of the zero-one loss and the smoothness of the function.

## 4 A Retention-Centric Framework

To measure model developmental safety, it is necessary to evaluate how the model's performance changes in protected tasks from the old model $\mathbf{w}_{\mathrm{old}}$ to a new model $\mathbf{w}_{\mathrm{new}}$. We introduce the formal definition of model developmental safety (MDS) in Definition 1, which ensures the new model strictly preserves performance on every protected task.

**Definition 1 (Model Developmental Safety (MDS))** *In model development process, the model developmental safety is satisfied if* $\mathcal{L}_k(\mathbf{w}_{new}, \mathfrak{D}_k) \leq \mathcal{L}_k(\mathbf{w}_{old}, \mathfrak{D}_k), \forall k \in \{1, \ldots, m\}$ *, where* $\mathcal{L}_k(\mathbf{w}, \mathfrak{D}_k) = \mathbb{E}_{\mathbf{x}, y \sim \mathfrak{D}_k} \ell_k(\mathbf{w}, \mathbf{x}, y)$.

In practice, model developmental safety will be measured using a set of empirical examples $\mathcal{S}_j \sim \mathfrak{D}_j$ for each protected task. Hence, we define the empirical developmental safety metric, corresponding to Definition 1, for evaluation:

$$\text{DevSafety} = \min_{k \in \{1, \cdots, m\}} \left( \mathcal{L}_k(\mathbf{w}_{\text{old}}, \mathcal{S}_k) - \mathcal{L}_k(\mathbf{w}_{\text{new}}, \mathcal{S}_k) \right).$$

When using the zero-one loss $\ell_{0-1}$ in the above definitions, we refer to the metric above as DevSafety(acc).

The key to our retention-centric framework is to utilize examples of protected tasks to define empirical retention constraints when updating the model on a target task. To improve the model performance on a target task $\mathbb{T}_o$, we assume that a set of data $\mathcal{D}$ for $\mathbb{T}_o$ is constructed and a proper objective is given based on application, denoted by $F(\mathbf{w}, \mathcal{D})$. Then, our retention-centric approach for model development is imposed by solving the following problem:

$$\mathbf{w}_{\text{new}} = \arg \min_{\mathbf{w}} \quad F(\mathbf{w}, \mathcal{D})$$

$$\text{s.t.} \quad \mathcal{L}_k(\mathbf{w}, \mathcal{D}_k) - \mathcal{L}_k(\mathbf{w}_{\text{old}}, \mathcal{D}_k) \leq 0, \ k = 1, \cdots, m. \tag{2}$$

We will propose an efficient algorithm to directly solve this data-dependent constrained optimization problem in the next section.

**Generalization Analysis.** Since, in practice, we can only use empirical data $\mathcal{D}_1, \ldots, \mathcal{D}_m$ in Eqn. (2), there exist generalization errors between the retention constraints in Eqn. (2) and the MDS we want to ensure in Def. 1. The lemma below uses a standard tool of statistical error analysis to bound the generalization error of retention. For simplicity, we assume each protected task is associated with the same loss function, namely, $\ell_k = \ell$ for $k = 1, \ldots, m$. In the analysis, we use the Rademacher complexity of the loss class $\mathcal{H} = \{\ell(\mathbf{w}, \cdot, \cdot) : \mathcal{X} \times \mathcal{Y} \to [0,1] | \mathbf{w} \in \mathbb{R}^d\}$ induced by the model $\mathbf{w}$ on $n$ data points, which is denoted by $R_n(\mathcal{H})$. We assume that $R_n(\mathcal{H}) \leq Cn^{-\alpha}$ for some $C \geq 0$ and $\alpha \leq 0.5$. We note that $\alpha = 0.5$ in the vast majority of model and loss families, including linear models (Kakade et al., 2008), deep neural networks (Bartlett & Mendelson, 2002), and model families with bounded VC dimension (Bartlett & Mendelson, 2002).

**Lemma 1 (Generalization Error of Retention)** *Suppose that $R_n(\mathcal{H}) \leq Cn^{-\alpha}$ for some $C \geq 0$ and $\alpha \leq 0.5$. Then, with probability at least $1 - \delta$, , $\forall k$ it holds that*

$$\mathcal{L}_k(\mathbf{w}_{new}, \mathfrak{D}_k) - \mathcal{L}_k(\mathbf{w}_{old}, \mathfrak{D}_k)$$

$$\leq \mathcal{L}_k(\mathbf{w}_{new}, \mathcal{D}_k) - \mathcal{L}_k(\mathbf{w}_{old}, \mathcal{D}_k) + \frac{4C}{n_k^\alpha} + 2\sqrt{\frac{\ln(2m/\delta)}{2n_k}}.$$

**Remark:** The lemma indicates that as long as the empirical retention constraints are satisfied, i.e., $\mathcal{L}_k(\mathbf{w}_{\text{new}}, \mathcal{D}_k) - \mathcal{L}_k(\mathbf{w}_{\text{old}}, \mathcal{D}_k) \leq 0$, the model developmental safety is ensured up to a statistical error in the order of $O(n^{-\alpha})$, where $n = \min_k n_k$. Hence, the more examples used to construct the constraints, the more likely the new model meets MDS requirement. The proof is given in C.1.

## 5 Retention-Centric Development of CLIP

In this section, we present an efficient algorithm for improving a pretrained CLIP model on a target task while ensuring MDS on a set of protected tasks. The CLIP model is of particular interest because (i) it is a foundation model that has been used extensively in many applications; and (ii) can adapt to the open-world for handling new classes using languages. However, existing studies have shown that directly applying a pretrained CLIP model (e.g., OpenAI's CLIP model) to a certain downstream application yields varying performance across different classes (Parashar et al., 2024). Rare concepts (e.g., foggy) usually has worse performance than frequent concepts (e.g., clear), making it necessary to continuously update.

Suppose a CLIP model $\mathbf{w}_{\text{old}}$ has been trained. We aim to improve it for a target task $\mathbb{T}_o$ (e.g., classifying foggy). To this end, we collect a set of image-text pairs related to the target task, denoted by $\mathcal{D} = \{(\mathbf{x}_i, \mathbf{t}_i)\}_{i=1}^{n_o}$. As labeled data for rare scenarios (e.g., *foggy*) are usually limited in practice, we consider augmenting the dataset $\mathcal{D}$ by using a query prompt to search for target-related image-text pairs from the internet (detailed in Appendix A.2). For each image-text pair, a set of negative texts has been collected to be contrasted w.r.t.

$\mathbf{x}_i$, which together with $\mathbf{t}_i$ form $\mathcal{T}_i^-$, and a set of negative images has been also collected to be contrasted w.r.t. $\mathbf{t}_i$, which together with $\mathbf{x}_i$ form $\mathcal{I}_i^-$.

To develop the CLIP model in our retention-centric framework, we instantiate (2) as:

$$\min_{\mathbf{w}} \quad F(\mathbf{w}, \mathcal{D}) := \frac{1}{n_o} \sum_{(\mathbf{x}_i, \mathbf{t}_i) \in \mathcal{D}} L_{\mathrm{ctr}}(\mathbf{w}; \mathbf{x}_i, \mathbf{t}_i, \mathcal{T}_i^-, \mathcal{I}_i^-)$$

$$\text{s.t.} \quad h_k(\mathbf{w}) := \mathcal{L}_k(\mathbf{w}, \mathcal{D}_k) - \mathcal{L}_k(\mathbf{w}_{\mathrm{old}}, \mathcal{D}_k) \leq 0, \ k \in [m] \tag{3}$$

### 5.1 Efficient Optimization

The optimization problem in (3) is challenging for multiple reasons. First, this problem involves a non-convex objective and non-convex constraints, so finding a global optimal solution is intractable in general. Second, the objective and constraint functions are formulated using a large dataset, so we need to sample from the dataset in order to construct stochastic gradients of the functions to update the solution. Lastly, (3) may contain a large number of constraints, so updating the solutions using the gradients of all constraints may be prohibited. Given these challenges, we need to develop a stochastic optimization for (3) based on advanced techniques and constraint sampling.

Our method is motivated by the stochastic quadratic penalty method in (Alacaoglu & Wright, 2024), which first converts (3) into an unconstrained problem by adding a quadratic penalty on the constraints violation to the objective function and then solves the unconstrained problem using a variance-reduced stochastic gradient method. Unfortunately, their method can not be directly applied to (3) because (i) they only consider equality constraints while (3) involves inequality constraints and (ii) they require an unbiased stochastic gradients for each update while the stochastic gradients for (3) will be biased due to the compositional structure.

A quadratic penalty method converts (3) into the following unconstrained problem:

$$\min_{\mathbf{w}} \Phi(\mathbf{w}) := F(\mathbf{w}, \mathcal{D}) + \frac{1}{m} \sum_{k=1}^{m} \frac{\beta}{2} ([h_k(\mathbf{w})]_+)^2 \tag{4}$$

where $[\cdot]_+ = \max\{\cdot, 0\}$ and $\beta \geq 0$ is the penalty parameter. Under mild conditions(Bertsekas, 2014), a large enough $\beta$ will ensure the optimal solution to (4) is also an optimal solution to (3). In the following, we introduce an efficient stochastic algorithm to solve (4). It is notable that both terms are of finite-sum coupled compositional structure (Wang & Yang, 2022), i.e., $\sum_i f(g_i(\mathbf{w}))$, where $f$ is non-linear.

We discuss how to approximate the gradient of two terms of the objective using mini-batch samples below. Let $s_{ij}^I = E_1(\mathbf{w}, \mathbf{x}_i)^\top E_2(\mathbf{w}, \mathbf{t}_j), s_{ij}^T = E_2(\mathbf{w}, \mathbf{t}_i)^\top E_1(\mathbf{w}, \mathbf{x}_j)$. Define

$$g_{1i}(\mathbf{w}) = g_{1i}(\mathbf{w}, \mathcal{T}_i^-) = \frac{1}{|\mathcal{T}_i^-|} \sum_{\mathbf{t}_j \in \mathcal{T}_i^-} \exp((s_{ij}^I - s_{ii}^I)/\tau), \quad g_{2i}(\mathbf{w}) = g_{2i}(\mathbf{w}, \mathcal{I}_i^-) = \frac{1}{|\mathcal{I}_i^-|} \sum_{\mathbf{x}_j \in \mathcal{I}_i^-} \exp((s_{ij}^T - s_{ii}^T)/\tau).$$

Then, we have $F(\mathbf{w}, \mathcal{D}) = \frac{1}{n_o} \sum_{(\mathbf{x}_i, \mathbf{t}_i) \in \mathcal{D}} \tau \log g_{1i}(\mathbf{w}) + \tau \log g_{2i}(\mathbf{w})$ and its gradient is given by

$$\nabla F(\mathbf{w}, \mathcal{D}) = \frac{\tau}{n_o} \sum_{(\mathbf{x}_i, \mathbf{t}_i) \in \mathcal{D}} \left( \frac{\nabla g_{1i}(\mathbf{w})}{g_{1i}(\mathbf{w})} + \frac{\nabla g_{2i}(\mathbf{w})}{g_{2i}(\mathbf{w})} \right).$$

The major cost of computing $\nabla F(\mathbf{w}; \mathcal{D})$ lies on calculating $g_{1i}(\mathbf{w})$ and $g_{2i}(\mathbf{w})$ and their gradient for each pair, as it involves all the samples in $\mathcal{T}_i^-$ and $\mathcal{I}_i^-$. Directly approximating $g_{1i}$ and $g_{2i}$ by a mini-batch of samples from $\mathcal{T}_i^-$ and $\mathcal{I}_i^-$ will reduce the computational cost but lead to a biased stochastic gradient of $\nabla F(\mathbf{w}, \mathcal{D})$ due to the non-linear dependence of $\nabla F(\mathbf{w}; \mathcal{D})$ on $g_{1i}$ and $g_{2i}$, causing the issue of requiring a large batch size in order to converge (Yuan et al., 2022).

To address this issue, we employ the moving average estimators for estimating $g_{1i}$ and $g_{2i}$ which gradually reduces the aforementioned biases to zero (Yuan et al., 2022). More specifically, let $\mathbf{w}^t$ be the solution at iteration $t$. We randomly sample a mini batch $\mathcal{B} \subset \mathcal{D}$, and construct mini-batch negatives $\mathcal{B}_{1,i} \subset \mathcal{T}_i^-$, $\mathcal{B}_{2,i} \subset \mathcal{I}_i^-$ for each data $(\mathbf{x}_i, \mathbf{t}_i) \in \mathcal{B}$ and construct the following stochastic estimations of $g_{1i}(\mathbf{w}^t)$ and $g_{2i}(\mathbf{w}^t)$, denoted by $\hat{g}_{1i}(\mathbf{w}^t) = g_{1i}(\mathbf{w}^t, \mathcal{B}_{1,i})$ and $\hat{g}_{2i}(\mathbf{w}^t) = g_{2i}(\mathbf{w}^t, \mathcal{B}_{2,i})$. The moving averaging estimators of $g_{1i}(\mathbf{w}^t)$ and $g_{2i}(\mathbf{w}^t)$ denoted by $u_{1i}^t$ and $u_{2i}^t$ are updated by:

$$u_{1i}^{t+1} = (1 - \gamma_1) u_{1i}^t + \gamma_1 \hat{g}_{1i}(\mathbf{w}^t)$$
$$u_{2i}^{t+1} = (1 - \gamma_1) u_{2i}^t + \gamma_1 \hat{g}_{2i}(\mathbf{w}^t), \tag{5}$$

---

**Algorithm 1** Algorithm for solving (3)

---

1: **Initialization:** choose $\mathbf{w}^0$, $\beta$, $\gamma_1$, $\gamma_2$, $\theta$ and step size $\eta$.
2: **for** $t = 0, 1, \cdots, T - 1$ **do**
3:     Sample image-text pairs $\mathcal{B}$ from $\mathcal{D}$ and protected tasks $\mathcal{B}_c$ from $\{1, \cdots, m\}$.
4:     **for** each $(\mathbf{x}_i, \mathbf{t}_i) \in \mathcal{B}$ **do**
5:       Update $u_{1i}^t$ and $u_{2i}^t$ by Eqn. (5)
6:     **end for**
7:     Update $G_1^t$ with Eqn. (6)
8:     **for** each $k \in \mathcal{B}_c$ **do**
9:       Sample a batch of data from $\mathcal{D}_k$ denoted by $\mathcal{B}_k$.
10:      Update the estimators of $h_k$ by Eqn. (8).
11:     **end for**
12:    Compute $G_2^t$ with (7)
13:    Update $v^{t+1} = (1 - \theta)v^t + \theta(G_1^t + G_2^t)$
14:    Update $\mathbf{w}$ by $\mathbf{w}^{t+1} = \mathbf{w}^t - \eta v^{t+1}$.
15: **end for**

---

where $\gamma_1 \in (0, 1)$ is a hyper-parameter. The gradient estimator of $F(\mathbf{w}^t, \mathcal{D})$ is computed by

$$G_1^t = \frac{\tau}{|\mathcal{B}|} \sum_{i \in \mathcal{B}} \left( \nabla \hat{g}_{1i} \left( \mathbf{w}^t \right) / u_{1i}^t + \nabla \hat{g}_{2i} \left( \mathbf{w}^t \right) / u_{2i}^t \right). \tag{6}$$

The gradient of the quadratic penalized term at $\mathbf{w}^t$ can be approximated similarly by

$$G_2^t = \frac{1}{|\mathcal{B}_c|} \sum_{k \in \mathcal{B}_c} \beta [u_k^t]_+ \nabla \hat{h}_k(\mathbf{w}^t), \tag{7}$$

where $\mathcal{B}_c$ denotes a sampled subset of protected tasks, $\hat{h}_k(\mathbf{w}^t)$ denotes a mini-batch estimator of $h_k(\mathbf{w}^t)$ using mini-batch $\mathcal{B}_k \subset \mathcal{D}_k$, and $u_k^t$ is the moving average estimator of $h_k(\mathbf{w}^t)$ computed by

$$u_k^{t+1} = (1 - \gamma_2)u_k^t + \gamma_2 \hat{h}_k(\mathbf{w}^t), \tag{8}$$

$$\hat{h}_k(\mathbf{w}^t) = \frac{1}{|\mathcal{B}_k|} \sum_{j \in \mathcal{B}_k} \ell_{ce}(\mathbf{w}, \mathbf{x}_j, y_j) - \ell_{ce}(\mathbf{w}_{\text{old}}, \mathbf{x}_j, y_j).$$

We highlight that the gradient estimator in (7) for protected tasks, where each task has a dynamically changing effective weight $\beta [u_k^t]_+$ during the learning process, is the key distinction from the naive weighting method mentioned earlier.

The key steps are presented in Algorithm 1. We would like to emphasize Algorithm 1 can be easily generalized to handling a general objective function $F(\mathbf{w})$ whose unbiased stochastic gradient is available by just replacing $G_1^t$ with the stochastic gradient estimator of $F(\mathbf{w}_t)$.

### 5.2 Convergence Analysis

Since our considered constrained optimization problem is non-convex for both objectives and constraints, a critical concern is whether Algorithm 1 has some convergence guarantee as standard learning algorithms such as SGD/Adam. We address this in this section. To the best of our knowlege, this is the first convergence analysis of a penalty method for solving non-convex inequaliaty constrained optimization. For analysis, we make the following assumptions.

**Assumption 1** *(a) $g_1(\cdot)$ and $g_2(\cdot)$ are $L_g$-Lipschitz continuous and $L_{\nabla g}$-smooth. (b) There exist $C_g > 0$ and $c_g > 0$ such that $c_g \leq \min\{g_1(\cdot), g_2(\cdot)\}$ and $\max\{g_1(\cdot), g_2(\cdot)\} \leq C_g$. (c) $h_k(\cdot)$ is $L_h$-Lipschitz continuous and $L_{\nabla h}$-smooth for $k = 1, \cdots, m$.*

**Assumption 2** *(a) $\mathbb{E}[\|\hat{g}_{1i}(\mathbf{w}) - g_{1i}(\mathbf{w})\|^2] \leq \sigma_g^2/|\mathcal{B}_{1i}|$, $\mathbb{E}[\|\hat{g}_{2i}(\mathbf{w}) - g_{2i}(\mathbf{w})\|^2] \leq \sigma_g^2/|\mathcal{B}_{2i}|$; (b) $\mathbb{E}[\|\nabla \hat{g}_1(\mathbf{w}) - \nabla g_{1i}(\mathbf{w})\|^2] \leq \sigma_{\nabla g}^2/|\mathcal{B}_{1i}|$, $\mathbb{E}[\|\nabla \hat{g}_{2i}(\mathbf{w}) - \nabla g_{2i}(\mathbf{w})\|^2] \leq \sigma_{\nabla g}^2/|\mathcal{B}_{2i}|$; (c) $\mathbb{E}[\|\nabla \hat{h}_k(\mathbf{w}) - \nabla h_k(\mathbf{w})\|^2] \leq \sigma_{\nabla h}^2/|\mathcal{B}_k|$; (d) $\mathbb{E}[\|\hat{h}_k(\mathbf{w}) - h_k(\mathbf{w})\|^2] \leq \sigma_h^2/|\mathcal{B}_k|$ for $k = 1, \cdots, m$.*

**Assumption 3** *There exists a constant $\delta > 0$ such that $\|\nabla \boldsymbol{h}(\mathbf{w}^t)[\boldsymbol{h}(\mathbf{w}^t)]_+\| \geq \delta \|[\boldsymbol{h}(\mathbf{w}^t)]_+\|$ for $t = 0, \cdots, T$, where $\boldsymbol{h}(\mathbf{w}) = [h_1(\mathbf{w}), \ldots, h_m(\mathbf{w})]^\top$ and $\nabla \boldsymbol{h}(\mathbf{w}) = [\nabla h_1(\mathbf{w}), \ldots, \nabla h_m(\mathbf{w})]$.*

**Remark:** Assumption 1 has been justified in the earlier work (Yuan et al., 2022; Qiu et al., 2023) for optimizing a global contrastive loss. Assumption 2 is a standard one that bounds the variance of mini-batch estimators. Assumption 3 is also made in many existing works on optimization with non-convex constraints (Sahin et al., 2019; Xie & Wright, 2021; Alacaoglu & Wright, 2024; Lin et al., 2022; Li et al., 2024). This assumption is equivalent to that the quadratic penalty term $H(\mathbf{w}) := \frac{\beta}{2m}\|[\mathbf{h}(\mathbf{w})]_+\|^2$ satisfies the Polyak-Lojasiewicz inequality at $\mathbf{w} = \mathbf{w}^t$, meaning that there exists $\delta \geq 0$ such that $\|\nabla H(\mathbf{w}^t)\|^2 \geq \frac{2\delta^2\beta}{m}H(\mathbf{w}^t)$. Without this assumption, Eqn.(3) may be intractable because there may exist an iterate $\mathbf{w}^t$ such that $H(\mathbf{w}^t) > 0$ but $\nabla H(\mathbf{w}^t) = 0$, meaning that $\mathbf{w}^t$ is infeasible but at a flat location of $H(\mathbf{w})$ so $\mathbf{w}^t$ may get trapped at this location forever. We will show later that a small $\delta$ in Assumption 3 will increase the complexity of our algorithm. Hence, we will present an approach in next subsection to increase $\delta$.

For a non-convex optimization problem like (3), finding a globally optimal solution is intractable, so almost all numerical algorithms for non-convex problems can only guarantee a Karush-Kuhn-Tucker (KKT) solution defined below.

**Definition 2** *A solution $\mathbf{w}$ is a KKT solution to (3) if there exist $\boldsymbol{\lambda} = (\lambda_1, \ldots, \lambda_m)^\top \in \mathbb{R}_+^m$ such that $\nabla F(\mathbf{w}, \mathcal{D}) + \nabla \boldsymbol{h}(\mathbf{w})\boldsymbol{\lambda} = \mathbf{0}$, $\boldsymbol{h}(\mathbf{w}) \leq \mathbf{0}$ and $\lambda_k h_k(\mathbf{w}) = 0$ for $\forall k$.*

We present the convergence theorem of Algorithm 1 as follows, which shows the iteration complexity of Algorithm 1 for finding an $\epsilon$-KKT solution, i.e., a solution satisfying the three conditions in Definition 2 up to $\epsilon$ precision. The proof of the theorem is presented in Appendix C.3.

**Theorem 1** *Suppose Assumptions 1, 2 and 3 hold. Also, suppose, in Algorithm 1, set $\beta = \frac{1}{\epsilon\delta}$, $\gamma_1 = \gamma_2 = \min\{\frac{5n_0\theta}{3|\mathcal{B}|}, \frac{5m\theta}{3|\mathcal{B}_c|}, \frac{\epsilon^4\delta^2|\mathcal{B}_k|}{26880\sigma_h^2\tilde{C}_{\nabla h}^2}\}$, $\theta = \min\{\frac{\epsilon^4\delta^2\min\{|\mathcal{B}_c|,|\mathcal{B}_k|\}}{672(\sigma_{\nabla h}^2+L_h^2)}, \frac{\epsilon^2\min\{|\mathcal{B}|,|\mathcal{B}_{1i}|,|\mathcal{B}_{2i}|\}}{1344L_f^2(\sigma_{\nabla g}^2+L_g^2)}\}$ and $\eta = \min\{\frac{1}{12(L_F+\beta L_H)}, \frac{\theta}{8\sqrt{3}L_F}, \frac{\theta}{8\sqrt{3}L_H\beta}, \frac{\gamma_1|\mathcal{B}|}{40\sqrt{6}L_gL_f\tilde{C}_{\nabla g}n_0}, \frac{\gamma_2|\mathcal{B}_c|}{40\sqrt{6}\beta L_h\tilde{C}_{\nabla h}m}\}$, where $\tilde{C}_{\nabla g} := \sigma_{\nabla g}+L_g$, $\tilde{C}_{\nabla h} := \sigma_{\nabla h}+L_h$, $L_f := \frac{\tau}{c_g}$, $L_{\nabla f} := \frac{\tau}{c_g^2}$, $L_F := 2(L_{\nabla g}L_f + L_{\nabla f}L_g^2)$ and $L_H := 2L_{\nabla h}+L_h^2$. Then there exists $\boldsymbol{\lambda} \in \mathbb{R}_+^m$ such that after $T = O(\epsilon^{-7}\delta^{-3})$ iterations Algorithm 1 satisfies $\mathbb{E}\left[\|\nabla F(\mathbf{w}^{\hat{t}}, \mathcal{D}) + \nabla \boldsymbol{h}(\mathbf{w}^{\hat{t}})\boldsymbol{\lambda}\|\right] \leq \epsilon$, $\mathbb{E}[\|[\boldsymbol{h}(\mathbf{w}^{\hat{t}})]_+\|] \leq \epsilon$, $\mathbb{E}[\boldsymbol{\lambda}^\top[\boldsymbol{h}(\mathbf{w}^{\hat{t}})]_+] \leq \epsilon$, where $\hat{t}$ selected uniformly at random from $\{1, \cdots, T\}$.*

**Remark:** It is notable that the order of complexity in terms of $\epsilon$ is higher than that of standard learning (i.e., $O(\epsilon^{-4})$). While the complexity for a stochastic constrained optimization could be inherently higher than unconstrained optimization (Alacaoglu & Wright, 2024), we note that the above complexity is also weaker than the state-of-the-art complexity of stochastic constrained optimization (Alacaoglu & Wright, 2024). We remark that this is a limitation of the present work due to two reasons: (i) we use the moving average gradient estimator for sake of implementation; in contrast, they use the advanced variance reduced gradient estimator (STORM), which incurs additional overhead; (ii) we use a constant $\beta$ and they use an increasing $\beta$. In our experiments shown in ablation studies, we find that using a constant $\beta$ is generally better than using an increasing $\beta$. Additionally, the dependence on $\delta$ could also slow down the convergence. We mitigate this issue by utilizing task-dependent heads for CLIP models justified below.

## 5.3 Promoting Developmental Safety via Task-dependent Heads

Below, we present a way to design the text encoder of the CLIP model such that the value of $\delta$ could be larger. Without causing confusion, we denote by $\mathbf{w}$ the parameter of the text encoder, which consists of two components $\mathbf{u}$ and $W$ such that the text embedding $E_2(\mathbf{w}, \mathbf{t}) \in \mathbb{R}^{d_2}$ can be represented as $E_2(\mathbf{w}, \mathbf{t}) = W \cdot \bar{E}_2(\mathbf{u}, \mathbf{t})$, where $\bar{E}_2(\mathbf{u}, \cdot) \in \mathbb{R}^{d_1}$ is a backbone encoder while $W \in \mathbb{R}^{d_2 \times d_1}$ is called the head. The idea of task-dependent heads is to let each task $k$ have its own head $W_k = W + U_kV_k^\top$ using low rank matrices $U_k \in \mathbb{R}^{d_2 \times r}$ and $V_k \in \mathbb{R}^{d_1 \times r}$, where $r < \min(d_1, d_2)$ is the rank chosen as a hyper-parameter. The output of this class-specific text encoder for task $k$ is $E_2(\mathbf{u}, W, U_k, V_k, \mathbf{t}_k) = (W + U_kV_k^\top) \cdot \bar{E}_2(\mathbf{u}, \mathbf{t}_k)$. Note that $\|\nabla\mathbf{h}(\mathbf{w}^t)^\top[\mathbf{h}(\mathbf{w}^t)]_+\|^2 \geq \lambda_{\min}(\nabla\mathbf{h}(\mathbf{w}^t)^\top\nabla\mathbf{h}(\mathbf{w}^t))\|[\mathbf{h}(\mathbf{w}^t)]_+\|^2$, where $\lambda_{\min}(\cdot)$ represents the smallest eigenvalue of a matrix. This means $\min_t \lambda_{\min}(\nabla\mathbf{h}(\mathbf{w}^t)^\top\nabla\mathbf{h}(\mathbf{w}^t))$ is a lower bound of $\delta$ in Assumption 3. The following lemma shows that, after expanding $\mathbf{w}$ with $U_k$ and $V_k$, $\lambda_{\min}(\nabla\mathbf{h}(\mathbf{w}^t)^\top\nabla\mathbf{h}(\mathbf{w}^t))$ may increase at some $U_k$ and $V_k$, providing some insight on why the task-dependent heads help to increase the parameter $\delta$ in Assumption 3, reducing the total complexity of our algorithm according to Theorem 1.

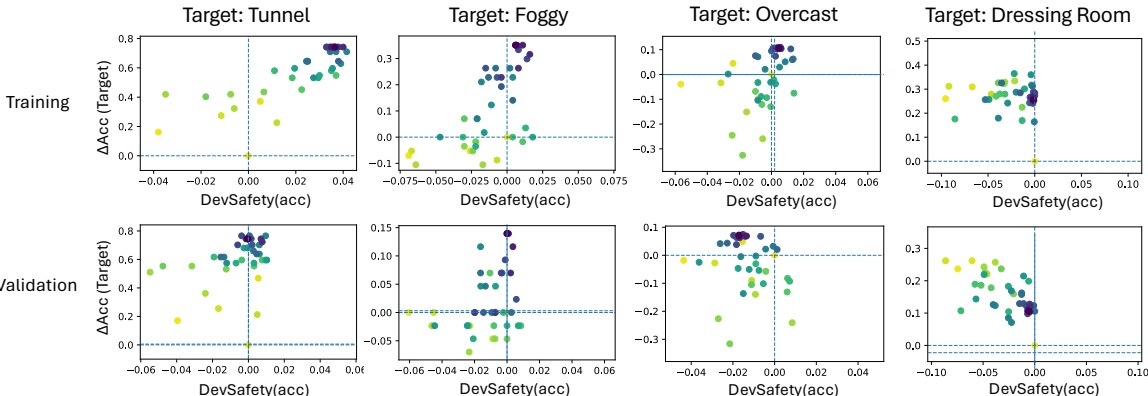

Figure 1: Visualization of the learning trajectory. Each dot denotes a solution with lighter color being earlier iterations and darker being later iterations.

**Lemma 2** *Let* $\mathbf{U} = (U_1, \ldots, U_m)$ *and* $\mathbf{V} = (V_1, \ldots, V_m)$. *Let* $\mathbf{w} = (W, \mathbf{u})$, $\hat{\mathbf{w}} = (W, \mathbf{u}, \mathbf{U}, \mathbf{V})$, $h_k(\mathbf{w}) = h_k(W, \mathbf{u})$, *and* $\hat{h}_k(\hat{\mathbf{w}}) = h_k(W + U_k V_k^\top, \mathbf{u})$. *Suppose* $U_k V_k^\top = \mathbf{0}$ *for all* $k$'s. *We have*

$$\lambda_{\min}\left(\nabla\widehat{\boldsymbol{h}}(\hat{\mathbf{w}})^\top \nabla\widehat{\boldsymbol{h}}(\hat{\mathbf{w}})\right) \geq \lambda_{\min}\left(\nabla\boldsymbol{h}(\mathbf{w})^\top \nabla\boldsymbol{h}(\mathbf{w})\right) +$$

$$\min_k \left\{ \left\|\nabla_W h_k(\mathbf{w}) V_k\right\|_F^2, \left\|\nabla_W h_k(\mathbf{w})^\top U_k\right\|_F^2 \right\},$$

*where* $\widehat{\boldsymbol{h}}(\hat{\mathbf{w}}) = [\hat{h}_1(\hat{\mathbf{w}}), \ldots, \hat{h}_m(\hat{\mathbf{w}})]^\top$ *and* $\nabla\widehat{\boldsymbol{h}}(\hat{\mathbf{w}}) = [\nabla\hat{h}_1(\hat{\mathbf{w}}), \ldots, \nabla\hat{h}_m(\hat{\mathbf{w}})]$.

Following this lemma, in our experiments, we employ the task-dependent heads with setting the initial value of $U_k$ to zero so $U_k V_k^\top = \mathbf{0}$. The proof is given in Appendix C.2.

## 6 Experiments

**Dataset.** We experiment on the large-scale driving image dataset, namely BDD100K (Seita, 2018). This dataset involves classification of six weather conditions, i.e., *clear, overcast, snowy, rainy, partly cloudy, foggy,* and of six scene types, i.e., *highway, residential area, city street, parking lot, gas station, tunnel*. We consider three settings with *foggy, overcast* and *tunnel* as the target class separately and other weather conditions or scenes as protected tasks. Moreover, we experiment on the scene recognition dataset Places365 (Zhou et al., 2017) with 365 classes to verify the effectiveness of the proposed method in handling a large number of constraints. We consider *dressing room* as the target class, as it has fewest samples in the dataset. More experimental settings are presented in Appendix A.1.

**Evaluation Metrics.** We measure improvement on target task with $\Delta\text{Acc}(\text{Target}) = \text{Acc}(\text{Target}, \mathbf{w}_{\text{new}}) - \text{Acc}(\text{Target}, \mathbf{w}_{\text{old}})$. Besides, we utilize "DevSafety(acc)" to measure the empirical MDS. As optimization involves randomness, we run all the experiments with five different random seeds then calculate the average target accuracy and the percentage of times that DevSafety(acc) is non-negative, denoted as **Retention Ratio**, to measure the possibility of strictly retaining the old model's performance on protected tasks. For example, the Retention Ratio is 60% if 3 out of 5 runs of the method preserve previous performance for all protected tasks.

**Baselines.** To verify the effectiveness of our algorithms, we compare our proposed algorithm with the following 6 baselines: (1) FLYP (Goyal et al., 2023), a state-of-the-art CLIP continual finetuning method that optimizes a contrastive loss on all available data including those used in our objective and constraints. In our experiments, we utilize the same global contrastive loss (GCL) (Yuan et al., 2022) instead of mini-batch contrastive loss; (2) Weighted Combination of Contrastive Losses (WCCL), which utilizes a weight to combine GCL losses on protected tasks and the target task to control the tradeoff between them; (3) GEM (Lopez-Paz & Ranzato, 2017b), which is a strong CL baseline motivated by a similar idea utilizing data of previous tasks for constraints; (4) Co$^2$L(Cha et al., 2021), which is an advanced contrastive continual learning baseline; (5) DER(Buzzega et al., 2020), a strong memory-based continual learning baseline, which mixes rehearsal

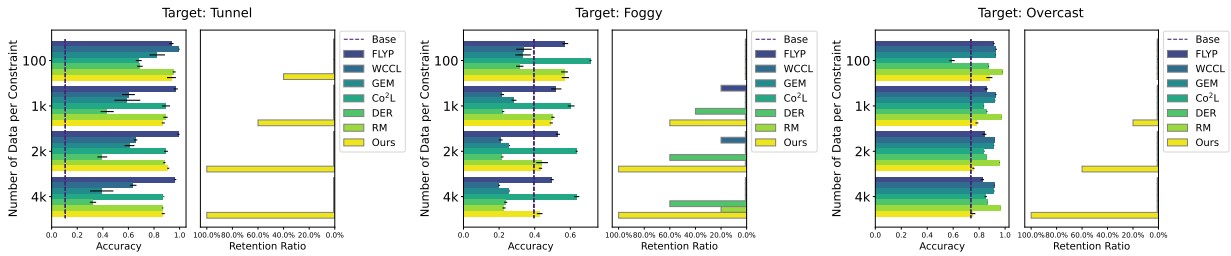

Figure 2: Performance Comparison with Baselines. Dot lines represent the performance of the base model on the target task. We also include a comparison in terms of ΔAcc (Target) and DevSafety(acc) in Fig. 8.

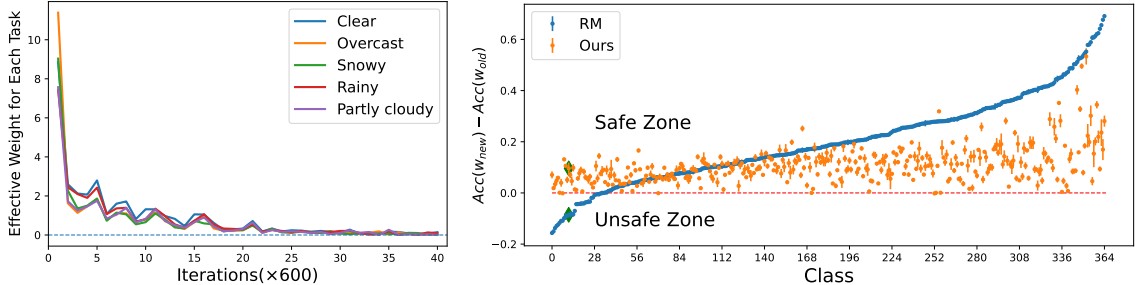

Figure 3: (Left) Adaptive weight adjustments for each protected task during training (Targeting *Foggy*). Weights shown are averaged over every 600 iterations for visualization. (Right) Performance comparison with baseline RM when targeting *Dresssing Room* on Places365 Dataset, with 2k samples per constraint. Red line denotes base model's performance, green diamonds denote target class. RM baseline shown is for weight $\alpha = 10000$ and more plots for other weights are in Fig. 7.

with knowledge distillation and regularization; (6) Regularization Method (RM), as commonly adopted in continual learning literature (Rebuffi et al., 2017; Castro et al., 2018), directly takes the constraints in Eqn. (3) as a regularization term by adding it to the objective function with a regularization weight $\alpha$. All methods start from the same CLIP model. More baselines' details are presented in Appendix A.1.4.

## 6.1 Visualization of Learning Process

To provide a direct understanding of why and how the proposed algorithm works, we present the learning trajectory of the algorithm in Figure 1. Each dot in this figure represents a solution during the learning processing, with lighter colors indicating earlier stages and darker colors representing later stages. From the top four figures for training sets, we can observe a common trend that solutions start from the lower left and move toward the upper right, indicating the algorithm endeavors to enhance the performance of the targeted task while ensuring developmental safety on protected tasks. Similarly, this trend extends to the validation sets, shown in the bottom row, demonstrating the generalization capability of the proposed algorithm. It is striking to see that, when targeting *Dressing Room* in Places365 dataset with all other 364 classes as protected tasks, our method are still able to ensure MDS in training set and generalize to validation set. These observations can also be found in separate views of DevSafety vs epochs and ΔAcc(Target) vs epochs in Figure 6.

## 6.2 Comparison with Baselines

In this part, we compare the proposed method with baselines to demonstrate the superiority. Specifically, we focus on two metrics, i.e., Retention Ratio for measuring the possibility of strictly preserving the performance on all protected tasks and accuracy on the target task. The details of hyperparameter tuning is presented in Appendix A.1.3. On BDD100K dataset, we conduct experiments with different numbers of data for constraints, i,e., 100, 1k, 2k, 4k from each task. The comparison results are presented in Figure 2. The figure illustrates that improving the base model on the target tasks is not challenging, as nearly all methods accomplish this effortlessly. However, all baselines, including the strong continual learning baselines GEM and

Figure 4: Performance of recognizing 6 weather conditions for autonomous driving with two rounds of model development using new data. The Round 1 development targets at *overcast* and Round 2 aims to improve recognizing *foggy*. Base refers to the CLIP model finetuned on BDD100K data.

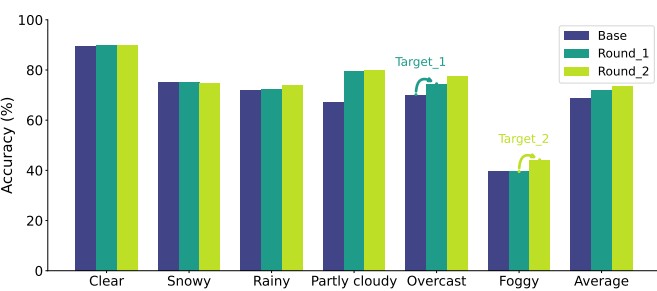

Co²L, exhibit a zero retention ratio across almost all settings, showing the insufficiency of existing methods for ensuring MDS on protected tasks. Although, when targeting *Foggy*, DER may achieve a retention ratio up to 60% with more data from protected tasks and large regularization weights, it fails to improve target performance at the same time. In contrast, our method begins to ensure developmental safety with 1k samples per protected task and even 100 samples for the target class *tunnel*. Besides, the retention ratio increases when using more data for constraints, consistent with the result obtained in Lemma 1 (Refer to Table 1 for more results). Notably, our method achieves a 100% retention ratio with 4k samples per protected task in all three settings, while improving accuracy on the target class. We also see that the target *overcast* is most difficult to improve as the base model already has 73.6% accuracy. From table 1, we further observe that higher retention ratios are accompanied by smaller gains on the target task, indicating a tradeoff between target task performance and meeting developmental safety requirements.

Figure 2 shows that, even with a tunable weight parameter $\alpha$ for protected tasks, RM fails to ensure MDS. The key difference between RM and our method is how protected tasks are handled. As shown in Eqn. (7), our approach assigns an adaptive weight $\beta[u_k^t]_+$ to each protected task, adjusting dynamically based on constraint violations—the larger the violation, the higher the weight. This mechanism is crucial for ensuring MDS with multiple protected tasks and improving target task. As shown in Figure 3 (left), these weights gradually decrease to zero, allowing the model to focus on the target task when satisfying constraints. In contrast, RM applies the same weight $\alpha$ to every protected task, which may fail due to varying task difficulty, and an overly large $\alpha$ can harm target-task performance. Further analysis in Appendix A.4 reveals that a uniform weight may preserve performance on some protected tasks but fail to ensure MDS across all, even with a large $\alpha$.

To further validate our method's effectiveness with a large number of constraints, we experiment on the Place365 dataset, compared with RM, targeting *Dressing room* class while protecting other 364 tasks. Figure 3 (right) shows that even with hundreds of protected tasks, our method is still effective in retaining their performance. In contrast, RM even with a large weight $\alpha$ not only causes performance drops in around 30 protected classes failing to ensure MDS but also fails to improve the performance of the target task.

### 6.3 Multiple Rounds of Model Development

To demonstrate the effectiveness of the proposed retention-centric framework in iterative model development process, we conduct two consecutive rounds of development on recognizing weather conditions. Specifically, we first target at *overcast* task, taking all the other five weather conditions as protected tasks, then with one selected improved model, we successively improve the model, targeting at improving the performance of the *foggy* task. As shown in Fig. 4, our method notably improves the performance of the *overcast* task in the first round while ensuring the performance of other tasks does not decrease. In the second round, it continues to enhance the performance of the *foggy* task. Simultaneously, it preserves the performance, if not boosts it, across other tasks, with only a slight decrease on the *snowy* task, showing the effectiveness of the proposed framework for maintaining the model developmental safety.

### 6.4 Ablation Studies

#### 6.4.1 Importance of the External Data

We conduct experiments on targeting *foggy* to investigate the benefits of the external data retrieved from LAION400M dataset. In detail, we vary the number of retrieved target-related image-text pairs utilized in the objective function, i.e., {0, 2k, 5k, 11k}, with 1k samples from each protected task as constraints.

Table 1: Effect of the Number of Samples for Constraints. Numbers in parentheses denote standard deviation.

| Target | Measures | Base model | 100 | 1k | 2k | 4k |
|---|---|---|---|---|---|---|
| Tunnel | DevSafety(acc) | 0.00(0.0000) | -0.0050(0.0076) | -0.0001(0.0043) | 0.0105(0.0053) | 0.0186(0.0058) |
| | Retention Ratio | 100.00% | 40.00% | 60.00% | 100.00% | 100.00% |
| | Target Acc | 0.1064(0.0000) | 0.9362(0.0699) | 0.8723(0.0233) | 0.9106(0.0159) | 0.8723(0.0233) |
| Foggy | DevSafety(acc) | 0.00(0.0000) | -0.0241(0.0082) | -0.0009(0.0044) | 0.0044(0.0033) | 0.0061(0.0047) |
| | Retention Ratio | 100.00% | 0.00% | 60.00% | 100.00% | 100.00% |
| | Target Acc | 0.3953(0.0000) | 0.5721(0.0406) | 0.4930(0.0174) | 0.4326(0.0186) | 0.4279(0.0316) |
| Overcast | DevSafety(acc) | 0.00(0.0000) | -0.0655(0.0249) | -0.0043(0.0037) | 0.0012(0.0029) | 0.0046(0.0016) |
| | Retention Ratio | 100.00% | 0.00% | 20.00% | 60.00% | 100.00% |
| | Target Acc | 0.7361(0.0000) | 0.8789(0.0464) | 0.7827(0.0225) | 0.7562(0.0167) | 0.7525(0.0366) |

Table 2: The Effect of External Image-text Pairs from LIAON400M. Numbers in parentheses denote std.

| | Ref(Base model) | 0 | 2k | 5k | 11k |
|---|---|---|---|---|---|
| Retention Ratio | 100.00% | 100.00% | 80.00% | 100.00% | 60.00% |
| Target Acc (Foggy) | 0.3953(0.0000) | 0.3674(0.0372) | 0.4047(0.0562) | 0.4186(0.0389) | 0.4930(0.0174) |

From Tab. 2, we can see that, with only 57 *foggy* samples from BDD100k dataset (i.e., 0 samples from the external data), the model does not improve the target accuracy at all. However, with more and more retrieved image-text pairs utilized to augment the dataset $\mathcal{D}$, the improvement on the targeted task appears and becomes significant, showing the advantages of incorporating the retrieved target-related image-text pairs for boosting target task accuracy. Regarding retention ratios, we don't observe a clear correlation between the amount of retrieved data and the retention ratios.

### 6.4.2 Importance of Task-dependent Heads

As introduced in Section 5.3, to reduce the total complexity of our algorithm, we propose task-dependent heads to increase the parameter $\delta$ in Assumption 3, avoiding getting trapped at a flat location where $\mathbf{w}^t$ is infeasible but $\nabla H(\mathbf{w}^t) = 0$. To verify the effectiveness of the design, we experiment on targeting the *foggy* task with varying numbers of data for constraints. The results are presented in Figure 5(a). The results show that models equipped with task-dependent heads almost consistently exhibit higher retention ratios and higher accuracy on the target task. Besides, without task-dependent heads, models may have trouble achieving a 100% retention ratio, demonstrating the effectiveness of task-dependent heads for promoting developmental safety.

To further verify the theoretical result in Lemma 2, we empirically calculate $\nabla \widehat{\mathbf{h}}(\widehat{\mathbf{w}})$ and $\nabla \mathbf{h}(\mathbf{w})$ with CLIP models. Specifically, we compute the minimal singular values of $\nabla \widehat{\mathbf{h}}(\widehat{\mathbf{w}})$ and $\nabla \mathbf{h}(\mathbf{w})$ on the base model and two trained models, with 1k samples for each protected task. The initial value of $U_k$ is set to zero so $U_k V_k^\top = \mathbf{0}$. From the results presented in Table 3, we can observe that, on the initial models, the minimal singular value of $\nabla \widehat{\mathbf{h}}(\widehat{\mathbf{w}})$ is slightly larger than that of $\nabla \mathbf{h}(\mathbf{w})$ and the gap become much significant after training, which is consistent with the theoretical result in Lemma 2 and also provides some insight on the empirical results in Figure 5(a).

### 6.4.3 Constant $\beta$ vs Increasing $\beta$

In theory, an increasing penalty parameter $\beta$ may help reduce the complexity of constrained problems as shown in Alacaoglu & Wright (2024), but in our empirical experiments, we find that using a constant $\beta$ is generally behave better than using an increasing $\beta$ . As shown in Fig. 5(b) for target task *foggy*, models with a constant $\beta$ are able to achieve 100% retention ratio with 2k or 4k sampler per constraint. On the contrary, models using a cosine increasing $\beta$ obtain both lower retention ratio and lower accuracy on the target task, compared with models with constant $\beta$. We conjecture that this is because models with an increasing $\beta$ might leave the developmental safety region too far in the initial stages as they have a relatively small penalty weight $\beta$ at this time. Given the high non-convexity and complexity of the model space, it becomes increasingly

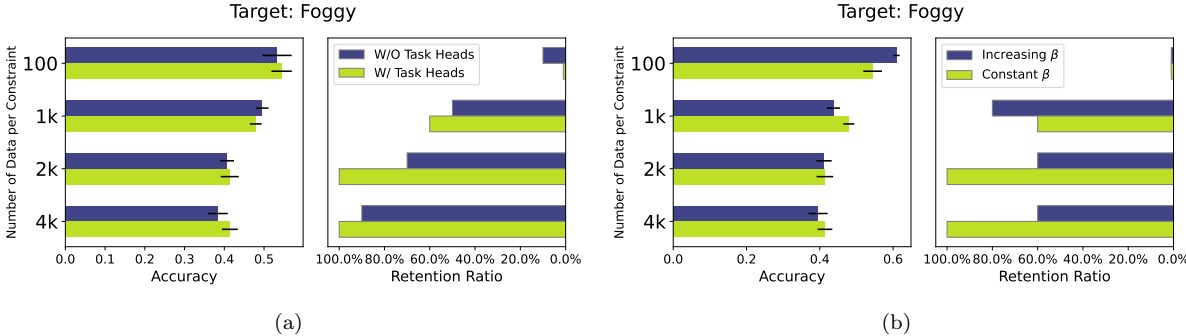

(a)  (b)

Figure 5: (a) Task-dependent heads promote developmental safety. (b) Performance Comparison between constant $\beta$ and increasing $\beta$. The results in (a) and (b) are both summarized over 10 runs with different random seeds to ensure reliability,

Table 3: Minimal Singular Values $\delta$ of $\nabla\mathbf{h}(\mathbf{w})$ and $\nabla\widehat{\mathbf{h}}(\hat{\mathbf{w}})$

|  |  | Initial Model | Final Model |
|---|---|---|---|
| w/o task-dependent heads | $\nabla\mathbf{h}(\mathbf{w})$ | 23.9183 | 15.0919 |
| w task-dependent heads | $\nabla\widehat{\mathbf{h}}(\hat{\mathbf{w}})$ | 24.1038 | 16.3397 |

challenging in the later stages to return to a feasible solution that satisfies developmental safety constraints while significantly improving target accuracy.

## 7 Discussion and Limitations

In this paper, we introduced a requirement of a learning system, namely model developmental safety, to ensure that model development not only improves target capabilities but also strictly retains existing essential ones, addressing a key oversight in ML/AI research. To ensure model developmental safety, we proposed a retention-centric framework and an efficient constrained optimization algorithm with theoretical guarantees to develop a pretrained CLIP model for acquiring new or improving existing image classification capabilities. Experiments on driving and scene recognition datasets validate its effectiveness, showing its practical value.

**Extention to other domains.** While the proposed retention-centric optimization framework is in principle generally applicable across a range of models and domains, the experimental validation in this work focuses on CLIP models in image classification. One may consider extending the framework to fine-tune LLMs, improving medical diagnosis systems, or enhancing autonomous driving systems. For example, for finetuning LLMs to improve math capability, $F(\mathbf{w}, \mathcal{D})$ in Eqn. (2) can be instantiated as an SFT (Supervised Finetuning) loss on math task data with $\mathcal{L}_k(\mathbf{w}, \mathcal{D}_k)$ being SFT loss on other tasks data, like harmlessness task data, then the optimization algorithm 1 is still applicable. We hope our work can inspire researchers in cost-sensitive domains for more exploration with other models and tasks.

**Desirability and feasibility.** As strictly preserving the model's essential capabilities is very important in cost-sensitive applications(e.g., autonomous driving, medical diagnosis), this work focuses on ensuring model developmental safety (i.e., zero forgetting) on essential established capabilities in high-stakes domains via MDS constraints. Nevertheless, in certain scenarios, strict preservation is not necessary, and a small performance drop can be tolerated in exchange for higher target performance. In such cases, classical continual learning methods and metrics (e.g., average accuracy) may be more appropriate. Moreover, since satisfying the MDS constraints while improving target performance is challenging, sufficient model capacity may be required to guarantee zero forgetting, and task-dependent heads can further help mitigate this issue with a small additional computation cost.

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

# A  More Experimental Details and Results

## A.1  Experimental Details.

### A.1.1  Dataset.

We choose the large-scale diverse driving image dataset, namely BDD100K (Seita, 2018), for part of our experiments. This dataset involves six weather conditions, i.e., *clear, overcast, snowy, rainy, partly cloudy, foggy*, and six scene types, i.e., *highway, residential area, city street, parking lot, gas station, tunnel*. Since the labels of the official testing dataset are not released, we utilize the official validation set for testing and partition the training dataset into training and validation sets using an 80%/20% ratio.

Moreover, we experiment on a scene recognition dataset, Places365 (Zhou et al., 2017), to verify the effectiveness of the proposed method in handling a large number of constraints. We utilize the standard version of the dataset (i.e., Places365-Standard), with 1.8 million training and 36500 validation images from 365 scene classes. The number of examples for each class varies between 3,068 and 5,000 in the training set. We merge the training dataset and validation dataset and randomly split the whole set into training set, validation set and test set with an 60%/20%/20% ratio.

Table 4: Datasets Statistics for BDD100K Dataset

| Weather | Training | Validation | Testing |
|---|---|---|---|
| Clear | 29865 | 7479 | 5346 |
| Snowy | 4445 | 1104 | 769 |
| Rainy | 4119 | 951 | 738 |
| Partly Cloudy | 3992 | 959 | 738 |
| Overcast | 7043 | 1727 | 1239 |
| Foggy | 57 | 43 | 43 |

| Scene | Training | Validation | Testing |
|---|---|---|---|
| Hightway | 13952 | 3427 | 2499 |
| Residential area | 6458 | 1616 | 1253 |
| City street | 34862 | 8654 | 6112 |
| Parking lot | 297 | 80 | 49 |
| Tunnel | 62 | 47 | 47 |

### A.1.2  Experimental Settings.

We employ the CLIP ViT-B/16 (Radford et al., 2021b) as the backbone network in all our experiments.

The image-text pairs for the objective function are from the training set of BDD100K and the external LAION400M (Schuhmann et al., 2021) dataset. Specifically, for each target class, we use a query prompt (detailed in Appendix A.2) to search for target-related image-text pairs in LAION400M to augment the target data set $\mathcal{D}$. Additionally, we randomly sample a set of image-text pairs from LAION400M that is 10 times larger than target-related pairs as negative data for contrasting. The data of protected tasks used for constructing constraints are sampled from the BDD100K training set with varying sizes. Statistics for BDD100K in our experiments are shown in Appendix Table 4. The text templates used for BDD100K dataset are "*the weather is [Weather]*" and "*the scene is a [Scene]*".

For Places365 dataset, we directly utilize the pretrained CLIP model released by Radford et al. (2021b) as the base mode. Then we conduct continual development to improve the performance of *dressing room* class, which has the fewest samples in the dataset, and consider all the other 364 classes as protected tasks. Similar to the setting for BDD100K dataset, we also use a query prompt (detailed in Appendix A.2) to search for target-related image-text pairs in LAION400M to augment the target set $\mathcal{D}$. The data of protected tasks used for constructing constraints are sampled from the Places365 training set. The text templates used for Places365 dataset are "*the scene is a(n) [Scene]*".

### A.1.3 Hyperparameter tuning.

For all methods in our experiments, we tune the learning rate in {1e-5, 1e-6} with Cosine scheduler and AdamW optimizer, using a weight decay of 0.1.

For BDD100K dataset, we set temperature $\tau_0$ as 0.05. We run each method for a total of 40 epochs with a batch size of 256 and 600 iterations per epoch, except for GEM whose total epochs are tuned in {1,2,5} with a batch size of 64 since more iterations lead to exacerbated catastrophic forgetting problems as shown in their paper. For our method, we tune $\beta$ in {100, 200, 400}, $\gamma_2$ in {0.4, 0.6, 0.8} and set $r = 32, |\mathcal{B}_c| = m, |\mathcal{B}_k| = 10$. We set $\gamma_1$ to 0.8, $\tau$ to 0.05 in FLYP, WCCL, RM, and our method. For WCCL, we vary the weight parameter $\alpha$ in {0.5,0.9,0.99}. For GEM, we tune their small constant $\gamma$ in {0.5, 1.0}. For Co$^2$L, we tune their $\tau$ in {0.05, 0.1}, $\kappa$ in {0.1, 0.2}, $\kappa^*$ in {0.01, 0.1}, $\lambda$ in {0.1, 1, 10}. For DER, we tune their $\alpha$ in {0.1, 1, 10} and $\beta$ in {0, 1}. For RM, we tune regularization weight $\alpha$ in {0.1, 1, 10}. In hyper-parameters selection for all methods, we prioritize larger retention ratio first and consider larger $\Delta$Acc (Target) if there is a tie in terms of retention ratio, as we look for models that maximize $\Delta$Acc (Target) while satisfying DevSafety $\geq 0$.

For Places365 dataset, the temperature $\tau_0$ is set as 0.01. Since there are as many as 364 constraints, we set $|\mathcal{B}_c| = 240, |\mathcal{B}_k| = 2$. We tune $\beta$ in {600, 1000, 4000} for our method and regularization weight $\alpha$ in {1, 10, 100, 1000, 10000} for RM. We run each method five times for a total of 40 epochs with 1400 iterations per epoch, with a batch size of 64.

### A.1.4 Details about Baselines

**FLYP.** In the original FLYP paper (Goyal et al., 2023), the author presents extensive experiments demonstrating the superiority of employing the contrastive loss used during pre-training instead of the typical cross-entropy for finetuning image-text models for zero-shot vision classification. As the local contrastive loss, defined over the mini-batch samples, utilized in their paper requires a very large mini-batch size to converge, we follow Yuan et al. (2022) to employ a global constrastive loss (GCL) as indicated in Eqn. 9 to address this issue:

$$\min_{\mathbf{w}} \quad \frac{1}{n_{all}} \sum_{(\mathbf{x}_i, \mathbf{t}_i) \in \mathcal{D}_{all}} L_{\text{ctr}}(\mathbf{w}; \mathbf{x}_i, \mathbf{t}_i, \mathcal{D}_{all}, \mathcal{D}_{all}) \tag{9}$$

where $\mathcal{D}_{all} = \mathcal{D} \cup \mathcal{D}_- \cup \mathcal{D}_1 \cup \cdots \cup \mathcal{D}_m, n_{all} = n_o + 10 * n_o + n_1 + \cdots + n_m$, $\mathcal{D}_-$ is the negative data collected form LAION400M as discussed in AppendixA.2. All available data, including those used in our objective and constraints, are utilized for fine-tuning. The simple text prompts for the labeled BDD100k dataset are the same as those used for our method, i.e., "*the weather is [Weather]*" and "*the scene is a [Scene]*".

**WCCL.** Weighted Combination of Contrastive Losses(WCCL) is a straightforward baseline that utilizes a weight to combine GCL losses on protected tasks and the target task to balance protected tasks and the target task and achieve model developmental safety. Specifically, the objective can be formulated as:

$$\min_{\mathbf{w}} \quad \alpha \Big( \frac{1}{m} \sum_{k=1}^{m} \frac{1}{n_k} \sum_{(\mathbf{x}_i, \mathbf{t}_i) \in \mathcal{D}_k} L_{\text{ctr}}(\mathbf{w}; \mathbf{x}_i, \mathbf{t}_i, \mathcal{T}_{ik}^-, \mathcal{I}_{ik}^-) \Big)$$
$$+ (1 - \alpha) \Big( \frac{1}{n_o} \sum_{(\mathbf{x}_i, \mathbf{t}_i) \in \mathcal{D}_o} L_{\text{ctr}}(\mathbf{w}; \mathbf{x}_i, \mathbf{t}_i, \mathcal{T}_{io}^-, \mathcal{I}_{io}^-) \Big) \tag{10}$$

where $\mathcal{T}_{ik}^- = \{\mathbf{t}_j : (\mathbf{x}_j, \mathbf{t}_j) \in \mathcal{D}_{all} \backslash \mathcal{D}_k\} \cup \{\mathbf{t}_i\}, \mathcal{I}_{ik}^- = \{\mathbf{x}_j : (\mathbf{x}_j, \mathbf{t}_j) \in \mathcal{D}_{all} \backslash \mathcal{D}_k\} \cup \{\mathbf{x}_i\}$, $\mathcal{D}_{all} \backslash \mathcal{D}_k$ denotes all training samples excluding samples from $\mathcal{D}_k$. Similarly, $\mathcal{T}_{io}^- = \{\mathbf{t}_j : (\mathbf{x}_j, \mathbf{t}_j) \in \mathcal{D}_{all} \backslash \mathcal{D}_o\} \cup \{\mathbf{t}_i\}, \mathcal{I}_{io}^- = \{\mathbf{x}_j : (\mathbf{x}_j, \mathbf{t}_j) \in \mathcal{D}_{all} \backslash \mathcal{D}_o\} \cup \{\mathbf{x}_i\}$. Consistent with other methods, the simple text prompts for this baseline are also "*the weather is [Weather]*" and "*the scene is a [Scene]*".

**GEM.** GEM (Lopez-Paz & Ranzato, 2017b) is a strong continual learning baseline which motivated by a similar idea, utilizing data of previous tasks for constraints. But it doesn't solve the constrained optimization problem directly but project gradients to reduce the increase in the loss of previous tasks. For GEM, we start from pretrained image encoder of the same CLIP model and initialize the linear classification heads $W \in \mathbb{R}^{d \times (m+1)}$ with the representations outputted by the text encoder with input "*the weather is [Weather]*" or "*the scene is a [Scene]*". For each task $k$, cross entropy loss is employed $\mathcal{L}_k(\mathbf{w}, W, \mathcal{D}_k) = \frac{1}{n_k} \sum_{(\mathbf{x}_i, y_i) \sim \mathcal{D}_k} - \log \frac{\exp(W_k^\top E_1(\mathbf{w}, \mathbf{x}_i)/\tau_0)}{\sum_{\ell=1}^{m+1} \exp(W_l^\top E_1(\mathbf{w}, \mathbf{x}_i)/\tau_0)}$, where $\tau_0 > 0$ is a temperature parameter, $W_k, W_l$ denoted the $k_{th}, l_{th}$ column vector of $W$ respectively, and $E_1(\mathbf{w}, \mathbf{x}_i)$ is the normalized image

representation of $\mathbf{x}_i$. For consistency, $\tau_0$ is fixed to 0.05 as the one used in our method. In each iteration, 10 examples are drawn from each protected task to calculate the corresponding loss gradient vector for each task.

**RM.** In continual learning literature, adding explicit regularization terms is a widely used approach to balance old and new tasks, exploiting a frozen copy of previously-learned model to help prevent catastrophic forgetting (Rebuffi et al., 2017; Castro et al., 2018). Similarly, the Regularization Method(RM) baseline incorporates the constraints from Eqn. (3) as a regularization term, adding it to the objective function with an associated regularization weight:

$$\min_{\mathbf{w}} \frac{1}{n_o} \sum_{(\mathbf{x}_i, \mathbf{t}_i) \in \mathcal{D}_o} L_{\mathrm{ctr}}(\mathbf{w}; \mathbf{x}_i, \mathbf{t}_i, \mathcal{T}_{io}^-, \mathcal{I}_{io}^-) + \alpha \left( \frac{1}{m} \sum_{k=1}^{m} \frac{1}{n_k} \sum_{(\mathbf{x}, y) \in \mathcal{D}_k} \ell_{ce}(\mathbf{w}, \mathbf{x}, y) \right) \tag{11}$$

## A.2 Retrieving external data from LIAON400M

To further boost the performance of the CLIP model on the target task, we utilize external data to augment the target data set. Specifically, for each target task, we retrieve task-related image-text pairs from Laion400M (Schuhmann et al., 2021), by going through the dataset and retrieving the image-text pairs with text containing the specific target task names, e.g., 'foggy', 'overcast', 'tunnel', 'dressing room'. Similar approaches have been used in (Liang et al., 2024; Mitchell et al., 2018; Chen et al., 2013), where Liang et al. (2024) used this approach to improve the detection of rare or unseen categories in object detection for autonomous driving systems. However, their study is different from ours in the sense that they do not provide guarantee of the model developmental safety.

Moreover, we refine the retrieved datasets. Let's take the task 'tunnel' as an example. For task 'tunnel', the retrieved data contained excessive noise, including numerous image-text pairs unrelated to tunnels, but contained 'tunnel' in the text. Therefore, we employed the GPT-4o API to filter the retrieved data with prompt "*Determine whether the following caption mentions a tunnel or related context. First provide reasoning for your answer, and then respond with 'True' if it mentions a tunnel, or 'False' if it does not.*", thereby decreasing the noise of our retrieved data. The statistics of obtained task-related image-text pairs are presented in the Table 5. Additionally, for each target class, we randomly sample a set of image-text pairs from LAION400M that is 10 times larger than the positive set as negative data for contrasting.

Table 5: Statistics of Data Collected from LIAON400M

| Task | Foggy | Overcast | Tunnel | Dressing room |
|---|---|---|---|---|
| Size | 11415 | 4134 | 23484 | 6786 |

## A.3 Visualization of Models' Learning Curves

Along with the learning trajectory in the main paper, we present the training and validation curves in Fig. 6 to further illustrate the learning process of the algorithm. From the figure, we can see that the DevSafety(acc) fluctuates along the MDS line while $\Delta Acc(Target)$ continues to increase, showing the model is striving to improve its performance on the target task while satisfying the MDS requirements.

## A.4 Deficiency of Weighting Methods

As observed in Figure 2, the naive weighting approach RM fail to achieve model developmental safety, even though they tradeoff the performance on the target task and protected tasks with weight parameter $\alpha$. To have a close look at why this happens, we show the detailed performance RM when targeting *foggy* with 4k samples for each protected task in Table 6. We find that, with a uniform weight for all the protected tasks, the method might preserve previous performance on some of the protected tasks but fail to achieve MDS for all the protected tasks, even with a very high $\alpha$. Moreover, with the weight $\alpha$ getting larger, the performance on the target task drops dramatically although the decline in protected tasks goes smaller, e.g., *Clear* tasks for RM. In contrast, our proposed method is able to strictly retain all the protected tasks' performance and improve the target task, as the mechanism of our algorithm is very different from using the uniform weight. In our method, weights for protected tasks depend on the constraint violation of those tasks, i.e.,

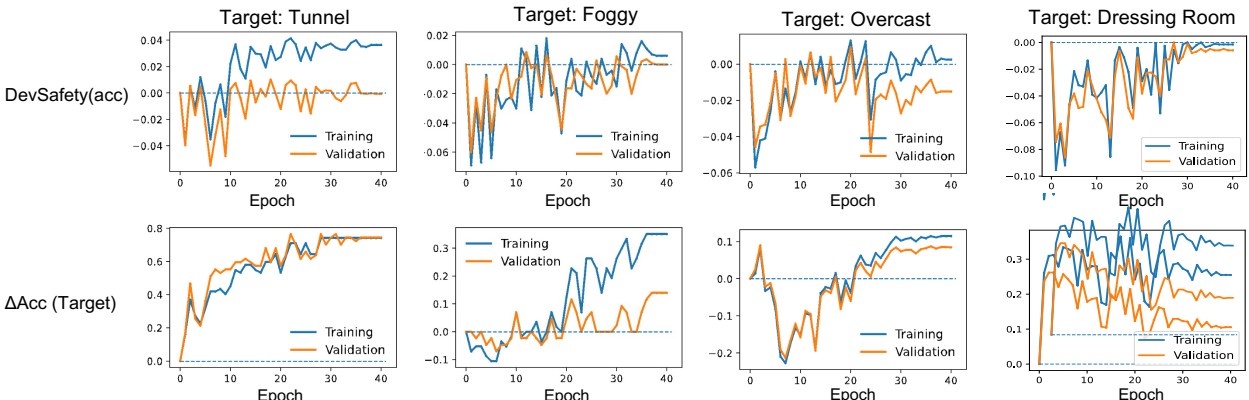

Figure 6: Models' Training and Validation Curves

Table 6: Detailed performance comparison between our method and baseline RM on targeting *Foggy* with 4k samples for each protected task. Bold numbers highlight the performance decrease over the base model.

| | Clear | | Protected Tasks | | | Target Task | Average |
| | | Overcast | Snowy | Rainy | Partly cloudy | Foggy | |
|---|---|---|---|---|---|---|---|
| Base | 0.8938 | 0.7014 | 0.7503 | 0.7195 | 0.6734 | 0.3953 | 0.6889 |
| Ours | +0.0115(0.0054) | +0.0831(0.0228) | +0.0120(0.0079) | +0.0230(0.0081) | +0.1047(0.0168) | 0.0326(0.0316) | +0.0430(0.0027) |
| RM $\alpha = 0.1$ | **-0.0189(0.0039)** | +0.0667(0.0392) | +0.0328(0.0113) | +0.0081(0.0074) | +0.1253(0.0227) | +0.0559(0.0617) | +0.0450(0.0071) |
| RM $\alpha = 1$ | **-0.0129(0.0055)** | +0.0910(0.0102) | +0.0666(0.0139) | +0.0217(0.0215) | +0.1168(0.0112) | **-0.0604(0.0634)** | +0.0372(0.0114) |
| RM $\alpha = 10$ | **-0.0106(0.0085)** | +0.1131(0.0068) | +0.0656(0.0302) | +0.0163(0.0182) | +0.0830(0.0201) | **-0.1674(0.0174)** | +0.0167(0.0050) |

the larger the violation, the larger the weight. As shown in Figure 3 (left), the weight for each protected task is adaptively adjusted during learning and once one protected task constraint is satisfied, it will not be penalized (weight becomes zero). This mechanism plays a big role in enabling the model to find feasible solutions to ensure zero-forgetting on all the protected tasks.

To further demonstrate the deficiency of the weighting method, we compare RM with our method on the Place365 dataset, targeting *Dressing room* class and protecting the other 364 tasks in Figure 7. With $\alpha = 1, 10, 100, 1000, 10000$, RM causes performance drops in 50, 35, 33, 32, and 35 classes, respectively. Although larger weights reduce the number of classes where performance drops, RM still cannot ensure MDS for all protected tasks. In contrast, we can see that even with hundreds of protected tasks, our method is still effective in preserving their performance whiling improving the target task.

### A.5    Performance Comparison Using ∆Acc (Target) and DevSafety (acc)

In this part, we present additional comparisons with baselines based on target performance change (∆Acc (Target)) and DevSafety (acc). DevSafety(acc) directly measures the largest decrease across all protected tasks. The results are summarized in Figure 8 and detailed numbers are included in Table 7, Table 8, Table 9. We can see that baselines usually lead to 1-14 percent decrease when targeting Tunnel, 0-13 percent decrease when targeting Foggy, 1-30 percent decrease when targeting Overcast. While none of the baselines may achieve both positive DevSafety(acc) and ∆Acc simultaneously, our method is the only one that attains both, demonstrating the superiority of the proposed method. Notably, when targeting Tunnel, our method achieves comparable target-task improvement to baselines while ensuring model developmental safety, highlighting the effectiveness of our method.

### A.6    Experiments on Medical Domain

To validate the applicability of our method to other models, we conduct additional experiments for classification on CheXpert dataset (Irvin et al., 2019) for detecting chest and lung diseases with traditional CNNs. Specifically, we first train a Desnet121 model on detecting four diseases, i.e., Cardiomegaly, Edema, Atelectasis, Pleural Effusion, and then continually train the model to detect a new disease (Consolidation) while preserving

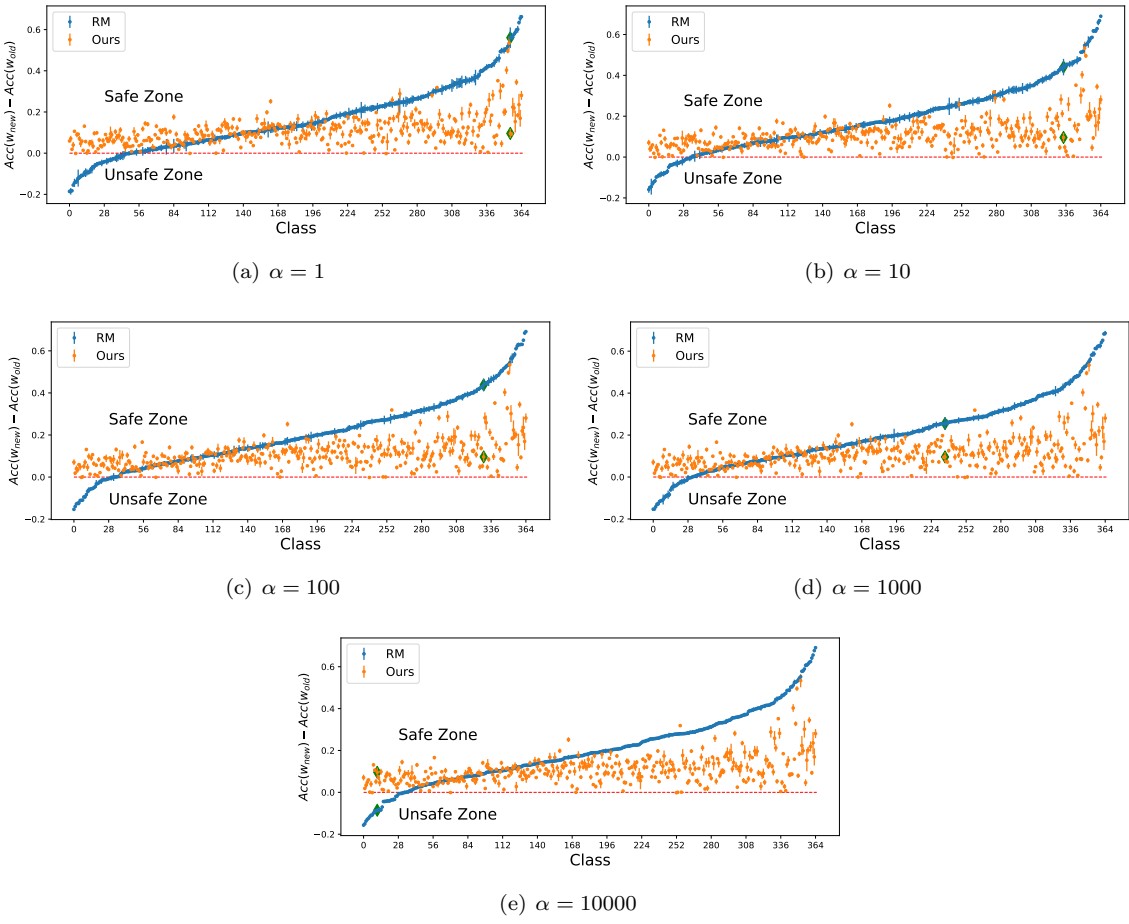

(a) $\alpha = 1$

(b) $\alpha = 10$

(c) $\alpha = 100$

(d) $\alpha = 1000$

(e) $\alpha = 10000$

Figure 7: Performance comparison between our method and baseline RM with different regularization weight $\alpha$ when targeting *Dresssing Room* on Places365 Dataset, with 2k samples per constraint. Red line denotes base model's performance, green diamonds denote the target class.

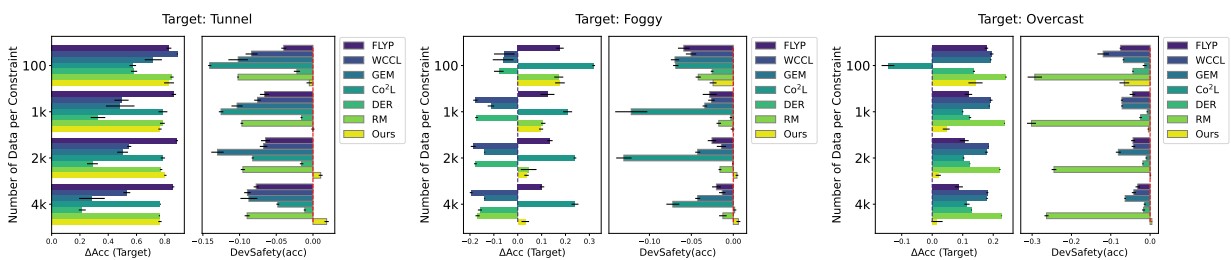

Figure 8: Performance comparison with baselines based on $\Delta$Acc (Target) and DevSafety (acc). Detailed numbers are presented in Table 7, Table 8, Table 9.

the performance on previously learned four diseases, with 4k samples for each protected task. For our method, we tune $\beta$ in 100, 400, 1000. For the baseline method RM, we tune $\alpha$ in 1, 4, 10, 40. The AUC score results are summarized in Table 10. We can see that our method is still effective for learning traditional CNN models while retaining protected tasks' performance, which cannot be achieved by baseline RM, demonstrating the effectiveness of our method when applied to other models.

Table 7: Detailed Performance Comparison on Targeting Tunnel

| Method | Measures | 100 | 1k | 2k | 4k |
|---|---|---|---|---|---|
| Base | Target Tunnel | 0.1064(0.0000) | 0.1064(0.0000) | 0.1064(0.0000) | 0.1064(0.0000) |
| FLYP | Retention Ratio | 0.00% | 0.00% | 0.00% | 0.00% |
| | DevSafety(acc) | -0.0398(0.0067) | -0.0660(0.0126) | -0.0647(0.0123) | -0.0774(0.0069) |
| | $\Delta$Acc (Tunnel) | 0.8296(0.0330) | 0.8636(0.0318) | 0.8846(0.0170) | 0.8586(0.0170) |
| WCCL | Retention Ratio | 0.00% | 0.00% | 0.00% | 0.00% |
| | DevSafety(acc) | -0.0836(0.0164) | -0.0756(0.0090) | -0.0673(0.0103) | -0.0893(0.0089) |
| | $\Delta$Acc (Tunnel) | 0.8886(0.0085) | 0.4936(0.1002) | 0.5486(0.0282) | 0.5316(0.0485) |
| GEM | Retention Ratio | 0.00% | 0.00% | 0.00% | 0.00% |
| | DevSafety(acc) | -0.1019(0.0267) | -0.1034(0.0153) | -0.1301(0.0169) | -0.0873(0.0231) |
| | $\Delta$Acc (Tunnel) | 0.7186(0.1214) | 0.4846(0.2020) | 0.5016(0.0768) | 0.2846(0.1819) |
| Co$^2$L | Retention Ratio | 0.00% | 0.00% | 0.00% | 0.00% |
| | DevSafety(acc) | -0.1407(0.0043) | -0.1252(0.0061) | -0.0821(0.0029) | -0.0479(0.0039) |
| | $\Delta$Acc (Tunnel) | 0.5736(0.0460) | 0.7866(0.0626) | 0.7866(0.0301) | 0.7656(0.0000) |
| DER | Retention Ratio | 0.00% | 0.00% | 0.00% | 0.00% |
| | DevSafety(acc) | -0.0219(0.0083) | -0.0156(0.0026) | -0.0155(0.0048) | -0.0111(0.0029) |
| | $\Delta$Acc (Tunnel) | 0.5826(0.0438) | 0.3276(0.1030) | 0.2886(0.0768) | 0.2166(0.0493) |
| RM | Retention Ratio | 0.00% | 0.00% | 0.00% | 0.00% |
| | DevSafety(acc) | -0.1021(0.0022) | -0.0969(0.0036) | -0.0955(0.0057) | -0.0897(0.0068) |
| | $\Delta$Acc (Tunnel) | 0.8506(0.0233) | 0.7826(0.0340) | 0.7736(0.0170) | 0.7616(0.0085) |
| Ours | Retention Ratio | 40.00% | 60.00% | 100.00% | 100.00% |
| | DevSafety(acc) | -0.0050(0.0076) | -0.0001(0.0043) | **0.0105(0.0053)** | **0.0186(0.0058)** |
| | $\Delta$Acc (Tunnel) | 0.8296(0.0699) | 0.7656(0.0233) | **0.8036(0.0159)** | **0.7656(0.0233)** |

## B  More Related Work

**Continual learning.** This work is closely related to Continual learning (CL), also known as lifelong learning, yet it exhibits nuanced differences. Continual learning usually refers to learning a sequence of tasks one by one and accumulating knowledge like human instead of substituting knowledge (Wang et al., 2024; Qu et al., 2021). The core issue in CL is known as catastrophic forgetting (McCloskey & Cohen, 1989), i.e., the learning of the later tasks may **significantly** degrade the performance of the model for the earlier tasks. There is a vast literature of CL of deep neural networks (DNNs) (Aljundi et al., 2018; Lopez-Paz & Ranzato, 2017a; Farajtabar et al., 2019; Lee et al., 2017; Guo et al., 2020; Parisi et al., 2018). Different approaches have been investigated to mitigate catastrophic forgetting, including regularization-based approaches (Castro et al., 2018; Kirkpatrick et al., 2017; Zenke et al., 2017; Li & Hoiem, 2016), expansion-based approaches (Zhou et al., 2022; Yan et al., 2021; Li et al., 2019; Rusu et al., 2016), and memory-based approaches (Wang et al., 2024; Buzzega et al., 2020; Cha et al., 2021; Guo et al., 2020; Lopez-Paz & Ranzato, 2017a; Chaudhry et al., 2019). The framework proposed in this work is similar to memory-based approaches in the sense that both use examples of existing tasks to regulate learning. However, the key difference is that most existing continual learning focuses on the trade-off between learning plasticity and memory stability and aims to find a proper balance between performance on previous tasks and new tasks (Wang et al., 2024). Hence, they do not provide a guarantee for MDS. A recent work (Peng et al., 2023) has proposed an ideal continual learner that never forgets by assuming that all tasks share the same optimal solution. However, it is not implementable for deep learning problems. Besides, existing continual learning studies usually highlight resource efficiency when accumulating knowledge by reducing the number of samples of previous tasks. In contrast, this work tends to utilize more examples to construct developmental safety constraints for protected tasks to facilitate MDS.

**AI Safety.** Our notion of model developmental safety should not be confused with AI safety. The latter is a field concerned with mitigating risks associated with AI, whose surge in attention stems from the growing

Table 8: Detailed Performance Comparison on Targeting Foggy

| Method | Measures | 100 | 1k | 2k | 4k |
|---|---|---|---|---|---|
| Base | Target Foggy | 0.3953(0.0000) | 0.3953(0.0000) | 0.3953(0.0000) | 0.3953(0.0000) |
| FLYP | Retention Ratio | 0.00% | 20.00% | 0.00% | 0.00% |
| | DevSafety(acc) | -0.0590(0.0140) | -0.0281(0.0167) | -0.0254(0.0101) | -0.0201(0.0105) |
| | $\Delta$Acc (Foggy) | 0.1767(0.0315) | 0.1247(0.0581) | 0.1347(0.0228) | 0.1017(0.0186) |
| WCCL | Retention Ratio | 0.00% | 0.00% | 20.00% | 0.00% |
| | DevSafety(acc) | -0.0504(0.0123) | -0.0259(0.0080) | -0.0141(0.0111) | -0.0132(0.0076) |
| | $\Delta$Acc (Foggy) | -0.0563(0.0865) | -0.1773(0.0186) | -0.1863(0.0208) | -0.1953(0.0114) |
| GEM | Retention Ratio | 0.00% | 0.00% | 0.00% | 0.00% |
| | DevSafety(acc) | -0.0695(0.0099) | -0.0339(0.0053) | -0.0424(0.0060) | -0.0424(0.0060) |
| | $\Delta$Acc (Foggy) | -0.0613(0.0865) | -0.1123(0.0271) | -0.1403(0.0000) | -0.1403(0.0000) |
| $Co^2L$ | Retention Ratio | 0.00% | 0.00% | 0.00% | 0.00% |
| | DevSafety(acc) | -0.0686(0.0064) | -0.1217(0.0383) | -0.1305(0.0183) | -0.0721(0.0154) |
| | $\Delta$Acc (Foggy) | 0.3177(0.0109) | 0.2087(0.0380) | 0.2397(0.0110) | 0.2397(0.0290) |
| DER | Retention Ratio | 0.00% | 40.00% | 40.00% | 60.00% |
| | DevSafety(acc) | -0.0252(0.0033) | -0.0029(0.0027) | -0.0010(0.0010) | 0.0017(0.0038) |
| | $\Delta$Acc (Foggy) | -0.0793(0.0405) | -0.1723(0.0114) | -0.1773(0.0114) | -0.1583(0.0174) |
| RM | Retention Ratio | 0.00% | 0.00% | 0.00% | 20.00% |
| | DevSafety(acc) | -0.0418(0.0062) | -0.0173(0.0054) | -0.0159(0.0034) | -0.0124(0.0091) |
| | $\Delta$Acc (Foggy) | 0.1717(0.0378) | 0.1067(0.0186) | 0.0457(0.0658) | -0.1683(0.0174) |
| Ours | Retention Ratio | 0.00% | 60.00% | 100.00% | 100.00% |
| | DevSafety(acc) | -0.0241(0.0082) | -0.0009(0.0044) | **0.0044(0.0033)** | **0.0061(0.0047)** |
| | $\Delta$Acc (Foggy) | 0.1767(0.0406) | 0.0977(0.0174) | **0.0367(0.0186)** | **0.0317(0.0316)** |

capabilities of AI systems, particularly large foundation models (Kojima et al., 2022; Wei et al., 2022; Bubeck et al., 2023; Radford et al., 2021b). As these models become more adept at complex tasks, concerns around potential misuse, bias, and unintended consequences rise proportionally. Amodei et al. (2016) presents several practical research problems related to AI safety, including avoiding side effects, avoiding reward hacking, scalable oversight, safe exploration, and robustness to distributional shift. More recently, Wang et al. (2023) elaborate on eight different perspectives to evaluate the trustworthiness of LLMs, including toxicity, stereotype bias, adversarial robustness, out-of-distribution robustness, robustness on adversarial demonstrations, privacy, machine ethics, and fairness. These AI safety issues arise in the usage of AI models, and they are distinctive from model developmental safety studied in this work, which arises in the development of AI models. Note that the term "safety" in model developmental safety is to underline that it is important and must be enforced in practice. Therefore, this work provides another dimension for consideration in AI safety, i.e., retention of safety. Any safety features of an AI system that have been acquired and validated should be retained safely in continuous development.

**Constrained Learning.** Constrained learning has attracted significant attention in the literature. Traditional works for constrained optimization include three primary categories: 1) primal methods which do not involve the Lagrange multipliers, e.g., cooperative subgradient methods (Lan & Zhou, 2016; Polyak & Tret'yakov, 1973) and level-set methods (Aravkin et al., 2019; Lin et al., 2018a;b); 2) primal-dual methods which reformulate constrained optimization problems as saddle point problems (Hamedani & Aybat, 2021; Nemirovski, 2004); 3) penalty-based approaches which incorporate constraints by adding a penalty term to the objective function (Xu, 2021; Lan & Monteiro, 2013; 2016). However, most of these works are limited to convex objectives or convex constraints. In recent years, due to its increasing importance in modern machine learning problems, such as in applications concerned with fairness (Cotter et al., 2019; Agarwal et al., 2018), robustness (Robey et al., 2021; Madry et al., 2017), and safety (Paternain et al., 2019b;a) problems, the research interest has been

Table 9: Detailed Performance Comparison on Targeting Overcast

| Method | Measures | 100 | 1k | 2k | 4k |
|---|---|---|---|---|---|
| Base | Target Overcast | 0.7361(0.0000) | 0.7361(0.0000) | 0.7361(0.0000) | 0.7361(0.0000) |
| FLYP | Retention Ratio | 0.00% | 0.00% | 0.00% | 0.00% |
| | DevSafety(acc) | -0.0749(0.0049) | -0.0449(0.0140) | -0.0434(0.0095) | -0.0314(0.0113) |
| | $\Delta$Acc (Overcast) | 0.1779(0.0111) | 0.1189(0.0241) | 0.1049(0.0294) | 0.0879(0.0255) |
| WCCL | Retention Ratio | 0.00% | 0.00% | 0.00% | 0.00% |
| | DevSafety(acc) | -0.1192(0.0294) | -0.0716(0.0053) | -0.0424(0.0091) | -0.0414(0.0102) |
| | $\Delta$Acc (Overcast) | 0.1949(0.0112) | 0.1929(0.0092) | 0.1839(0.0022) | 0.1809(0.0064) |
| GEM | Retention Ratio | 0.00% | 0.00% | 0.00% | 0.00% |
| | DevSafety(acc) | -0.0677(0.0042) | -0.0711(0.0050) | -0.0807(0.0128) | -0.0634(0.0042) |
| | $\Delta$Acc (Overcast) | 0.1919(0.0051) | 0.1869(0.0037) | 0.1779(0.0088) | 0.1799(0.0049) |
| $Co^2L$ | Retention Ratio | 0.00% | 0.00% | 0.00% | 0.00% |
| | DevSafety(acc) | -0.0138(0.0099) | -0.0072(0.0032) | -0.0095(0.0043) | -0.0137(0.0052) |
| | $\Delta$Acc (Overcast) | -0.1451(0.0417) | 0.0999(0.0049) | 0.1029(0.0055) | 0.1139(0.0172) |
| DER | Retention Ratio | 0.00% | 0.00% | 0.00% | 0.00% |
| | DevSafety(acc) | -0.0435(0.0037) | -0.0241(0.0064) | -0.0182(0.0032) | -0.0166(0.0077) |
| | $\Delta$Acc (Overcast) | 0.1369(0.0066) | 0.1239(0.0075) | 0.1239(0.0023) | 0.1289(0.0027) |
| RM | Retention Ratio | 0.00% | 0.00% | 0.00% | 0.00% |
| | DevSafety(acc) | -0.2932(0.0365) | -0.3016(0.0228) | -0.2444(0.0120) | -0.2634(0.0105) |
| | $\Delta$Acc (Overcast) | 0.2419(0.0050) | 0.2369(0.0028) | 0.2219(0.0041) | 0.2279(0.0023 |
| Ours | Retention Ratio | 0.00% | 20.00% | 60.00% | 100.00% |
| | DevSafety(acc) | -0.0655(0.0249) | -0.0043(0.0037) | **0.0012(0.0029)** | **0.0046(0.0016)** |
| | $\Delta$Acc (Overcast) | 0.1419(0.0464) | 0.0459(0.0225) | **0.0199(0.0167)** | **0.0159(0.0366)** |

Table 10: Performance on the medical domain with traditional CNNs

| Method | Measures | Performance |
|---|---|---|
| Base | Retention Ratio//DevSafety | 100%//0.00(0.0000) |
| | Target Consolidation (AUC) | 0.500 |
| RM | Retention Ratio//DevSafety | 0.00%//-0.0031(0.0032) |
| | Target Consolidation (AUC) | 0.8846(0.0014) |
| Ours | Retention Ratio//DevSafety | 100.00%//0.0009(0.0007) |
| | Target Consolidation (AUC) | 0.8852(0.0034) |

directed to developing efficient algorithms for non-convex optimization (non-convex objective and non-convex constraint) (Boob et al., 2023; Facchinei et al., 2021; Ma et al., 2020; Li et al., 2024; Chamon et al., 2022; Alacaoglu & Wright, 2024). Among these, Chamon et al. (2022) studies how to solve constrained learning learning with expected non-convex loss and expected non-convex constraints by using empirical data to ensure the PAC learnability, and proposed a primal-dual algorithm to solve constrained optimization problems in the empirical dual domain. However, their algorithm requires solving the primal problem up to a certain accuracy, which is theoretically not feasible for general non-convex problems. Boob et al. (2023) introduces a new proximal point method that transforms a non-convex problem into a sequence of convex problems by adding quadratic terms to both the objective and constraints. For solving non-convex optimization problems with equality constraints, Alacaoglu & Wright (2024) propose single-loop quadratic penalty and augmented Lagrangian algorithms with variance reduction techniques to improve the complexity. Nevertheless, none of

these algorithms can be directly applied to our large-scale deep learning problem (3), due to either prohibitive running cost or failure to handle biased stochastic gradients caused by compositional structure.

## C  Proofs

### C.1  Proof of Lemma 1

**Proof** *Consider task $k$. Recall that $\mathcal{D}_k$ contains $n_k$ data points. According to Theorem 3.2 in Boucheron et al. (2005), we have with probability at least $1 - \delta/m$, for all $\mathbf{w}$,*

$$|\mathcal{L}_k(\mathbf{w}, \mathfrak{D}_k) - \mathcal{L}_k(\mathbf{w}, \mathcal{D}_k)| \leq 2R_{n_k}(\mathcal{H}) + \sqrt{\frac{\ln(2m/\delta)}{2n_k}} \leq \frac{2C}{n_k^\alpha} + \sqrt{\frac{\ln(2m/\delta)}{2n_k}},$$

*where the second inequality is by the assumption on $R_n(\mathcal{H})$. Combining the inequalities above with $\mathbf{w} = \mathbf{w}_{new}$ and $\mathbf{w} = \mathbf{w}_{old}$, we have with probability at least $1 - \delta/m$*

$$\mathcal{L}_k(\mathbf{w}_{new}, \mathfrak{D}_k) - \mathcal{L}_k(\mathbf{w}_{old}, \mathfrak{D}_k) \leq \mathcal{L}_k(\mathbf{w}_{new}, \mathcal{D}_k) - \mathcal{L}_k(\mathbf{w}_{old}, \mathcal{D}_k) + \frac{4C}{n_k^\alpha} + 2\sqrt{\frac{\ln(2m/\delta)}{2n_k}}.$$

*Applying the union bound with the events above for $k = 1, \ldots, m$ leads to the conclusion of this lemma.*

### C.2  Proof of Lemma 2

**Proof** *Recall that $\mathbf{w}$ has two component $\mathbf{u}$ and $W$. The gradient of $h_k(\mathbf{w})$ with respect to $W$ and $\mathbf{u}$ are denoted by $\nabla_W h_k(\mathbf{w})$ and $\nabla_\mathbf{u} h_k(\mathbf{w})$, respectively. Hence,*

$$\nabla h_k(\mathbf{w}) = (\nabla_\mathbf{u} h_k(\mathbf{w}), \nabla_W h_k(\mathbf{w}))$$

*for $k = 1, \ldots, m$. Similarly, after adding the task-dependent heads, $\hat{\mathbf{w}}$ has four component $\mathbf{u}$, $W$, $\mathbf{U}$ and $\mathbf{V}$. The gradients $\nabla_\mathbf{u} \hat{h}_k(\hat{\mathbf{w}})$, $\nabla_W \hat{h}_k(\hat{\mathbf{w}})$ $\nabla_\mathbf{U} \hat{h}_k(\hat{\mathbf{w}})$ and $\nabla_\mathbf{V} \hat{h}_k(\hat{\mathbf{w}})$ are defined correspondingly, and*

$$\nabla \hat{h}_k(\hat{\mathbf{w}}) = \left(\nabla_\mathbf{u} \hat{h}_k(\hat{\mathbf{w}}), \nabla_W \hat{h}_k(\hat{\mathbf{w}}), \nabla_\mathbf{U} \hat{h}_k(\hat{\mathbf{w}}), \nabla_\mathbf{V} \hat{h}_k(\hat{\mathbf{w}})\right).$$

*Recall that*

$$\hat{h}_k(\hat{\mathbf{w}}) = h_k(W + U_k V_k^\top, \mathbf{u}) \text{ for } k = 1, \ldots, m.$$

*Therefore,*

$$\nabla_\mathbf{u} \hat{h}_k(\hat{\mathbf{w}}) = \nabla_\mathbf{u} h_k(W + U_k V_k^\top, \mathbf{u}), \qquad \nabla_W \hat{h}_k(\hat{\mathbf{w}}) = \nabla_W h_k(W + U_k V_k^\top, \mathbf{u})$$

*and*

$$\nabla_\mathbf{U} \hat{h}_k(\hat{\mathbf{w}}) = \left(\mathbf{0}, \ldots, \mathbf{0}, \underbrace{\nabla_W h_k(\mathbf{W} + U_k V_k^\top, \mathbf{u}) V_k}_{\text{The } k\text{th block}}, \mathbf{0}, \ldots, \mathbf{0}\right)^\top$$

$$\nabla_\mathbf{V} \hat{h}_k(\hat{\mathbf{w}}) = \left(\mathbf{0}, \ldots, \mathbf{0}, \underbrace{\nabla_W h_k(\mathbf{W} + U_k V_k^\top, \mathbf{u})^\top U_k}_{\text{The } k\text{th block}}, \mathbf{0}, \ldots, \mathbf{0}\right)^\top,$$

*where the sparsity patterns of $\nabla_\mathbf{U} \hat{h}_k(\hat{\mathbf{w}})$ and $\nabla_\mathbf{V} \hat{h}_k(\hat{\mathbf{w}})$ are because $\hat{h}_k$ does not depend on $U_j$ and $V_j$ with $j \neq k$.*

*Suppose $U_k V_k^\top = \mathbf{0}$ for all $k$. It holds that $h_k(\mathbf{w}) = \hat{h}_k(\hat{\mathbf{w}})$ and*

$$\nabla h_k(\mathbf{w}) = (\nabla_\mathbf{u} h_k(\mathbf{w}), \nabla_W h_k(\mathbf{w})) = \left(\nabla_\mathbf{u} \hat{h}_k(\hat{\mathbf{w}}), \nabla_W \hat{h}_k(\hat{\mathbf{w}})\right).$$

*Consider any $\boldsymbol{\alpha} = (\alpha_1, \ldots, \alpha_m) \in \mathbb{R}^m$. We have*

$$\lambda_{\min}\left(\left[\nabla \hat{h}_1(\hat{\mathbf{w}}), \ldots, \nabla \hat{h}_m(\hat{\mathbf{w}})\right]^\top \left[\nabla \hat{h}_1(\hat{\mathbf{w}}), \ldots, \nabla \hat{h}_m(\hat{\mathbf{w}})\right]\right)$$

$$= \min_{\boldsymbol{\alpha}, s.t. \|\boldsymbol{\alpha}\| = 1} \left\|\sum_{k=1}^m \alpha_k \nabla \hat{h}_k(\hat{\mathbf{w}})\right\|^2$$

$$= \min_{\boldsymbol{\alpha}, s.t. \|\boldsymbol{\alpha}\| = 1} \left(\left\|\sum_{k=1}^m \alpha_k \nabla_{\mathbf{u}} \hat{h}_k(\hat{\mathbf{w}})\right\|^2 + \left\|\sum_{k=1}^m \alpha_k \nabla_W \hat{h}_k(\hat{\mathbf{w}})\right\|^2 + \left\|\sum_{k=1}^m \alpha_k \nabla_{\mathbf{U}} \hat{h}_k(\hat{\mathbf{w}})\right\|^2 + \left\|\sum_{k=1}^m \alpha_k \nabla_{\mathbf{V}} \hat{h}_k(\hat{\mathbf{w}})\right\|^2\right)$$

$$= \min_{\boldsymbol{\alpha}, s.t. \|\boldsymbol{\alpha}\| = 1} \left(\left\|\sum_{k=1}^m \alpha_k \nabla h_k(\mathbf{w})\right\|^2 + \sum_{k=1}^m \alpha_k^2 \|\nabla_W h_k(\mathbf{w}) V_k\|_F^2 + \sum_{k=1}^m \alpha_k^2 \left\|\nabla_W h_k(\mathbf{w})^\top U_k\right\|_F^2\right)$$

$$\geq \lambda_{\min}\left(\left[\nabla h_1(\mathbf{w}), \ldots, \nabla h_m(\mathbf{w})\right]^\top \left[\nabla h_1(\mathbf{w}), \ldots, \nabla h_m(\mathbf{w})\right]\right)$$

$$+ \min_k \|\nabla_W h_k(\mathbf{w}) V_k\|_F^2 + \min_k \left\|\nabla_W h_k(\mathbf{w})^\top U_k\right\|_F^2,$$

*where the first two equalities are by definitions and the third equality is because $U_k V_k^\top = \mathbf{0}$ for all $k$.*

## C.3  Proof of Theorem 1

We present a formal statement of Theorem 1 below.

**Theorem** *Suppose Assumptions 1, 2 and 3 hold. Also, suppose, in Algorithm 1, set $\beta = \frac{1}{\epsilon\delta}$, $\gamma_1 = \gamma_2 = \min\{\frac{5n_0\theta}{3|\mathcal{B}|}, \frac{5m\theta}{3|\mathcal{B}_c|}, \frac{\epsilon^4\delta^2|\mathcal{B}_k|}{26880\sigma_h^2 \tilde{C}_{\nabla h}^2}\}$, $\theta = \min\{\frac{\epsilon^4\delta^2 \min\{|\mathcal{B}_c|, |\mathcal{B}_k|\}}{672(\sigma_{\nabla h}^2 + L_h^2)}, \frac{\epsilon^2 \min\{|\mathcal{B}|, |\mathcal{B}_{1i}|, |\mathcal{B}_{2i}|\}}{1344L_f^2(\sigma_{\nabla g}^2 + L_g^2)}\}$ and $\eta = \min\{\frac{1}{12(L_F + \beta L_H)}, \frac{\theta}{8\sqrt{3}L_F},$ $\frac{\theta}{8\sqrt{3}L_H\beta}, \frac{\gamma_1|\mathcal{B}|}{40\sqrt{6}L_g L_f \tilde{C}_{\nabla g} n_0}, \frac{\gamma_2|\mathcal{B}_c|}{40\sqrt{6}\beta L_h \tilde{C}_{\nabla h} m}\}$ , where $\tilde{C}_{\nabla g} := \sigma_{\nabla g} + L_g$, $\tilde{C}_{\nabla h} := \sigma_{\nabla h} + L_h$, $L_f := \frac{\tau}{c_g}$, $L_{\nabla f} := \frac{\tau}{c_g^2}$, $L_F := 2(L_{\nabla g} L_f + L_{\nabla f} L_g^2)$ and $L_H := 2L_{\nabla h} + L_h^2$. Then there exists $\boldsymbol{\lambda} \in \mathbb{R}_+^m$ such that after $T = O(\epsilon^{-7}\delta^{-3})$ iterations Algorithm 1 satisfies $\mathbb{E}\left[\|\nabla F(\mathbf{w}^{\hat{t}}, \mathcal{D}) + \nabla \boldsymbol{h}(\mathbf{w}^{\hat{t}})\boldsymbol{\lambda}\|\right] \leq \epsilon$, $\mathbb{E}[\|[\boldsymbol{h}(\mathbf{w}^{\hat{t}})]_+\|] \leq \epsilon$, $\mathbb{E}[\boldsymbol{\lambda}^\top[\boldsymbol{h}(\mathbf{w}^{\hat{t}})]_+] \leq \epsilon$, where $\hat{t}$ selected uniformly at random from $\{1, \cdots, T\}$.*

In this section, we present the proof of the Theorem 1. Recall that the problem is formulated as

$$\min_{\mathbf{w}} \ F(\mathbf{w}, \mathcal{D}) := \frac{1}{n_0} \sum_{i=1}^{n_0} (f(g_{1i}(\mathbf{w})) + f(g_{2i}(\mathbf{w}))) \quad \text{s.t.} \quad \frac{1}{m} h_k(\mathbf{w}; \mathcal{D}_k) \leq 0, \ k = 1, \cdots, m. \quad (12)$$

with $f(\cdot) = \tau \log(\cdot)$. With the quadratic penalty method, the problem is converted to

$$\min_{\mathbf{w}} \ \Phi(\mathbf{w}) := F(\mathbf{w}, \mathcal{D}) + \underbrace{\frac{1}{m} \sum_{k=1}^m \frac{\beta}{2} ([h_k(\mathbf{w}; \mathcal{D}_k)]_+)^2}_{H(\mathbf{w})}. \quad (13)$$

By Assumption 1, we can get $f$ is $L_f$-Lipschitz continuous and $L_{\nabla f}$-smooth with $L_f = \frac{\tau}{c_g}$ and $L_{\nabla f} = \frac{\tau}{c_g^2}$. By noticing that $\ell_{ce}$ is a cross entropy loss, we find that $|h_k(\cdot)|$ can be bounded by a constant $C_h$ with $C_h = 2$. Then, we can get $\Phi(\mathbf{w})$ is $L_\beta$-smooth with $L_\beta := L_F + \beta L_H$ where $L_F := 2(L_{\nabla g} L_f + L_{\nabla f} L_g^2)$ and $L_H := L_{\nabla h} C_h + L_h^2$. We also define $\tilde{C}_{\nabla g} := \sigma_{\nabla g} + L_g$ and $\tilde{C}_{\nabla h} := \sigma_{\nabla h} + L_h$. To facilitate our discussion, we let

$$v_1^t = (1 - \theta)v_1^{t-1} + \theta G_1^t,$$
$$v_2^t = (1 - \theta)v_2^{t-1} + \theta G_2^t,$$
$$v^t = v_1^t + v_2^t.$$

To prove our main theorem, we need following lemmas.

**Lemma 3** *If $\theta \leq \frac{1}{3}$, the gradient variance $\Delta_1^t := \|v_1^t - \nabla F(\mathbf{w}^t, \mathcal{D})\|^2$ can be bounded as*

$$
\mathbb{E}[\Delta_1^{t+1}] \leq (1-\theta)\mathbb{E}[\Delta_1^t] + \frac{2L_F^2}{\theta}\mathbb{E}[\|\mathbf{w}^{t+1} - \mathbf{w}^t\|^2] + 5\theta L_f^2 \tilde{C}_{\nabla g}^2 \mathbb{E}[\Xi_1^{t+1} + \Xi_2^{t+1}]
$$
$$
+ 3L_f^2 \tilde{C}_{\nabla g}^2 \mathbb{E}\left[\frac{1}{n_0}\sum_{i \in \mathcal{B}^{t+1}}\left(\left\|u_{1i}^{t+1} - u_{1i}^t\right\|^2 + \left\|u_{2i}^{t+1} - u_{2i}^t\right\|^2\right)\right] + \frac{2\theta^2 L_f^2 (\sigma_{\nabla g}^2 + L_g^2)}{min\{|\mathcal{B}|, |\mathcal{B}_{1i}|, |\mathcal{B}_{2i}|\}}, \tag{14}
$$

*with $\Xi_1^{t+1} := \frac{1}{n_0}\|\mathbf{u}_1^{t+1} - \mathbf{g}_1(\mathbf{w}^{t+1})\|^2 = \frac{1}{n_0}\sum_{i=1}^{n_0}\|u_{1i}^{t+1} - g_{1i}(\mathbf{w}^{t+1})\|^2$ and $\Xi_2^{t+1} := \frac{1}{n_0}\|\mathbf{u}_2^{t+1} - \mathbf{g}_2(\mathbf{w}^{t+1})\|^2 = \frac{1}{n_0}\sum_{i=1}^{n_0}\|u_{2i}^{t+1} - g_{2i}(\mathbf{w}^{t+1})\|^2$.*

**Proof**

$$
\Delta_1^{t+1} = \|v_1^{t+1} - \nabla F(\mathbf{w}^{t+1})\|^2 = \|(1-\theta)v_1^t + \theta G_1^t - \nabla F(\mathbf{w}^{t+1})\|^2
$$
$$
= \left\|①+②+③+④\right\|^2,
$$

*where $①$, $②$, $③$, $④$ are defined as*

$$
① = (1-\theta)(v_1^t - \nabla F(\mathbf{w}^t)), \quad ② = (1-\theta)(\nabla F(\mathbf{w}^t) - \nabla F(\mathbf{w}^{t+1})),
$$
$$
③ = \frac{\theta}{|\mathcal{B}|}\sum_{i \in \mathcal{B}^{t+1}} \nabla \hat{g}_{1i}(\mathbf{w}^{t+1})\left(\nabla f(u_{1i}^t) - \nabla f(g_{1i}(\mathbf{w}^{t+1}))\right) + \nabla \hat{g}_{2i}(\mathbf{w}^{t+1})\left(\nabla f(u_{2i}^t) - \nabla f(g_{2i}(\mathbf{w}^{t+1}))\right),
$$
$$
④ = \frac{\theta}{|\mathcal{B}|}\sum_{i \in \mathcal{B}^{t+1}} \nabla \hat{g}_{1i}(\mathbf{w}^{t+1})\nabla f(g_{1i}(\mathbf{w}^{t+1})) + \nabla \hat{g}_{2i}(\mathbf{w}^{t+1})\nabla f(g_{2i}(\mathbf{w}^{t+1})) - \nabla F(\mathbf{w}^{t+1}).
$$

*Note that $\mathbb{E}_t[\langle ①, ④\rangle] = \mathbb{E}_t[\langle ②, ④\rangle] = 0$. Then, by the Young's inequality, we can get*

$$
\mathbb{E}_t\left[\left\|①+②+③+④\right\|^2\right]
$$
$$
= \|①\|^2 + \|②\|^2 + \mathbb{E}_t\|③\|^2 + \mathbb{E}_t\|④\|^2 + 2\langle ①, ②\rangle + 2\mathbb{E}_t[\langle ①, ③\rangle] + 2\mathbb{E}_t[\langle ②, ③\rangle] + 2\mathbb{E}_t[\langle ③, ④\rangle]
$$
$$
\leq (1+\theta)\|①\|^2 + 2\left(1 + \frac{1}{\theta}\right)\|②\|^2 + \frac{2+3\theta}{\theta}\mathbb{E}_t\|③\|^2 + 2\mathbb{E}_t\|④\|^2.
$$

*We can also get*

$$
(1+\theta)\|①\|^2 = (1+\theta)(1-\theta)^2\|v_1^t - \nabla F(\mathbf{w}^t)\|^2 \leq (1-\theta)\|v_1^t - \nabla F(\mathbf{w}^t)\|^2
$$
$$
2\left(1 + \frac{1}{\theta}\right)\|②\|^2 = 2\left(1 + \frac{1}{\theta}\right)(1-\theta)^2\|\nabla F(\mathbf{w}^t) - \nabla F(\mathbf{w}^{t+1})\|^2 \leq \frac{2L_F^2}{\theta}\|\mathbf{w}^{t+1} - \mathbf{w}^t\|^2
$$

$$
\frac{2+3\theta}{\theta}\mathbb{E}_t\left[\|③\|^2\right]
$$
$$
= \frac{2+3\theta}{\theta}\frac{\theta^2}{|\mathcal{B}|}\mathbb{E}_t\sum_{i \in \mathcal{B}^{t+1}}\left(\left\|\nabla \hat{g}_{1i}(\mathbf{w}^{t+1})\right\|^2 \left\|\nabla f(u_{1i}^t) - \nabla f(g_{1i}(\mathbf{w}^{t+1}))\right\|^2 + \left\|\nabla \hat{g}_{2i}(\mathbf{w}^{t+1})\right\|^2 \left\|\nabla f(u_{2i}^t) - \nabla f(g_{2i}(\mathbf{w}^{t+1}))\right\|^2\right)
$$

We first bound the first term

$$\frac{(2+3\theta)\theta}{|\mathcal{B}|}\mathbb{E}_t \sum_{i\in\mathcal{B}^{t+1}} \left\|\nabla\hat{g}_{1i}(\mathbf{w}^{t+1})\right\|^2 \left\|\nabla f(u_{1i}^t) - \nabla f(g_{1i}(\mathbf{w}^{t+1}))\right\|^2$$

$$\leq \frac{(2+3\theta)\theta L_f^2}{|\mathcal{B}|}\mathbb{E}_t \left[\sum_{i\in\mathcal{B}^{k+1}} \left\|\nabla\hat{g}_{1i}(\mathbf{w}^{t+1})\right\|^2 \left\|u_{1i}^t - g_{1i}(\mathbf{w}^{t+1})\right\|^2\right]$$

$$= (2+3\theta)\theta L_f^2 \mathbb{E}_t \left[\frac{1}{|\mathcal{B}|}\sum_{i\in\mathcal{B}^{t+1}} \mathbb{E}_t\left[\left\|\nabla\hat{g}_{1i}(\mathbf{w}^{k+1})\right\|^2 | i\in\mathcal{B}^{t+1}\right] \left\|u_{1i}^t - g_{1i}(\mathbf{w}^{k+1})\right\|^2\right]$$

$$\leq (2+3\theta)\theta L_f^2 \tilde{C}_{\nabla g}^2 \mathbb{E}_t \left[\frac{1}{|\mathcal{B}|}\sum_{i\in\mathcal{B}^{t+1}} \left\|u_{1i}^t - g_{1i}(\mathbf{w}^{t+1})\right\|^2\right]$$

$$\leq (2+3\theta)\theta L_f^2 \tilde{C}_{\nabla g}^2 \left((1+\delta)\mathbb{E}_t\left[\frac{1}{n_0}\sum_{i=1}^{n_0}\left\|u_{1i}^{t+1} - g_{1i}(\mathbf{w}^{t+1})\right\|^2\right] + (1+1/\delta)\mathbb{E}_t\left[\frac{1}{n_0}\sum_{i=1}^{n_0}\left\|u_{1i}^{t+1} - u_{1i}^t\right\|^2\right]\right)$$

$$= (2+3\theta)\theta L_f^2 \tilde{C}_{\nabla g}^2 \left((1+\delta)\mathbb{E}_t\left[\frac{1}{n_0}\sum_{i=1}^{n_0}\left\|u_{1i}^{t+1} - g_i(\mathbf{w}^{t+1})\right\|^2\right] + (1+1/\delta)\mathbb{E}_t\left[\frac{1}{n_0}\sum_{i\in\mathcal{B}^{t+1}}\left\|u_{1i}^{t+1} - u_{1i}^t\right\|^2\right]\right)$$

If $\theta \leq \frac{1}{3}$ and $\delta = \frac{3\theta}{2}$, we have $(2+3\theta)\theta(1+\delta) \leq 5\theta$ and $(2+3\theta)\theta(1+1/\delta) \leq 3$. And similarly, we can get the bound for the second term. Then, by combining them, we can get

$$\frac{2+3\theta}{\theta}\mathbb{E}\left[\|③\|^2\right] \leq 5\theta L_f^2\tilde{C}_{\nabla g}^2 \mathbb{E}[\Xi_1^{t+1} + \Xi_2^{t+1}] + 3L_f^2\tilde{C}_{\nabla g}^2 \mathbb{E}\left[\frac{1}{n_0}\sum_{i\in\mathcal{B}^{t+1}}\left(\left\|u_{1i}^{t+1} - u_{1i}^t\right\|^2 + \left\|u_{2i}^{t+1} - u_{2i}^t\right\|^2\right)\right].$$

$$\mathbb{E}_t\left[\|④\|^2\right]$$

$$=\theta^2\mathbb{E}_t\left[\left\|\frac{1}{|\mathcal{B}|}\sum_{i\in\mathcal{B}^{t+1}}\nabla\hat{g}_{1i}(\mathbf{w}^{k+1})\nabla f(g_{1i}(\mathbf{w}^{k+1})) - \frac{1}{n_0}\sum_{i=1}^{n_0}\nabla g_{1i}(\mathbf{w}^{t+1})\nabla f(g_{1i}(\mathbf{w}^{t+1}))\right\|^2\right]$$

$$+ \theta^2\mathbb{E}_t\left[\left\|\frac{1}{|\mathcal{B}|}\sum_{i\in\mathcal{B}^{t+1}}\nabla\hat{g}_{2i}(\mathbf{w}^{k+1})\nabla f(g_{2i}(\mathbf{w}^{t+1})) - \frac{1}{n_0}\sum_{i=1}^{n_0}\nabla g_{2i}(\mathbf{w}^{t+1})\nabla f(g_{2i}(\mathbf{w}^{t+1}))\right\|^2\right]$$

$$=\theta^2\mathbb{E}_t\left[\left\|\frac{1}{|\mathcal{B}|}\sum_{i\in\mathcal{B}^{t+1}}\nabla\hat{g}_{1i}(\mathbf{w}^{t+1})\nabla f(g_{1i}(\mathbf{w}^{t+1})) - \frac{1}{|\mathcal{B}|}\sum_{i\in\mathcal{B}^{t+1}}\nabla g_{1i}(\mathbf{w}^{t+1})\nabla f(g_{1i}(\mathbf{w}^{t+1}))\right\|^2\right]$$

$$+ \theta^2\mathbb{E}_t\left[\left\|\frac{1}{|\mathcal{B}|}\sum_{i\in\mathcal{B}^{t+1}}\nabla g_{1i}(\mathbf{w}^{t+1})\nabla f(g_{1i}(\mathbf{w}^{t+1})) - \frac{1}{n_0}\sum_{i=1}^{n_0}\nabla g_{1i}(\mathbf{w}^{t+1})\nabla f(g_{1i}(\mathbf{w}^{t+1}))\right\|^2\right]$$

$$+ \theta^2\mathbb{E}_t\left[\left\|\frac{1}{|\mathcal{B}|}\sum_{i\in\mathcal{B}^{t+1}}\nabla\hat{g}_{2i}(\mathbf{w}^{t+1})\nabla f(g_{2i}(\mathbf{w}^{t+1})) - \frac{1}{|\mathcal{B}|}\sum_{i\in\mathcal{B}^{t+1}}\nabla g_{2i}(\mathbf{w}^{t+1})\nabla f(g_{2i}(\mathbf{w}^{t+1}))\right\|^2\right]$$

$$+ \theta^2\mathbb{E}_t\left[\left\|\frac{1}{|\mathcal{B}|}\sum_{i\in\mathcal{B}^{t+1}}\nabla g_{2i}(\mathbf{w}^{t+1})\nabla f(g_{2i}(\mathbf{w}^{t+1})) - \frac{1}{n_0}\sum_{i=1}^{n_0}\nabla g_{2i}(\mathbf{w}^{t+1})\nabla f(g_{2i}(\mathbf{w}^{t+1}))\right\|^2\right]$$

$$\leq \frac{2\theta^2 L_f^2(\sigma_{\nabla g}^2 + L_g^2)}{min\{|\mathcal{B}|, |\mathcal{B}_{1i}|, |\mathcal{B}_{2i}|\}}.$$

Therefore, we can get

$$\mathbb{E}[\Delta_1^{t+1}] \leq (1-\theta)\mathbb{E}[\Delta_1^t] + \frac{2L_F^2}{\theta}\mathbb{E}[\|\mathbf{w}^{t+1} - \mathbf{w}^t\|^2] + 5\theta L_f^2\tilde{C}_{\nabla g}^2 \mathbb{E}[\Xi_1^{t+1} + \Xi_2^{t+1}]$$

$$+ 3L_f^2\tilde{C}_{\nabla g}^2 \mathbb{E}\left[\frac{1}{n_0}\sum_{i\in\mathcal{B}^{t+1}}\left(\left\|u_{1i}^{t+1} - u_{1i}^t\right\|^2 + \left\|u_{2i}^{t+1} - u_{2i}^t\right\|^2\right)\right] + \frac{2\theta^2 L_f^2(\sigma_{\nabla g}^2 + L_g^2)}{min\{|\mathcal{B}|, |\mathcal{B}_{1i}|, |\mathcal{B}_{2i}|\}}.$$

**Lemma 4** *If $\gamma_1 \leq 1/5$, function value variance $\Xi_1^t := \frac{1}{n_0}\|\boldsymbol{u}_1^t - \boldsymbol{g}_1(\mathbf{w}^t)\|^2$ can be bounded as*

$$\mathbb{E}[\Xi_1^{t+1}] \leq \left(1 - \frac{\gamma_1|\mathcal{B}|}{4n_0}\right)\mathbb{E}\left[\Xi_1^t\right] + \frac{5n_0 L_g^2 \mathbb{E}[\|\mathbf{w}^{t+1} - \mathbf{w}^t\|^2]}{\gamma_1|\mathcal{B}|} + \frac{2\gamma_1^2\sigma_g^2|\mathcal{B}|}{n_0|\mathcal{B}_{1i}|} - \frac{1}{4n_0}\mathbb{E}\left[\sum_{i\in\mathcal{B}^{t+1}}\|u_{1i}^{t+1} - u_{1i}^t\|^2\right]. \quad (15)$$

**Lemma 5** *If $\gamma_1 \leq 1/5$, function value variance $\Xi_2^t := \frac{1}{n_0}\|\boldsymbol{u}_2^t - \boldsymbol{g}_2(\mathbf{w}^t)\|^2$ can be bounded as*

$$\mathbb{E}[\Xi_2^{t+1}] \leq \left(1 - \frac{\gamma_1|\mathcal{B}|}{4n_0}\right)\mathbb{E}\left[\Xi_2^t\right] + \frac{5n_0 L_g^2 \mathbb{E}[\|\mathbf{w}^{t+1} - \mathbf{w}^t\|^2]}{\gamma_1|\mathcal{B}|} + \frac{2\gamma_1^2\sigma_g^2|\mathcal{B}|}{n_0|\mathcal{B}_{2i}|} - \frac{1}{4n_0}\mathbb{E}\left[\sum_{i\in\mathcal{B}^{t+1}}\|u_{2i}^{t+1} - u_{2i}^t\|^2\right]. \quad (16)$$

Since the proof of Lemma 4 and Lemma 5 are almost the same, we only presents the proof of Lemma 4 as follows.

**Proof** *Define $\phi_1^t(\boldsymbol{u}_1) = \frac{1}{2}\|\boldsymbol{u}_1 - \boldsymbol{g}_1(\mathbf{w}^k)\|^2 = \frac{1}{2}\sum_{i=1}^{n_0}\|u_{1i} - g_{1i}(\mathbf{w}^k)\|^2$, which is 1-strongly convex.*

$$\phi_1^{t+1}(\boldsymbol{u}_1^{t+1}) = \frac{1}{2}\|\boldsymbol{u}_1^{t+1} - \boldsymbol{g}_1(\mathbf{w}^{t+1})\|^2 = \frac{1}{2}\|\boldsymbol{u}_1^t - \boldsymbol{g}_1(\mathbf{w}^{t+1})\|^2 + \langle \boldsymbol{u}_1^k - \boldsymbol{g}_1(\mathbf{w}^{t+1}), \boldsymbol{u}_1^{t+1} - \boldsymbol{u}_1^t\rangle + \frac{1}{2}\|\boldsymbol{u}_1^{t+1} - \boldsymbol{u}_1^t\|^2$$

$$= \frac{1}{2}\|\boldsymbol{u}_1^t - \boldsymbol{g}_1(\mathbf{w}^{t+1})\|^2 + \sum_{i\in\mathcal{B}^{t+1}}\langle u_{1i}^t - \hat{g}_{1i}(\mathbf{w}^{t+1}), u_{1i}^{t+1} - u_{1i}^t\rangle + \frac{1}{2}\sum_{i\in\mathcal{B}^{t+1}}\|u_{1i}^{t+1} - u_{1i}^t\|^2$$

$$+ \sum_{i\in\mathcal{B}^{t+1}}\langle \hat{g}_{1i}(\mathbf{w}^{t+1}) - g_{1i}(\mathbf{w}^{k+1}), u_{1i}^{t+1} - u_{1i}^t\rangle$$

$$(17)$$

*Note that $u_{1i}^t - \hat{g}_{1i}(\mathbf{w}^{t+1}) = (u_{1i}^t - u_{1i}^{t+1})/\gamma_1$ and $2\langle b - a, a - c\rangle \leq \|b - c\|^2 - \|a - b\|^2 - \|a - c\|^2$.*

$$\sum_{i\in\mathcal{B}^{t+1}}\langle u_{1i}^k - \hat{g}_{1i}(\mathbf{w}^{t+1}), u_{1i}^{t+1} - u_{1i}^t\rangle$$

$$= \sum_{i\in\mathcal{B}^{t+1}}\langle u_{1i}^t - \hat{g}_{1i}(\mathbf{w}^{t+1}), g_{1i}(\mathbf{w}^{k+1}) - u_{1i}^t\rangle + \sum_{i\in\mathcal{B}^{t+1}}\langle u_{1i}^t - \hat{g}_{1i}(\mathbf{w}^{t+1}), u_{1i}^{t+1} - g_{1i}(\mathbf{w}^{t+1})\rangle$$

$$= \sum_{i\in\mathcal{B}^{t+1}}\langle u_{1i}^t - \hat{g}_{1i}(\mathbf{w}^{t+1}), g_{1i}(\mathbf{w}^{t+1}) - u_{1i}^t\rangle + \frac{1}{\gamma_1}\sum_{i\in\mathcal{B}^{t+1}}\langle u_{1i}^t - u_{1i}^{t+1}, u_{1i}^{t+1} - g_{1i}(\mathbf{w}^{t+1})\rangle$$

$$\leq \sum_{i\in\mathcal{B}^{t+1}}\langle u_{1i}^t - \hat{g}_{1i}(\mathbf{w}^{t+1}), g_{1i}(\mathbf{w}^{t+1}) - u_{1i}^t\rangle$$

$$+ \frac{1}{2\gamma_1}\sum_{i\in\mathcal{B}^{t+1}}\left(\|u_{1i}^t - g_{1i}(\mathbf{w}^{t+1})\|^2 - \|u_{1i}^{t+1} - u_{1i}^t\|^2 - \|u_{1i}^{t+1} - g_{1i}(\mathbf{w}^{t+1})\|^2\right)$$

*If $\gamma_1 \leq \frac{1}{5}$, we have*

$$-\frac{1}{2}\left(\frac{1}{\gamma_1} - 1 - \frac{\gamma_1 + 1}{4\gamma_1}\right)\sum_{i\in\mathcal{B}^{t+1}}\|u_{1i}^{t+1} - u_{1i}^t\|^2 + \sum_{i\in\mathcal{B}^{t+1}}\langle \hat{g}_{1i}(\mathbf{w}^{t+1}) - g_{1i}(\mathbf{w}^{t+1}), u_{1i}^{t+1} - u_{1i}^t\rangle$$

$$\leq -\frac{1}{4\gamma_1}\sum_{i\in\mathcal{B}^{t+1}}\|u_{1i}^{t+1} - u_{1i}^t\|^2 + \gamma_1\sum_{i\in\mathcal{B}^{t+1}}\left\|\hat{g}_{1i}(\mathbf{w}^{t+1}) - g_{1i}(\mathbf{w}^{t+1})\right\|^2 + \frac{1}{4\gamma_1}\sum_{i\in\mathcal{B}^{t+1}}\|u_{1i}^{t+1} - u_{1i}^t\|^2$$

$$= \gamma_1\sum_{i\in\mathcal{B}^{t+1}}\left\|\hat{g}_{1i}(\mathbf{w}^{t+1}) - g_{1i}(\mathbf{w}^{t+1})\right\|^2.$$

*Then we can get*

$$\frac{1}{2}\|\boldsymbol{u}_1^{t+1} - \boldsymbol{g}_1(\mathbf{w}^{t+1})\|^2 \leq \frac{1}{2}\|\boldsymbol{u}_1^t - \boldsymbol{g}_1(\mathbf{w}^{t+1})\|^2 + \frac{1}{2\gamma_1}\sum_{i\in\mathcal{B}^{t+1}}\|u_{1i}^t - g_{1i}(\mathbf{w}^{t+1})\|^2$$

$$- \frac{1}{2\gamma_1}\sum_{i\in\mathcal{B}^{t+1}}\|u_{1i}^{t+1} - g_{1i}(\mathbf{w}^{t+1})\|^2$$

$$+ \gamma_1\sum_{i\in\mathcal{B}^{t+1}}\left\|\hat{g}_{1i}(\mathbf{w}^{t+1}) - g_{1i}(\mathbf{w}^{t+1})\right\|^2 - \frac{\gamma_1 + 1}{8\gamma_1}\sum_{i\in\mathcal{B}^{t+1}}\|u_{1i}^{t+1} - u_{1i}^t\|^2$$

$$+ \sum_{i\in\mathcal{B}^{t+1}}\langle u_{1i}^t - \hat{g}_{1i}(\mathbf{w}^{t+1}), g_{1i}(\mathbf{w}^{t+1}) - u_{1i}^t\rangle.$$

Note that $\frac{1}{2\gamma_1}\sum_{i\notin\mathcal{B}^{t+1}}\|u_{1i}^t - g_{1i}(\mathbf{w}^{t+1})\|^2 = \frac{1}{2\gamma_1}\sum_{i\notin\mathcal{B}^{t+1}}\|u_{1i}^{t+1} - g_{1i}(\mathbf{w}^{t+1})\|^2$, which implies that

$$\frac{1}{2\gamma_1}\sum_{i\in\mathcal{B}^{t+1}}\left(\|u_{1i}^t - g_{1i}(\mathbf{w}^{t+1})\|^2 - \|u_{1i}^{t+1} - g_{1i}(\mathbf{w}^{t+1})\|^2\right) = \frac{1}{2\gamma_1}\left(\|\boldsymbol{u}_1^t - \boldsymbol{g}_1(\mathbf{w}^{t+1})\|^2 - \|\boldsymbol{u}_1^{t+1} - \boldsymbol{g}_1(\mathbf{w}^{t+1})\|^2\right).$$

Besides, we also have $\mathbb{E}\left[\sum_{i\in\mathcal{B}^{t+1}}\left\|\hat{g}_{1i}(\mathbf{w}^{t+1}) - g_{1i}(\mathbf{w}^{t+1})\right\|^2\right] \leq \frac{|\mathcal{B}|\sigma_g^2}{|\mathcal{B}_{1i}|}$ and

$$\mathbb{E}\left[\sum_{i\in\mathcal{B}^{t+1}}\langle u_{1i}^t - \hat{g}_{1i}(\mathbf{w}^{t+1}), g_{1i}(\mathbf{w}^{t+1}) - u_{1i}^t\rangle\right] = \frac{|\mathcal{B}|}{n_0}\sum_{i=1}^{n_0}\langle u_{1i}^t - g_{1i}(\mathbf{w}^{t+1}), g_{1i}(\mathbf{w}^{t+1}) - u_{1i}^t\rangle$$

$$= -\frac{|\mathcal{B}|}{n_0}\|\boldsymbol{u}_1^t - \boldsymbol{g}_1(\mathbf{w}^{t+1})\|^2.$$

Then we can obtain

$$\left(\frac{1}{2} + \frac{1}{2\gamma_1}\right)\mathbb{E}\left[\|\boldsymbol{u}_1^{t+1} - \boldsymbol{g}_1(\mathbf{w}^{t+1})\|^2\right]$$

$$\leq \left(\frac{1}{2} + \frac{1}{2\gamma_1} - \frac{|\mathcal{B}|}{n_0}\right)\mathbb{E}\left[\|\boldsymbol{u}_1^t - \boldsymbol{g}_1(\mathbf{w}^{t+1})\|^2\right] + \frac{\gamma_1|\mathcal{B}|\sigma_g^2}{|\mathcal{B}_{1i}|} - \frac{\gamma_1 + 1}{8\gamma_1}\mathbb{E}\left[\sum_{i\in\mathcal{B}^{t+1}}\|u_{1i}^{t+1} - u_{1i}^t\|^2\right].$$

Divide both sides by $\frac{\gamma_1+1}{2\gamma_1}$ we can get

$$\mathbb{E}\left[\|\boldsymbol{u}_1^{t+1} - \boldsymbol{g}_1(\mathbf{w}^{t+1})\|^2\right] \leq \frac{\gamma_1 + 1 - 2\gamma_1\frac{|\mathcal{B}|}{n_0}}{\gamma_1 + 1}\mathbb{E}\left[\|\boldsymbol{u}_1^t - \boldsymbol{g}_1(\mathbf{w}^{t+1})\|^2\right] + \frac{2}{\gamma_1 + 1}\frac{\gamma_1^2|\mathcal{B}|\sigma_g^2}{|\mathcal{B}_{1i}|}$$

$$- \frac{1}{4}\mathbb{E}\left[\sum_{i\in\mathcal{B}^{t+1}}\|u_{1i}^{t+1} - u_{1i}^t\|^2\right].$$

Note that $\frac{\gamma_1 + 1 - 2\gamma_1\frac{|\mathcal{B}|}{n_0}}{\gamma_1 + 1} \leq \frac{\gamma_1(1 - \frac{|\mathcal{B}|}{n_0}) + 1}{\gamma_1 + 1} = 1 - \frac{\gamma_1|\mathcal{B}|}{(\gamma_1 + 1)n_0} \leq 1 - \frac{\gamma_1|\mathcal{B}|}{2n_0}$ and $\frac{1}{\gamma_1+1} \leq 1$ for $\gamma_1 \in (0, 1]$. Besides, we have $\|\boldsymbol{u}_1^t - \boldsymbol{g}_1(\mathbf{w}^{t+1})\|^2 \leq (1 + \frac{\gamma_1|\mathcal{B}|}{4n_0})\|\boldsymbol{u}_1^t - \boldsymbol{g}_1(\mathbf{w}^t)\|^2 + (1 + \frac{4n_0}{\gamma_1|\mathcal{B}|})\|\boldsymbol{g}_1(\mathbf{w}^{t+1}) - \boldsymbol{g}_1(\mathbf{w}^t)\|^2$ due to Young's inequality, $(1 + \frac{\gamma_1|\mathcal{B}|}{4n_0})(1 - \frac{\gamma_1|\mathcal{B}|}{2n_0}) \leq (1 - \frac{\gamma_1|\mathcal{B}|}{4n_0})$ and $(1 + \frac{4n_0}{\gamma_1|\mathcal{B}|})(1 - \frac{\gamma_1|\mathcal{B}|}{2n_0}) \leq \frac{5n_0}{\gamma_1|\mathcal{B}|}$.

$$\mathbb{E}\left[\Xi_1^{t+1}\right] = \mathbb{E}\left[\frac{1}{n_0}\|\boldsymbol{u}_1^{t+1} - \boldsymbol{g}_1(\mathbf{w}^{t+1})\|^2\right]$$

$$\leq \left(1 - \frac{\gamma_1|\mathcal{B}|}{4n_0}\right)\mathbb{E}\left[\frac{1}{n_0}\|\boldsymbol{u}_1^t - \boldsymbol{g}_1(\mathbf{w}^t)\|^2\right] + \frac{5n_0 L_g^2\|\mathbf{w}^{t+1} - \mathbf{w}^t\|^2}{\gamma_1|\mathcal{B}|} + \frac{2\gamma_1^2\sigma_g^2|\mathcal{B}|}{n_0|\mathcal{B}_{1i}|} - \frac{1}{4n_0}\mathbb{E}\left[\sum_{i\in\mathcal{B}^{t+1}}\|u_{1i}^{t+1} - u_{1i}^t\|^2\right]$$

$$= \left(1 - \frac{\gamma_1|\mathcal{B}|}{4n_0}\right)\mathbb{E}\left[\Xi_1^t\right] + \frac{5n_0 L_g^2\mathbb{E}[\|\mathbf{w}^{t+1} - \mathbf{w}^t\|^2]}{\gamma_1|\mathcal{B}|} + \frac{2\gamma_1^2\sigma_g^2|\mathcal{B}|}{n_0|\mathcal{B}_{1i}|} - \frac{1}{4n_0}\mathbb{E}\left[\sum_{i\in\mathcal{B}^{t+1}}\|u_{1i}^{t+1} - u_{1i}^t\|^2\right]$$

**Lemma 6** The gradient variance $\Delta_2^t := \|v_2^t - \nabla H(\mathbf{w}^t)\|^2$ can be bounded as

$$\mathbb{E}[\Delta_2^{t+1}] \leq (1 - \theta)\mathbb{E}[\Delta_2^t] + \frac{2\beta^2 L_H^2}{\theta}\mathbb{E}\left[\|\mathbf{w}^{t+1} - \mathbf{w}^t\|^2\right] + 5\theta\beta^2\tilde{C}_{\nabla h}^2\mathbb{E}[\Gamma_{t+1}]$$

$$+ \frac{3\beta^2\tilde{C}_{\nabla h}^2}{m}\mathbb{E}\left[\sum_{k\in\mathcal{B}_c^{t+1}}\|u_k^{t+1} - u_k^t\|^2\right] + \frac{\theta^2\beta^2 C_h^2(\sigma_{\nabla h}^2 + L_h^2)}{\min\{|\mathcal{B}_c|, |\mathcal{B}_k|\}} \tag{18}$$

with $\Gamma_{t+1} := \frac{1}{m}\|\boldsymbol{u}^{t+1} - \boldsymbol{h}(\mathbf{w}^{t+1})\|^2$.

**Proof**

$$\Delta_2^{t+1} = \|v_2^{t+1} - \nabla H(\mathbf{w}^{t+1})\|^2 = \|(1 - \theta)v_2^t + \theta G_2^t - \nabla H(\mathbf{w}^{t+1})\|^2$$

$$\|①+②+③+④\|^2,$$

where ①, ②, ③ and ④ are defined as

$$① = (1-\theta)(v_2^t - \nabla H(\mathbf{w}^t)), \quad ② = (1-\theta)(\nabla H(\mathbf{w}^t) - \nabla H(\mathbf{w}^{t+1})),$$

$$③ = \frac{\theta}{|\mathcal{B}_c|}\beta \sum_{k\in\mathcal{B}_c^{t+1}} \left([u_k^t]_+ \nabla\hat{h}_k(\mathbf{w}^{t+1}) - [h_k(\mathbf{w}^{t+1})]_+ \nabla\hat{h}_k(\mathbf{w}^{t+1})\right)$$

$$④ = \theta\left(\frac{1}{|\mathcal{B}_c|}\beta \sum_{k\in\mathcal{B}_c^{t+1}} [h_k(\mathbf{w}^{t+1})]_+ \nabla\hat{h}_k(\mathbf{w}^{t+1}) - \nabla H(\mathbf{w}^{t+1})\right)$$

Note that $\mathbb{E}_t[\langle①,④\rangle] = \mathbb{E}_t[\langle②,④\rangle] = 0$. Then, by the Young's inequality, we can get

$$\mathbb{E}_t\left[\|①+②+③+④\|^2\right]$$

$$=\|①\|^2 + \|②\|^2 + \mathbb{E}_t\|③\|^2 + \mathbb{E}_t\|④\|^2 + 2\langle①,②\rangle + 2\mathbb{E}_t[\langle①,③\rangle] + 2\mathbb{E}_t[\langle②,③\rangle] + 2\mathbb{E}_t[\langle③,④\rangle]$$

$$\leq(1+\theta)\|①\|^2 + 2\left(1+\frac{1}{\theta}\right)\|②\|^2 + \frac{2+3\theta}{\theta}\mathbb{E}_t\|③\|^2 + 2\mathbb{E}_t\|④\|^2.$$

We can also get

$$(1+\theta)\|①\|^2 = (1+\theta)(1-\theta)^2\|v_2^t - \nabla H(\mathbf{w}^t)\|^2 \leq (1-\theta)\|v_2^t - \nabla H(\mathbf{w}^t)\|^2$$

$$2\left(1+\frac{1}{\theta}\right)\|②\|^2 = 2\left(1+\frac{1}{\theta}\right)(1-\theta)^2\|\nabla H(\mathbf{w}^t) - \nabla H(\mathbf{w}^{t+1})\|^2$$

$$\leq\frac{2}{\theta}\left\|\frac{1}{m}\sum_{k=1}^m \beta\left(\nabla h_k(\mathbf{w}^{t+1})^\top[h_k(\mathbf{w}^{t+1})]_+ - \nabla h_k(\mathbf{w}^t)^\top[h_k(\mathbf{w}^t)]_+\right)\right\|^2$$

$$\leq\frac{2\beta^2 L_H^2}{\theta}\|\mathbf{w}^{t+1} - \mathbf{w}^t\|^2$$

$$\frac{2+3\theta}{\theta}\|③\|^2 \leq \frac{2+3\theta}{\theta}\frac{\theta^2\beta^2}{|\mathcal{B}_c|} \sum_{k\in\mathcal{B}_c^{t+1}} \|\nabla\hat{h}_k(\mathbf{w}^{t+1})\|^2 \|[u_k^t]_+ - [h_k(\mathbf{w}^{t+1})]_+\|^2$$

$$\leq\frac{(2+3\theta)\theta\beta^2}{|\mathcal{B}_c|} \sum_{k\in\mathcal{B}_c^{t+1}} \|\nabla\hat{h}_k(\mathbf{w}^{t+1})\|^2 \|u_k^t - h_k(\mathbf{w}^{t+1})\|^2$$

Consider that $\mathbf{w}^{t+1}$ and $u_k^t$ do not depend on either $\mathcal{B}_c^{t+1}$ or $\mathcal{B}_k$, we have

$$(2+3\theta)\theta\beta^2\mathbb{E}_t\left[\frac{1}{|\mathcal{B}_c|}\sum_{k\in\mathcal{B}_c^{t+1}} \|\nabla\hat{h}_k(\mathbf{w}^{t+1})\|^2 \|u_k^t - h_k(\mathbf{w}^{t+1})\|^2\right]$$

$$= (2+3\theta)\theta\beta^2\mathbb{E}_t\left[\frac{1}{|\mathcal{B}_c|}\sum_{k\in\mathcal{B}_c^{t+1}} \mathbb{E}_t\left[\|\nabla\hat{h}_k(\mathbf{w}^{t+1})\|^2 | k\in\mathcal{B}_c^{t+1}\right] \|u_k^t - h_k(\mathbf{w}^{t+1})\|^2\right]$$

$$\leq (2+3\theta)\theta\beta^2\tilde{C}_{\nabla h}^2\mathbb{E}_t\left[\frac{1}{|\mathcal{B}_c|}\sum_{k\in\mathcal{B}_c^{t+1}} \|u_k^t - h_k(\mathbf{w}^{t+1})\|^2\right]$$

$$\leq \frac{(2+3\theta)\theta(1+\delta)\beta^2\tilde{C}_{\nabla h}^2}{m} \sum_{k\in[m]} \mathbb{E}_t\left[\|u_k^{t+1} - h_k(\mathbf{w}^{t+1})\|^2\right] + \frac{(2+3\theta)\theta(1+1/\delta)\beta^2\tilde{C}_{\nabla h}^2}{m}\mathbb{E}_t\left[\sum_{k\in[m]} \|u_k^{t+1} - u_k^t\|^2\right]$$

$$= \frac{(2+3\theta)\theta(1+\delta)\beta^2\tilde{C}_{\nabla h}^2}{m} \sum_{k\in[m]} \mathbb{E}_t\left[\|u_k^{t+1} - h_k(\mathbf{w}^{t+1})\|^2\right] + \frac{(2+3\theta)\theta(1+1/\delta)\beta^2\tilde{C}_{\nabla h}^2}{m}\mathbb{E}_t\left[\sum_{k\in\mathcal{B}_c^{t+1}} \|u_k^{t+1} - u_k^t\|^2\right]$$

where the last equation holds by noting that $u_k^{t+1} = u_k^t$ for all $i\notin\mathcal{B}_c^{t+1}$.

*If $\theta \leq \frac{1}{3}$ and $\delta = \frac{3\theta}{2}$, we have $(2+3\beta)\beta(1+\delta) \leq 5\theta$ and $(2+3\beta)\beta(1+1/\delta) \leq 3$. Therefore, we can get*

$$\mathbb{E}\left[\frac{2+3\theta}{\theta}\|\text{③}\|^2\right] \leq 5\theta\beta^2\tilde{C}_{\nabla h}^2\mathbb{E}[\Gamma_{t+1}] + \frac{3\beta^2\tilde{C}_{\nabla h}^2}{m}\mathbb{E}\left[\sum_{k\in\mathcal{B}_c^{t+1}}\|u_k^{t+1}-u_k^t\|^2\right]$$

*Next, we give the upper bound of $\mathbb{E}_t\|\text{④}\|^2$.*

$$\mathbb{E}_t\|\text{④}\|^2 = \theta^2\beta^2\mathbb{E}_k\left[\left\|\frac{1}{|\mathcal{B}_c|}\sum_{k\in\mathcal{B}_c^{t+1}}[h_k(\mathbf{w}^{t+1})]_+\nabla\hat{h}_k(\mathbf{w}^{t+1}) - \frac{1}{m}\sum_{k=1}^m[h_k(\mathbf{w}^{t+1})]_+\nabla h_k(\mathbf{w}^{t+1})\right\|^2\right]$$

$$\leq \theta^2\beta^2\mathbb{E}_t\left[\left\|\frac{1}{|\mathcal{B}_c|}\sum_{k\in\mathcal{B}_c^{t+1}}[h_k(\mathbf{w}^{t+1})]_+\nabla\hat{h}_k(\mathbf{w}^{t+1}) - \frac{1}{|\mathcal{B}_c|}\sum_{k\in\mathcal{B}_c^{t+1}}[h_k(\mathbf{w}^{t+1})]_+\nabla h_k(\mathbf{w}^{t+1})\right\|^2\right]$$

$$+ \theta^2\beta^2\mathbb{E}_t\left[\left\|\frac{1}{|\mathcal{B}_c|}\sum_{k\in\mathcal{B}_c^{t+1}}[h_k(\mathbf{w}^{t+1})]_+\nabla h_k(\mathbf{w}^{t+1}) - \frac{1}{m}\sum_{k=1}^m[h_k(\mathbf{w}^{t+1})]_+\nabla h_k(\mathbf{w}^{t+1})\right\|^2\right]$$

$$\leq \frac{\theta^2\beta^2C_h^2(\sigma_{\nabla h}^2 + L_h^2)}{\min\{|\mathcal{B}_c|, |\mathcal{B}_k|\}}$$

*Combine above inequalities, we can get*

$$\mathbb{E}[\Delta_2^{t+1}] \leq (1-\theta)\mathbb{E}[\Delta_2^t] + \frac{2\beta^2 L_H^2}{\theta}\mathbb{E}\left[\|\mathbf{w}^{t+1}-\mathbf{w}^t\|^2\right] + 5\theta\beta^2\tilde{C}_{\nabla h}^2\mathbb{E}[\Gamma_{t+1}]$$

$$+ \frac{3\beta^2\tilde{C}_{\nabla h}^2}{m}\mathbb{E}\left[\sum_{k\in\mathcal{B}_c^{t+1}}\|u_k^{t+1}-u_k^t\|^2\right] + \frac{\theta^2\beta^2C_h^2(\sigma_{\nabla h}^2 + L_h^2)}{\min\{|\mathcal{B}_c|, |\mathcal{B}_k|\}}.$$

**Lemma 7** *If $\gamma_2 \leq 1/5$, function value variance $\Gamma_t := \frac{1}{m}\|\boldsymbol{u}^t - \boldsymbol{h}(\mathbf{w}^t)\|^2$ can be bounded as*

$$\mathbb{E}[\Gamma_{t+1}] \leq \left(1 - \frac{\gamma_2|\mathcal{B}_c|}{4m}\right)\mathbb{E}[\Gamma_t] + \frac{5mL_h^2\mathbb{E}[\|\mathbf{w}^{t+1}-\mathbf{w}^t\|^2]}{\gamma|\mathcal{B}_c|} + \frac{2\gamma_2^2\sigma_h^2|\mathcal{B}_c|}{m|\mathcal{B}_k|} - \frac{1}{4m}\mathbb{E}\left[\sum_{k\in\mathcal{B}_c^{t+1}}\|u_k^{t+1}-u_k^t\|^2\right]. \quad (19)$$

**Proof** *Define $\psi_k(\boldsymbol{u}) = \frac{1}{2}\|\boldsymbol{u} - \boldsymbol{h}(\mathbf{w}^t)\|^2 = \frac{1}{2}\sum_{k=1}^m\|u_k - h_k(\mathbf{w}^t)\|^2$, which is 1-strongly convex.*

$$\psi_{t+1}(\boldsymbol{u}^{t+1}) = \frac{1}{2}\|\boldsymbol{u}^{t+1} - \boldsymbol{h}(\mathbf{w}^{t+1})\|^2 = \frac{1}{2}\|\boldsymbol{u}^t - \boldsymbol{h}(\mathbf{w}^{t+1})\|^2 + \langle\boldsymbol{u}^t - \boldsymbol{h}(\mathbf{w}^{t+1}), \boldsymbol{u}^{t+1} - \boldsymbol{u}^t\rangle + \frac{1}{2}\|\boldsymbol{u}^{t+1} - \boldsymbol{u}^t\|^2$$

$$= \frac{1}{2}\|\boldsymbol{u}^t - \boldsymbol{h}(\mathbf{w}^{t+1})\|^2 + \sum_{k\in\mathcal{B}_c^{t+1}}\langle u_k^t - \hat{h}_k(\mathbf{w}^{t+1}), u_k^{t+1} - u_k^t\rangle + \frac{1}{2}\sum_{k\in\mathcal{B}_c^{t+1}}\|u_k^{t+1} - u_k^t\|^2$$

$$+ \sum_{k\in\mathcal{B}_c^{t+1}}\langle\hat{h}_k(\mathbf{w}^{t+1}) - h_k(\mathbf{w}^{t+1}), u_k^{t+1} - u_k^t\rangle \quad (20)$$

*Note that $u_k^t - \hat{h}_k(\mathbf{w}^{t+1}) = (q_i^k - q_i^{k+1})/\gamma_2$ and $2\langle b - a, a - c\rangle \leq \|b - c\|^2 - \|a - b\|^2 - \|a - c\|^2$.*

$$\sum_{k\in\mathcal{B}_c^{t+1}}\langle u_k^t - \hat{h}_k(\mathbf{w}^{t+1}), u_k^{t+1} - u_k^t\rangle$$

$$= \sum_{k\in\mathcal{B}_c^{t+1}}\langle u_k^t - \hat{h}_k(\mathbf{w}^{t+1}), h_k(\mathbf{w}^{t+1}) - u_k^t\rangle + \sum_{k\in\mathcal{B}_c^{t+1}}\langle u_k^t - \hat{h}_k(\mathbf{w}^{t+1}), u_k^{t+1} - h_k(\mathbf{w}^{t+1})\rangle$$

$$= \sum_{k\in\mathcal{B}_c^{t+1}}\langle u_k^t - \hat{h}_k(\mathbf{w}^{t+1}), h_k(\mathbf{w}^{t+1}) - u_k^t\rangle + \frac{1}{\gamma_2}\sum_{k\in\mathcal{B}_c^{t+1}}\langle u_k^t - u_k^{t+1}, u_k^{t+1} - h_k(\mathbf{w}^{t+1})\rangle$$

$$\leq \sum_{k\in\mathcal{B}_c^{t+1}}\langle u_k^t - \hat{h}_k(\mathbf{w}^{t+1}), h_k(\mathbf{w}^{t+1}) - u_k^t\rangle$$

$$+ \frac{1}{2\gamma_2}\sum_{k\in\mathcal{B}_c^{t+1}}\left(\|u_k^t - h_k(\mathbf{w}^{t+1})\|^2 - \|u_k^{t+1} - u_k^t\|^2 - \|u_k^{t+1} - h_k(\mathbf{w}^{t+1})\|^2\right)$$

*If $\gamma_2 \leq \frac{1}{5}$, we have*

$$-\frac{1}{2}\left(\frac{1}{\gamma_2} - 1 - \frac{\gamma_2+1}{4\gamma_2}\right)\sum_{k\in\mathcal{B}_c^{t+1}}\|u_k^{t+1} - u_k^t\|^2 + \sum_{k\in\mathcal{B}_c^{t+1}}\langle \hat{h}_k(\mathbf{w}^{t+1}) - h_k(\mathbf{w}^{t+1}), u_k^{t+1} - u_k^t\rangle$$

$$\leq -\frac{1}{4\gamma_2}\sum_{k\in\mathcal{B}_c^{t+1}}\|u_k^{t+1} - u_k^t\|^2 + \gamma_2\sum_{k\in\mathcal{B}_c^{t+1}}\left\|\hat{h}_k(\mathbf{w}^{t+1}) - h_k(\mathbf{w}^{t+1})\right\|^2 + \frac{1}{4\gamma_2}\sum_{k\in\mathcal{B}_c^{t+1}}\|u_k^{t+1} - u_k^t\|^2$$

$$=\gamma_2\sum_{k\in\mathcal{B}_c^{t+1}}\left\|\hat{h}_k(\mathbf{w}^{t+1}) - h_k(\mathbf{w}^{t+1})\right\|^2.$$

*Then we can get*

$$\frac{1}{2}\|\boldsymbol{u}^{t+1} - \boldsymbol{h}(\mathbf{w}^{t+1})\|^2 \leq \frac{1}{2}\|\boldsymbol{u}^t - \boldsymbol{h}(\mathbf{w}^{t+1})\|^2 + \frac{1}{2\gamma_2}\sum_{k\in\mathcal{B}_c^{t+1}}\|u_k^t - h_k(\mathbf{w}^{t+1})\|^2 - \frac{1}{2\gamma_2}\sum_{k\in\mathcal{B}_c^{t+1}}\|u_k^{t+1} - h_k(\mathbf{w}^{t+1})\|^2$$

$$+ \gamma_2\sum_{k\in\mathcal{B}_c^{t+1}}\left\|\hat{h}_k(\mathbf{w}^{t+1}) - h_k(\mathbf{w}^{t+1})\right\|^2 - \frac{\gamma_2+1}{8\gamma_2}\sum_{k\in\mathcal{B}_c^{t+1}}\|u_k^{t+1} - u_k^t\|^2$$

$$+ \sum_{k\in\mathcal{B}_2^{t+1}}\langle u_k^t - \hat{h}_k(\mathbf{w}^{t+1}), h_k(\mathbf{w}^{t+1}) - u_k^t\rangle.$$

*Note that $\frac{1}{2\gamma_2}\sum_{k\notin\mathcal{B}_c^{t+1}}\|u_k^t - h_k(\mathbf{w}^{t+1})\|^2 = \frac{1}{2\gamma_2}\sum_{k\notin\mathcal{B}_c^{t+1}}\|u_k^{t+1} - h_k(\mathbf{w}^{t+1})\|^2$, which implies that*

$$\frac{1}{2\gamma_2}\sum_{k\in\mathcal{B}_c^{t+1}}\left(\|u_k^t - h_k(\mathbf{w}^{t+1})\|^2 - \|u_k^{t+1} - h_k(\mathbf{w}^{t+1})\|^2\right) = \frac{1}{2\gamma_2}\left(\|\boldsymbol{u}^t - \boldsymbol{h}(\mathbf{w}^{t+1})\|^2 - \|\boldsymbol{u}^{t+1} - \boldsymbol{h}(\mathbf{w}^{t+1})\|^2\right).$$

*Besides, we also have $\mathbb{E}\left[\sum_{k\in\mathcal{B}_c^{t+1}}\left\|\hat{h}_k(\mathbf{w}^{t+1}) - h_k(\mathbf{w}^{t+1})\right\|^2\right] \leq \frac{|\mathcal{B}_c|\sigma_h^2}{|\mathcal{B}_k|}$ and*

$$\mathbb{E}\left[\sum_{k\in\mathcal{B}_c^{t+1}}\langle u_k^t - \hat{h}_k(\mathbf{w}^{t+1}), h_k(\mathbf{w}^{t+1}) - u_k^t\rangle\right] = \frac{|\mathcal{B}_c|}{m}\sum_{k=1}^{m}\langle u_k^t - h_k(\mathbf{w}^{t+1}), h_k(\mathbf{w}^{t+1}) - u_k^t\rangle$$

$$= -\frac{|\mathcal{B}_c|}{m}\|\boldsymbol{u}^t - \boldsymbol{h}(\mathbf{w}^{t+1})\|^2.$$

*Then we can obtain*

$$\left(\frac{1}{2} + \frac{1}{2\gamma_2}\right)\mathbb{E}\left[\|\boldsymbol{u}^{t+1} - \boldsymbol{h}(\mathbf{w}^{t+1})\|^2\right]$$

$$\leq \left(\frac{1}{2} + \frac{1}{2\gamma_2} - \frac{|\mathcal{B}_c|}{m}\right)\mathbb{E}\left[\|\boldsymbol{u}^t - \boldsymbol{h}(\mathbf{w}^{t+1})\|^2\right] + \frac{\gamma_2|\mathcal{B}_c|\sigma_h^2}{|\mathcal{B}_k|} - \frac{\gamma_2+1}{8\gamma_2}\mathbb{E}\left[\sum_{k\in\mathcal{B}_c^{t+1}}\|u_k^{t+1} - u_k^t\|^2\right].$$

*Divide both sides by $\frac{\gamma_2+1}{2\gamma_2}$ we can get*

$$\mathbb{E}\left[\|\boldsymbol{u}^{t+1} - \boldsymbol{h}(\mathbf{w}^{t+1})\|^2\right] \leq \frac{\gamma_2 + 1 - 2\gamma_2\frac{|\mathcal{B}_c|}{m}}{\gamma_2+1}\mathbb{E}\left[\|\boldsymbol{u}^t - \boldsymbol{h}(\mathbf{w}^{t+1})\|^2\right] + \frac{2}{\gamma_2+1}\frac{\gamma_2^2|\mathcal{B}_c|\sigma_h^2}{|\mathcal{B}_k|}$$

$$- \frac{1}{4}\mathbb{E}\left[\sum_{k\in\mathcal{B}_c^{t+1}}\|u_k^{t+1} - u_k^t\|^2\right].$$

*Note that $\frac{\gamma_2+1-2\gamma_2\frac{|\mathcal{B}_c|}{m}}{\gamma_2+1} \leq \frac{\gamma_2(1-\frac{|\mathcal{B}_c|}{m})+1}{\gamma_2+1} = 1 - \frac{\gamma_2|\mathcal{B}_c|}{(\gamma_2+1)m} \leq 1 - \frac{\gamma_2|\mathcal{B}_c|}{2m}$ and $\frac{1}{\gamma_2+1} \leq 1$ for $\gamma_2 \in (0,1]$. Besides, we have $\|\boldsymbol{u}^t - \boldsymbol{h}(\mathbf{w}^{t+1})\|^2 \leq (1+\frac{\gamma_2|\mathcal{B}_c|}{4m})\|\boldsymbol{u}^t - \boldsymbol{h}(\mathbf{w}^t)\|^2 + (1+\frac{4m}{\gamma_2|\mathcal{B}_c|})\|\boldsymbol{h}(\mathbf{w}^{t+1}) - \boldsymbol{h}(\mathbf{w}^t)\|^2$ due to Young's inequality,*

$(1 + \frac{\gamma_2|\mathcal{B}_c|}{4m})(1 - \frac{\gamma_2|\mathcal{B}_c|}{2m}) \le (1 - \frac{\gamma_2|\mathcal{B}_c|}{4m})$ and $(1 + \frac{4m}{\gamma_2|\mathcal{B}_c|})(1 - \frac{\gamma_2|\mathcal{B}_c|}{2m}) \le \frac{5m}{\gamma_2|\mathcal{B}_c|}$.

$$\mathbb{E}\left[\Gamma_{t+1}\right] = \mathbb{E}\left[\frac{1}{m}\|\boldsymbol{u}^{t+1} - \boldsymbol{h}(\mathbf{w}^{t+1})\|^2\right]$$

$$\le \left(1 - \frac{\gamma_2|\mathcal{B}_c|}{4m}\right)\mathbb{E}\left[\frac{1}{m}\|\boldsymbol{u}^t - \boldsymbol{h}(\mathbf{w}^t)\|^2\right] + \frac{5mL_h^2\|\mathbf{w}^{t+1} - \mathbf{w}^t\|^2}{\gamma_2|\mathcal{B}_c|} + \frac{2\gamma_2^2\sigma_h^2|\mathcal{B}_c|}{m|\mathcal{B}_k|} - \frac{1}{4m}\mathbb{E}\left[\sum_{k\in\mathcal{B}_c^{t+1}}\|u_k^{t+1} - u_k^t\|^2\right]$$

$$= \left(1 - \frac{\gamma_2|\mathcal{B}_c|}{4m}\right)\mathbb{E}\left[\Gamma_t\right] + \frac{5mL_h^2\mathbb{E}[\|\mathbf{w}^{t+1} - \mathbf{w}^t\|^2]}{\gamma_2|\mathcal{B}_c|} + \frac{2\gamma_2^2\sigma_h^2|\mathcal{B}_c|}{m|\mathcal{B}_k|} - \frac{1}{4m}\mathbb{E}\left[\sum_{k\in\mathcal{B}_c^{t+1}}\|u_k^{t+1} - u_k^t\|^2\right]$$

We state the main theorem again for convenience and present the proof.

**Theorem** *Suppose Assumptions 1, 2 and 3 hold, and set $\beta = \frac{1}{\epsilon\delta}$, $\theta = \min\{\frac{\epsilon^4\delta^2\min\{|\mathcal{B}_k|,|\mathcal{B}_c|\}}{672(\sigma_{\nabla h}^2 + L_h^2)}, \frac{\epsilon^2\min\{|\mathcal{B}|,|\mathcal{B}_{1i}|,|\mathcal{B}_{2i}|\}}{1344L_f^2(\sigma_{\nabla g}^2 + L_g^2)}\}$, $\gamma_1 = \gamma_2 = \min\{\frac{5n_0\theta}{3|\mathcal{B}|}, \frac{5m\theta}{3|\mathcal{B}_c|}, \frac{\epsilon^4\delta^2|\mathcal{B}_k|}{26880\sigma_h^2\bar{C}_{\nabla h}^2}\}$ and $\eta = \min\left\{\frac{1}{12(L_F + \beta L_H)}, \frac{\theta}{8\sqrt{3}L_F}, \frac{\theta}{8\sqrt{3}L_H\beta}, \frac{\gamma_1|\mathcal{B}|}{40\sqrt{6}L_gL_f\bar{C}_{\nabla g}n_0}, \frac{\gamma_2|\mathcal{B}_c|}{40\sqrt{6}\beta L_h\bar{C}_{\nabla h}m}\right\}$. Then there exists $\boldsymbol{\lambda}$ such that*

$$\mathbb{E}\left[\|\nabla F(\mathbf{w}^{\hat{t}}) + \nabla\boldsymbol{h}(\mathbf{w}^{\hat{t}})\boldsymbol{\lambda})\|\right] \le \epsilon$$

$$\mathbb{E}[\|[\boldsymbol{h}(\mathbf{w}^{\hat{t}})]_+\|] \le \epsilon$$

$$\mathbb{E}[\boldsymbol{\lambda}^\top[\boldsymbol{h}(\mathbf{w}^{\hat{t}})]_+] \le \epsilon$$

*with number of iterations $T$ of Algorithm 1 bounded by $O(\epsilon^{-7}\delta^{-3})$ and $\hat{t}$ selected uniformly at random from $\{1, \cdots, T\}$.*

**Proof** *Since $\Phi(\mathbf{w})$ is $L_\beta$-smooth with $L_\beta = L_F + \beta L_H$ where $L_F := 2(L_{\nabla g}L_f + L_{\nabla f}L_g^2)$ and $L_H := L_{\nabla h}C_h + L_hC_{\nabla h}$, we have*

$$\Phi(\mathbf{w}^{t+1}) \le \Phi(\mathbf{w}^t) + \langle\nabla\Phi(\mathbf{w}^t), \mathbf{w}^{t+1} - \mathbf{w}^t\rangle + \frac{L_\beta}{2}\|\mathbf{w}^{t+1} - \mathbf{w}^t\|^2$$

$$= \Phi(\mathbf{w}^t) + \langle v^t, \mathbf{w}^{t+1} - \mathbf{w}^t\rangle + \langle\nabla\Phi(\mathbf{w}^t) - v^t, \mathbf{w}^{t+1} - \mathbf{w}^t\rangle + \frac{L_\beta}{2}\|\mathbf{w}^{t+1} - \mathbf{w}^t\|^2 \qquad (21)$$

$$\le \Phi(\mathbf{w}^t) + \langle v^t, \mathbf{w}^{t+1} - \mathbf{w}^t\rangle + \left(\frac{L_\beta}{2} + \frac{1}{4\eta}\right)\|\mathbf{w}^{t+1} - \mathbf{w}^t\|^2 + \eta\|\nabla\Phi(\mathbf{w}^t) - v^t\|^2.$$

*Since $\mathbf{w}^{t+1} = \mathbf{w}^t - \eta v^t$, which is equivalent to $\mathbf{w}^{t+1} = \arg\min_{\mathbf{w}}\langle v^t, \mathbf{w}\rangle + \frac{1}{2\eta}\|\mathbf{w} - \mathbf{w}^t\|^2$, we have*

$$\langle v^t, \mathbf{w}^{t+1} - \mathbf{w}^t\rangle \le -\frac{1}{2\eta}\|\mathbf{w}^{t+1} - \mathbf{w}^t\|^2. \qquad (22)$$

*Then we can get*

$$\Phi(\mathbf{w}^{t+1}) \le \Phi(\mathbf{w}^t) + \left(\frac{L_\beta}{2} - \frac{1}{4\eta}\right)\|\mathbf{w}^{t+1} - \mathbf{w}^t\|^2 + \eta\|\nabla\Phi(\mathbf{w}^t) - v^t\|^2 \qquad (23)$$

$$\Phi(\mathbf{w}^{t+1}) \le \Phi(\mathbf{w}^t) + \left(\frac{L_\beta}{2} - \frac{1}{4\eta}\right)\|\mathbf{w}^{t+1} - \mathbf{w}^t\|^2 + 2\eta\|\nabla\Phi(\mathbf{w}^t) - v^t\|^2 - \eta\|\nabla\Phi(\mathbf{w}^t) - v^t\|^2 \qquad (24)$$

$$\eta\|\nabla\Phi(\mathbf{w}^t) - v^t\|^2 \le \Phi(\mathbf{w}^t) - \Phi(\mathbf{w}^{t+1}) + \left(\frac{L_\beta}{2} - \frac{1}{4\eta}\right)\|\mathbf{w}^{t+1} - \mathbf{w}^t\|^2 + 2\eta\|\nabla\Phi(\mathbf{w}^t) - v^t\|^2. \qquad (25)$$

*Then we want to bound $\mathbb{E}\|\nabla\Phi(\mathbf{w}^t) - v^t\|^2$.*

$$\|\nabla\Phi(\mathbf{w}^t) - v^t\|^2 = \|(1-\theta)(v_1^{t-1} + v_2^{t-1}) + \theta(G_1^t + G_2^t) - \nabla\Phi(\mathbf{w}^t)\|^2$$

$$= \|(1-\theta)v_1^{t-1} + \theta G_1^t - \nabla F(\mathbf{w}^t) + (1-\theta)v_2^{t-1} + \theta G_2^t - \nabla H(\mathbf{w}^t)\|^2 \qquad (26)$$

$$= \|v_1^t - \nabla F(\mathbf{w}^t) + v_2^t - \nabla H(\mathbf{w}^t)\|^2$$

*Since $\mathbb{E}_t[\langle v_1^t - \nabla F(\mathbf{w}^t), v_2^t - \nabla H(\mathbf{w}^t)\rangle] = 0$, we have*

$$\mathbb{E}_t\|\nabla\Phi(\mathbf{w}_{t+1}) - v_{t+1}\|^2 = \mathbb{E}_t\|v_1^{t+1} - \nabla F(\mathbf{w}^{t+1})\|^2 + \mathbb{E}_t\|v_2^{t+1} - \nabla H(\mathbf{w}^{t+1})\|^2 \qquad (27)$$

Summing (14), $\frac{20\theta L_f^2 \tilde{C}_{\nabla g}^2 n_0}{\gamma_1 |\mathcal{B}|} \times$ (15) and $\frac{20\theta L_f^2 \tilde{C}_{\nabla g}^2 n_0}{\gamma_1 |\mathcal{B}|} \times$ (16), we can get

$$
\begin{aligned}
\mathbb{E}[\Delta_1^{t+1}] \leq & (1-\theta)\mathbb{E}[\Delta_1^t] + \left( \frac{2L_F^2}{\theta} + \frac{100\theta L_g^2 L_f^2 \tilde{C}_{\nabla g}^2 n_0^2}{\gamma_1^2 |\mathcal{B}|^2} \right) \mathbb{E}[\|\mathbf{w}^{t+1} - \mathbf{w}^t\|^2] \\
& + \frac{20\theta L_f^2 \tilde{C}_{\nabla g}^2 n_0}{\gamma_1 |\mathcal{B}|} \left( 1 - \frac{\gamma_1 |\mathcal{B}|}{4n_0} \right) \mathbb{E} \left[ \Xi_1^t + \Xi_2^t - \Xi_1^{t+1} - \Xi_2^{t+1} \right] \\
& - L_f^2 \tilde{C}_{\nabla g}^2 \left( \frac{5\theta n_0}{\gamma_1 |\mathcal{B}|} - 3 \right) \mathbb{E} \left[ \frac{1}{n_0} \sum_{i \in \mathcal{B}^{t+1}} \left\| u_{1i}^{t+1} - u_{1i}^t \right\|^2 + \left\| u_{2i}^{t+1} - u_{2i}^t \right\|^2 \right] \\
& + \frac{2\theta^2 L_f^2 (\sigma_{\nabla g}^2 + L_g^2)}{min\{|\mathcal{B}|, |\mathcal{B}_{1i}|, |\mathcal{B}_{2i}|\}} + \frac{80\theta\gamma_1 \sigma_g^2 L_f^2 \tilde{C}_{\nabla g}^2}{\min\{|\mathcal{B}_{1i}|, |\mathcal{B}_{2i}|\}}
\end{aligned}
\tag{28}
$$

Summing (18) and $\frac{20\theta\beta^2 \tilde{C}_{\nabla h}^2 m}{\gamma_2 |\mathcal{B}_c|} \times$ (19), we can get

$$
\begin{aligned}
\mathbb{E}[\Delta_2^{t+1}] \leq & (1-\theta)\mathbb{E}[\Delta_2^t] + \left( \frac{2\beta^2 L_H^2}{\theta} + \frac{100\theta\beta^2 L_h^2 \tilde{C}_{\nabla h}^2 m^2}{\gamma_2^2 |\mathcal{B}_c|^2} \right) \mathbb{E}[\|\mathbf{w}^{t+1} - \mathbf{w}^t\|^2] \\
& + \frac{20\theta\beta^2 \tilde{C}_{\nabla h}^2 m}{\gamma_2 |\mathcal{B}_c|} \left( 1 - \frac{\gamma_2 |\mathcal{B}_c|}{4m} \right) \mathbb{E} \left[ \Gamma_t - \Gamma_{t+1} \right] \\
& - \beta^2 \tilde{C}_{\nabla g}^2 \left( \frac{5\theta m}{\gamma_2 |\mathcal{B}_c|} - 3 \right) \mathbb{E} \left[ \frac{1}{m} \sum_{k \in \mathcal{B}_c^{t+1}} \left\| u_k^{t+1} - u_k^t \right\|^2 \right] \\
& + \frac{\theta^2 \beta^2 (\sigma_{\nabla h}^2 + L_h^2)}{min\{|\mathcal{B}_c|, |\mathcal{B}_k|\}} + \frac{40\theta\gamma_2 \beta^2 \sigma_h^2 \tilde{C}_{\nabla h}^2}{|\mathcal{B}_k|}
\end{aligned}
\tag{29}
$$

Summing (25), $\frac{4\eta}{\theta} \times$ (28) and $\frac{4\eta}{\theta} \times$ (29), let $\gamma_1 = \gamma_2 = \gamma \leq \min\{\frac{5n_0\theta}{3|\mathcal{B}|}, \frac{5m\theta}{3|\mathcal{B}_c|}\}$, we can get $\frac{5\theta n_0}{\gamma|\mathcal{B}|} - 3 \geq 0$, $\frac{5\theta m}{\gamma|\mathcal{B}_c|} - 3 \geq 0$ and

$$
\begin{aligned}
& \eta\mathbb{E}\|\nabla\Phi(\mathbf{w}^t) - v_t\|^2 \\
\leq & \mathbb{E}\left[Y_t - Y_{t+1}\right] \\
& - \left( \frac{1}{4\eta} - \frac{L_F + \beta L_H}{2} - \frac{8\eta L_F^2}{\theta^2} - \frac{400\eta L_g^2 L_f^2 \tilde{C}_{\nabla g}^2 n_0^2}{\gamma^2 |\mathcal{B}|^2} - \frac{8\eta\beta^2 L_H^2}{\theta^2} - \frac{400\eta\beta^2 L_h^2 \tilde{C}_{\nabla h}^2 m^2}{\gamma^2 |\mathcal{B}_c|^2} \right) \mathbb{E}\left[\|\mathbf{w}^{t+1} - \mathbf{w}^t\|^2\right] \\
& + \frac{8\eta\theta L_f^2 (\sigma_{\nabla g}^2 + L_g^2)}{min\{|\mathcal{B}|, |\mathcal{B}_{1i}|, |\mathcal{B}_{2i}|\}} + \frac{320\eta\gamma\sigma_g^2 L_f^2 \tilde{C}_{\nabla g}^2}{\min\{|\mathcal{B}_{1i}|, |\mathcal{B}_{2i}|\}} + \frac{4\eta\theta\beta^2 (\sigma_{\nabla h}^2 + L_h^2)}{min\{|\mathcal{B}_c|, |\mathcal{B}_k|\}} + \frac{160\eta\gamma\beta^2 \sigma_h^2 \tilde{C}_{\nabla h}^2}{|\mathcal{B}_k|}
\end{aligned}
\tag{30}
$$

where

$$
Y_{t+1} = \Phi(\mathbf{w}^{t+1}) + \frac{4\eta}{\theta}\|\nabla\Phi(\mathbf{w}^{t+1}) - v^{t+1}\|^2 + \frac{80\eta L_f^2 \tilde{C}_{\nabla g}^2 n_0}{\gamma|\mathcal{B}|}(\Xi_1^{t+1} + \Xi_2^{t+1}) + \frac{80\eta\beta^2 \tilde{C}_{\nabla h}^2 m}{\gamma|\mathcal{B}_c|}\Gamma_{t+1}.
$$

If $\eta = \min\left\{ \frac{1}{12(L_F + \beta L_H)}, \frac{\theta}{8\sqrt{3}L_F}, \frac{\theta}{8\sqrt{3}L_H\beta}, \frac{\gamma|\mathcal{B}|}{40\sqrt{6}L_g L_f \tilde{C}_{\nabla g} n_0}, \frac{\gamma|\mathcal{B}_c|}{40\sqrt{6}\beta L_h \tilde{C}_{\nabla h} m} \right\}$, we have

$$
\begin{aligned}
& \frac{\eta}{24}\mathbb{E}\left[ \eta^{-2}\|\mathbf{w}^{t+1} - \mathbf{w}^t\|^2 + \|\nabla\Phi(\mathbf{w}^t) - v_t\|^2 \right] \\
\leq & \mathbb{E}\left[Y_t - Y_{t+1}\right] + \frac{8\eta\theta L_f^2 (\sigma_{\nabla g}^2 + L_g^2)}{min\{|\mathcal{B}|, |\mathcal{B}_{1i}|, |\mathcal{B}_{2i}|\}} + \frac{320\eta\gamma\sigma_g^2 L_f^2 \tilde{C}_{\nabla g}^2}{\min\{|\mathcal{B}_{1i}|, |\mathcal{B}_{2i}|\}} + \frac{4\eta\theta\beta^2 (\sigma_{\nabla h}^2 + L_h^2)}{min\{|\mathcal{B}_c|, |\mathcal{B}_k||\}} + \frac{160\eta\gamma\beta^2 \sigma_h^2 \tilde{C}_{\nabla h}^2}{|\mathcal{B}_k|}.
\end{aligned}
\tag{31}
$$

Dividing both sides by $\frac{\eta}{24}$ and taking the average over $T$ we can get

$$
\begin{aligned}
& \frac{1}{T}\sum_{t=0}^{T-1} \mathbb{E}\left[ \eta^{-2}\|\mathbf{w}^{t+1} - \mathbf{w}^t\|^2 + \|\nabla\Phi(\mathbf{w}^t) - v^t\|^2 \right] \\
\leq & \frac{24\mathbb{E}[Y_0]}{\eta T} + \frac{192\theta L_f^2 (\sigma_{\nabla g}^2 + L_g^2)}{min\{|\mathcal{B}|, |\mathcal{B}_{1i}|, |\mathcal{B}_{2i}|\}} + \frac{7680\gamma\sigma_g^2 L_f^2 \tilde{C}_{\nabla g}^2}{\min\{|\mathcal{B}_{1i}|, |\mathcal{B}_{2i}|\}} + \frac{96\theta\beta^2 (\sigma_{\nabla h}^2 + L_h^2)}{min\{|\mathcal{B}_c|, |\mathcal{B}_k|\}} + \frac{3840\gamma\beta^2 \sigma_h^2 \tilde{C}_{\nabla h}^2}{|\mathcal{B}_k|}.
\end{aligned}
\tag{32}
$$

$$Y_0 = \Phi(\mathbf{w}^0) + \frac{4\eta}{\theta} \|\nabla\Phi(\mathbf{w}^0) - v^0\|^2 + \frac{80\eta L_f^2 \tilde{C}_{\nabla g}^2 n_0}{\gamma |\mathcal{B}|}(\Xi_1^0 + \Xi_2^0) + \frac{80\eta\beta^2 \tilde{C}_{\nabla h}^2 m}{\gamma |\mathcal{B}_c|}\Gamma_0$$

$$= \Phi(\mathbf{w}^0) + \frac{4\eta}{\theta} \|\nabla\Phi(\mathbf{w}^0) - v^0\|^2 + \frac{80\eta L_f^2 \tilde{C}_{\nabla g}^2}{\gamma |\mathcal{B}|}(\|\boldsymbol{u}_1^0 - \boldsymbol{g}_1(\mathbf{w}^0)\|^2 + \|\boldsymbol{u}_2^0 - \boldsymbol{g}_2(\mathbf{w}^0)\|^2)$$

$$+ \frac{80\eta\beta^2 \tilde{C}_{\nabla h}^2}{\gamma |\mathcal{B}_c|}\|\boldsymbol{u}^0 - \boldsymbol{h}(\mathbf{w}^0)\|^2.$$

*Since $\mathbf{w}^0$ is a feasible solution, we have $\Phi(\mathbf{w}^0) = \frac{1}{n_0}\sum_{i=1}^{n_0} f(g_i(\mathbf{w}^0))$. Since $g$ is bounded by Assumption 1 and $f$ is Lipschitz continuous, we can show that there exists a constant $C_F := \max\{\tau|\log(c_g^2)|, \tau|\log(C_g^2)|\}$ such that $|F(\mathbf{w}, \mathcal{D})| \le C_F$. We assume that $u_k^0 = h_k(\mathbf{w}^0)$, $u_{1i}^0 = \hat{g}_{1i}(\mathbf{w}^0)$, $u_{2i}^0 = \hat{g}_{2i}(\mathbf{w}^0)$ and $v^0 = \frac{1}{n_0}\sum_{i=1}^{n_0}(\nabla\hat{g}_{1i}(\mathbf{w}^0)^\top\nabla f(\hat{g}_{1i}(\mathbf{w}^0) + \nabla\hat{g}_{2i}(\mathbf{w}^0)^\top\nabla f(\hat{g}_{2i}(\mathbf{w}^0)))$, we can get*

$$\mathbb{E}[\|\nabla\Phi(\mathbf{w}^0) - v^0\|^2] \le 2(C_{\nabla g}^2 + \sigma_{\nabla g}^2)C_{\nabla f}^2$$
$$\mathbb{E}[\|\boldsymbol{u}_1^0 - \boldsymbol{g}_1(\mathbf{w}^0)\|^2] \le \sigma_g^2$$
$$\mathbb{E}[\|\boldsymbol{u}_2^0 - \boldsymbol{g}_2(\mathbf{w}^0)\|^2] \le \sigma_g^2 \quad (33)$$
$$\mathbb{E}[\|\boldsymbol{u}^0 - \boldsymbol{h}(\mathbf{w}^0)\|^2] = 0$$

*Therefore, we can get*

$$\frac{1}{T}\sum_{t=0}^{T-1}\mathbb{E}\left[\eta^{-2}\|\mathbf{w}^{t+1} - \mathbf{w}^t\|^2 + \|\nabla\Phi(\mathbf{w}^t) - v^t\|^2\right]$$

$$\le \frac{24C_F}{\eta T} + \frac{192(C_{\nabla g}^2 + \sigma_{\nabla g}^2)C_{\nabla f}^2}{\theta T} + \frac{1920 L_f^2 \tilde{C}_{\nabla g}^2 \sigma_g^2}{|\mathcal{B}|\gamma T} \quad (34)$$

$$+ \frac{192\theta L_f^2(\sigma_{\nabla g}^2 + L_g^2)}{min\{|\mathcal{B}|, |\mathcal{B}_{1i}|, |\mathcal{B}_{2i}|\}} + \frac{7680\gamma\sigma_g^2 L_f^2 \tilde{C}_{\nabla g}^2}{\min\{|\mathcal{B}_{1i}|, |\mathcal{B}_{2i}|\}} + \frac{96\theta\beta^2(\sigma_{\nabla h}^2 + L_h^2)}{min\{|\mathcal{B}_c|, |\mathcal{B}_k|\}} + \frac{3840\gamma\beta^2\sigma_h^2 \tilde{C}_{\nabla h}^2}{|\mathcal{B}_k|}.$$

*Let $\beta = \frac{1}{\epsilon\delta}$, $\theta = \min\left\{\frac{\epsilon^4\delta^2\min\{|\mathcal{B}_k|, |\mathcal{B}_c|\}}{672(\sigma_{\nabla h}^2 + L_h^2)}, \frac{\epsilon^2\min\{|\mathcal{B}|, |\mathcal{B}_{1i}|, |\mathcal{B}_{2i}|\}}{1344 L_f^2(\sigma_{\nabla g}^2 + L_g^2)}\right\} = O(\epsilon^4\delta^2)$,*
*$\gamma_1 = \gamma_2 = \gamma \le \min\left\{\frac{5n_0\theta}{3|\mathcal{B}|}, \frac{5m\theta}{3|\mathcal{B}_c|}, \frac{\epsilon^4\delta^2|\mathcal{B}_k|}{26880\sigma_h^2 \tilde{C}_{\nabla h}^2}\right\} = O(\epsilon^4\delta^2)$,*
*$\eta = \min\left\{\frac{1}{12(L_F + \beta L_H)}, \frac{\theta}{8\sqrt{3}L_F}, \frac{\theta}{8\sqrt{3}L_H\beta}, \frac{\gamma_1|\mathcal{B}|}{40\sqrt{6}L_g L_f \tilde{C}_{\nabla g}n}, \frac{\gamma_2|\mathcal{B}_c|}{40\sqrt{6}\beta L_h \tilde{C}_{\nabla h}m}\right\} = O(\epsilon^5\delta^3)$ and*
*$T = O(\epsilon^{-7}\delta^{-3})$, we have*

$$\frac{1}{T}\sum_{t=0}^{T-1}\mathbb{E}\left[\eta^{-2}\|\mathbf{w}^{t+1} - \mathbf{w}^t\|^2 + \|\nabla\Phi(\mathbf{w}^t) - v^t\|^2\right] \le O(\epsilon^2)$$

*By the definition of $\mathbf{w}^{t+1}$, we have*

$$\mathbf{w}^{t+1} - \mathbf{w}^t + \eta v^t = 0$$
$$\Leftrightarrow \eta^{-1}(\mathbf{w}^t - \mathbf{w}^{t+1}) + (\nabla\Phi(\mathbf{w}^t) - v^t) + (\nabla\Phi(\mathbf{w}^{t+1}) - \nabla\Phi(\mathbf{w}^t)) = \nabla\Phi(\mathbf{w}^{t+1})$$
$$\Leftrightarrow \eta^{-1}(\mathbf{w}^t - \mathbf{w}^{t+1}) + (\nabla\Phi(\mathbf{w}^t) - v^t) + (\nabla\Phi(\mathbf{w}^{t+1}) - \nabla\Phi(\mathbf{w}^t))$$
$$= \nabla F(\mathbf{w}^{t+1}) + \frac{\beta}{m}\nabla\boldsymbol{h}(\mathbf{w}^{t+1})[\boldsymbol{h}(\mathbf{w}^{t+1})]_+$$

*This gives*

$$\left\|\nabla F(\mathbf{w}^{t+1}) + \frac{\beta}{m}\nabla\boldsymbol{h}(\mathbf{w}^{t+1})^\top[\boldsymbol{h}(\mathbf{w}^{t+1})]_+\right\|^2$$

$$\le 3\left(\eta^{-2}\|\mathbf{w}^t - \mathbf{w}^{t+1}\|^2 + \|\nabla\Phi(\mathbf{w}^t) - v^t\|^2 + \|\nabla\Phi(\mathbf{w}^{t+1}) - \Phi(\mathbf{w}^t)\|^2\right)$$

$$\le 3\left(\eta^{-2}\|\mathbf{w}^t - \mathbf{w}^{t+1}\|^2 + \|\nabla\Phi(\mathbf{w}^t) - v^t\|^2 + L_\beta^2\|\mathbf{w}^t - \mathbf{w}^{t+1}\|^2\right)$$

$$\le 3\left(\eta^{-2}\|\mathbf{w}^t - \mathbf{w}^{t+1}\|^2 + \|\nabla\Phi(\mathbf{w}^t) - v^t\|^2 + \eta^{-2}\|\mathbf{w}^t - \mathbf{w}^{t+1}\|^2\right)$$

$$\le 6(\eta^{-2}\|\mathbf{w}^t - \mathbf{w}^{t+1}\|^2 + \|\nabla\Phi(\mathbf{w}^t) - v^t\|^2)$$

*Therefore, we can achieve that*

$$\frac{1}{T}\sum_{t=0}^{T-1}\mathbb{E}\left[\left\|\nabla F(\mathbf{w}^{t+1}) + \frac{\beta}{m}\nabla \boldsymbol{h}(\mathbf{w}^{t+1})[\boldsymbol{h}(\mathbf{w}^{t+1})]_+\right\|^2\right] \le O(\epsilon^2) \tag{35}$$

*By Jensen's inequality, we can get*

$$\mathbb{E}\left[\left\|\nabla F(\mathbf{w}^{\hat{t}}) + \frac{\beta}{m}\nabla \boldsymbol{h}(\mathbf{w}^{\hat{t}})[\boldsymbol{h}(\mathbf{w}^{\hat{t}})]_+\right\|\right] \le O(\epsilon), \tag{36}$$

*with $\hat{t}$ selected uniformly at random from $\{1, \cdots, T\}$.*

*Then, with the full rank assumption on the Jacobian, which is $\|\nabla \boldsymbol{h}(\mathbf{w}^t)[\boldsymbol{h}(\mathbf{w}^t)]_+\| \ge \delta\|[\boldsymbol{h}(\mathbf{w}^t)]_+\|$ as in Assumption 3, we can get*

$$
\begin{aligned}
\|[\boldsymbol{h}(\mathbf{w}^{t+1})]_+\|^2 \le & \frac{1}{\delta^2}\|\nabla \boldsymbol{h}(\mathbf{w}^{t+1})[\boldsymbol{h}(\mathbf{w}^{t+1})]_+\|^2 \\
= & \frac{m^2}{\beta^2\delta^2}\|\nabla F(\mathbf{w}^{t+1}) + \frac{\beta}{m}\nabla \boldsymbol{h}(\mathbf{w}^{t+1})[\boldsymbol{h}(\mathbf{w}^{t+1})]_+ - \nabla F(\mathbf{w}^{t+1})\|^2 \\
\le & \frac{2m^2}{\beta^2\delta^2}\left[\left\|\nabla F(\mathbf{w}^{t+1})\right\|^2 + \left\|\nabla F(\mathbf{w}^{t+1}) + \frac{\beta}{m}\nabla \boldsymbol{h}(\mathbf{w}^{t+1})[\boldsymbol{h}(\mathbf{w}^{t+1})]_+\right\|^2\right]
\end{aligned}
\tag{37}
$$

*Taking the average over $T$, we can get*

$$
\begin{aligned}
\frac{1}{T}\sum_{t=0}^{T-1}\mathbb{E}\|[\boldsymbol{h}(\mathbf{w}^{t+1})]_+\|^2 \le & \frac{1}{T}\sum_{t=0}^{T-1}\frac{2m^2}{\beta^2\delta^2}\mathbb{E}\left[\left\|\nabla F(\mathbf{w}^{t+1})\right\|^2 + \left\|\nabla F(\mathbf{w}^{t+1}) + \frac{\beta}{m}\nabla \boldsymbol{h}(\mathbf{w}^{t+1})[\boldsymbol{h}(\mathbf{w}^{t+1})]_+\right\|^2\right] \\
\le & O(\epsilon^2)
\end{aligned}
\tag{38}
$$

*and using $\boldsymbol{\lambda} = \frac{\beta}{m}[\boldsymbol{h}(\mathbf{w}^{\hat{t}})]_+$. By Jensen's inequality, we can get*

$$\mathbb{E}\|[\boldsymbol{h}(\mathbf{w}^{\hat{t}})]_+\| \le O(\epsilon) \tag{39}$$

$$
\begin{aligned}
\mathbb{E}|\boldsymbol{\lambda}^\top[\boldsymbol{h}(\mathbf{w}^{\hat{t}})]_+| = \mathbb{E}\left|\frac{\beta}{m}[\boldsymbol{h}(\mathbf{w}^{\hat{t}})]_+^\top[\boldsymbol{h}(\mathbf{w}^{\hat{t}})]_+\right| = & \frac{\beta}{m}\mathbb{E}\|[\boldsymbol{h}(\mathbf{w}^{\hat{t}})]_+\|^2 \\
= & \frac{1}{m\delta\epsilon}\mathbb{E}\|[\boldsymbol{h}(\mathbf{w}^{\hat{t}})]_+\|^2 \\
\le & O(\epsilon).
\end{aligned}
\tag{40}
$$

