# OpenReview forum: "A Retention-Centric Framework for Continual Learning with Guaranteed Model Developmental Safety"
_TMLR — Rejected by TMLR_

### Review · Reviewer_7MoH · 2025-09-06

**Summary Of Contributions:**

This paper proposes a novel evaluation framework for continual learning, Model Developmental Safety (MDS), which aims to ensure old capabilities are maintained as new data is added during training. It proposes a simple penalty-based method which jointly optimizes accuracy on the new task while also applying gradient updates on previous tasks whose estimated accuracy has dropped below their initial value prior to fine-tuning.

**Audience:**

Yes

**Audience Explanation:**

**Positives**

- MDS is an interesting concept, and seems relevant for certain safety-critical applications.
- The basic idea behind the algorithm (leverage low-rank adapters on the final layer weights for each task, apply a gradient mask to only apply updates to tasks whose performance is dropping too much) makes a lot of sense.

**Negatives**

- The paper does not discuss the feasibility or desirability of the MDS constraint in cases where tasks are learned sequentially and it is not possible, either due to interference or due to limited model capacity, to meaningfully reduce the loss on the target without increasing the loss on the new data
- The theoretical significance of the paper is limited, meaning that I don't believe readers would read the paper in order to understand the theoretical results. In particular, the significance of Lemma 2 seems somewhat mischaracterized, as it seems to be a straightforward result of adding more parameters to an optimization problem. I could obtain a similar result in a trivial one-dimensional optimization problem where $m=1$, e.g. $\min_x (x - c)^2$, by adding an extra variable $y$ and obtaining a new optimization problem $\min_{x, y} (x + y - c)^2$ and setting $y=0$, for which we have that $\|\nabla_{x,y} (x+ y - c)^2\| = \|(x - c)e_1 + (x - c)e_2\| = \sqrt{2}|x - c|$.
      - The simplest explanation of the benefit from the additional optimizer heads is, in my view, the increased expressivity and reduced interference of the resulting classifier head between tasks.
      - I don't believe that Lemma 2, which only states that adding a no-op LoRA head increases the minimum eigenvalue of the gradient covariance matrix, is able to explain the empirical results discussed in section 6.4.2. This is because after training one would assume that the low-rank adapters would be non-zero if they are doing anything useful.

**Claims And Evidence:**

No

**Claims Explanation:**

**Strengths**
    - The paper evaluates the method against a wide variety of baselines.
    - The evaluations are ambitious in evaluating on a high-performing, complex network architecture and on large, highly nonstationary datasets such as the task-incremental Places365.
    - While I did not review the proof of theorem 1 in detail, lemma 2 seems correct, if somewhat trivial as I will describe below.


**Weaknesses**

- It was somewhat difficult to evaluate the correctness of the paper due to notational choices. For example:
      - g is a function but G is a gradient estimate
      - double usage of $u^{t}_{1, i}$, $u^t_{2, i}$, $u^t_k$ to refer to estimators of completely different functions
      - $\mathcal{B}_c$ is a subset of $\{1, \dots, m\}$ but $\mathcal{B}$ is a set of datapoints $(\mathbf{x}_i, \mathbf{t}_i)$, while $\mathcal{B}_k$ is again a set of datapoints.
      - $m$ is not defined in the algorithm (it is defined elsewhere, but it is not clear if in Algorithm 1 the target task is $D_m$ or $D_{m+1}$), nor is $v^0$
      - I couldn't find where in the algorithm the cross-entropy on the *target* task is minimized
      - Similarly, it is unclear to me whether $\mathcal{D}$ contains all of the previous data or only the new task data.

- The ablation in Section 6.4.2 suggests that task-dependent heads are critical. However, it is not clear whether the baselines against which the method was compared were also afforded the additional low-rank adapters.
    - The binary nature of MDS makes it difficult to assess the *degree* to which it is violated, which could matter a great deal in many applications. It also makes it placing Figure 3 in context difficult, as the difference between a 100% and 0% retention rate might correspond to a negligible (but nonzero) drop in accuracy on the prior tasks.
      - Further, it wasn't entirely clear what the constraint-based formulation gives us that traditional notions of forgetting do not.

    - The bar graphs in Figure 5 are a bit misleading, the difference between 4/5 seeds and 5/5 seeds not violating a constraint translates to an apparent 20% gap between two methods, which while technically true in the observed data doesn't necessarily correspond to a statistically significant difference. More generally, I would have liked to see greater statistical rigor in how these results were presented.
    - The trajectories in Figure 2 look somewhat odd, in that training on the e.g. dressing room target does not seem to improve training accuracy. It is also odd to me that there is an immediate drop in performance on both the old and new tasks in many cases, which makes me wonder if the optimizer has been tuned appropriately.

**Requested Changes:**

**Evaluations:**
  - Please augment the binary developmental safety rate in Figure 3 with a more continuous variant which measures the drop in performance (e.g. as in the x-axis on Figure 2)
  -  Extend the evaluations against baselines to include a mix-and-match approach where the baseline methods and the proposed method were each evaluated with and without the task-specific heads, to get a sense of what role they play in the performance gap.
  -  Explain why there is so often a drop-off to the lower left quadrant in Figure 2, along with why some tasks have extremely low accuracy (e.g. overcast)
  -  Evaluate the method on at least one additional architecture to demonstrate the generality of the approach, and evaluate the trade-offs associated with allowing some "forgetting budget" (e.g. rather than restricting to zero drop in accuracy, can we get significantly better performance on the new task if we allow a small drop in accuracy on previous ones by shifting the threshold for zeroing the gradient updates to some value $-\epsilon$?)

**Positioning of method and discussion**

  - Revisit the discussion of Lemma 2 to ensure that it does not overclaim beyond the actual result.
  - Disentangle the two aspects of the proposed algorithm: the task-specific adapters and the gradient masking rule, and ensure the independent effects of each, along with their interaction, are rigorously discussed and analyzed.
  - Discuss how the algorithm can be generalized to other learning protocols (related to my fourth point in the evaluations section)
  - In my view, the motivation for the fairly draconian constraints enforced by the MDS objective is best positioned in a setting where e.g. a practitioner is continually fine-tuning a large expensive model in a high-stakes environment; however, in most cases learning is not done *fully* online, and it is worth giving a stronger motivation for following the approach outlined in this paper as opposed to simply starting a new training/finetuning run from scratch on all of the new data, with task-specific weights.

---

> ### Author Response · Authors · 2025-12-11
>
> We thank the reviewer for dedicating the time to providing helpful feedback on our paper. Below we would like to address the raised concerns.
>
>
> **Q1:** It was somewhat difficult to evaluate the correctness of the paper due to notational choices. For example: g is a function but G is a gradient estimate — double usage of $u^t_{1i}, u^t_{2i}, u^t_{k}$ to refer to estimators of completely different functions — $\mathcal{B}_c$ is a subset of {1, …, m} but $\mathcal{B}$ is a set of datapoints $ (x_i, t_i)$, while $\mathcal{B}_k$ is again a set of datapoints.
>
> $m$ is not defined in the algorithm (it is defined elsewhere, but it is not clear in Algorithm 1 the target task is $D_m$ or $D_{m+1}$. I couldn't find where in the algorithm the cross-entropy on the target task is minimized — Similarly, it is unclear to me whether D contains all of the previous data or only the new task data.
>
> **A:** We apologize for the confusion. We adopt the convention that "$u^t$-related terms" denote estimators, whereas "$\mathcal B$-related terms" denote mini-batches, for easier understanding. Since, we have text-image pairs with CLIP models and a set of $m$ protected tasks, $\mathcal{B}$ is a mini-batch of text-image pairs, $\mathcal{B}_c$ is a mini-batch of protected task, and $\mathcal{B}_k$ is a mini-batch of samples for protected task $k$. As introduced in notation section, $m$ denotes the number of protected task and the target task is $\mathbb T_o$ with dataset $\mathcal D$. We only apply contrastive loss on the target task, following prior FLYP work. $\mathcal D$ only contains the new task data, which we clarified in the revised notation section.
>
>
> **Q2:** The ablation in Section 6.4.2 suggests that task-dependent heads are critical. However, it is not clear whether the baselines also afforded the additional low-rank adapters. Extend the evaluations against baselines where the baseline methods and the proposed method were each evaluated without the task-specific heads.
>
>
> **A:** Since the proposed task-heads are motivated by our theoretical analysis to speed up convergence of our algorithm, baseline methods didn't afford the additional low-rank adapters as following their paper. We also experimented with our method without task-heads in Figure 5(a), which shows that our method without task-heads are still able to achieve a high retention ratio, which baseline methods fail to attain shown in Figure 2(revised paper), demonstrating the effectiveness of the proposed algorithm. Moreover, with task-dependent heads, our method can achieve even higher retention ratios and higher accuracy on the target task, highlighting the benefit of task-dependent heads for promoting model performance.
>
>
> **Q3:** The binary nature of MDS makes it difficult to assess the degree to which it is violated, which could matter a great deal in many applications. Please augment the binary developmental safety rate in Figure 3 with a more continuous variant which measures the drop in performance (e.g. as in the x-axis on Figure 2)
>
> **A:** Thank you for the valuable comment. As suggested, in Table 7, 8, 9 in Appendix A.5, we summarized target performance changes and DevSafety(acc) that measures the largest decrease across protected tasks. We can see that baselines usually lead to 1-14 percent decrease when targeting Tunnel, 0-13 percent decrease when targeting Foggy, 1-30 percent decrease when targeting Overcast. While none of the baselines may achieve both positive DevSafety(acc) and $\Delta$Acc simultaneously, our method is the only one that attains both, highlighting the superiority of the proposed method.
>
> **Q4:** Further, it wasn't entirely clear what the constraint-based formulation gives us that traditional notions of forgetting do not.
>
> **A:** As discussed in the introduction, classic continual learning focus on trading off performance on previous tasks and new tasks to have good average performance. Our constraint-based formulation ensures strictly reservation of critical existing abilities (i.e., **zero forgetting**) while learning new tasks, which cannot be achieved by existing continual learning methods as shown in our paper.
>
>
> **Q5:** The bar graphs in Figure 5 are a bit misleading,  with only five runs summarized.
>
> **A:** To ensure more reliable results, we summarized results over 10 runs with different random seeds in Figure 5 in the revision. We can still observe that our method with task-dependent heads almost consistently exhibit higher retention ratios and higher accuracy on the target task. And using a constant $\beta$ still outperforms an increasing $\beta$.

---

> > ### Author Response · Authors · 2025-12-11
> >
> > **Q6:** The trajectories in Figure 2 look somewhat odd. Explain why there is so often a drop-off to the lower left quadrant in Figure 2, along with why some tasks have extremely low accuracy (e.g. overcast)
> >
> > **A:** Note that x-axis on Figure 2 is DevSafety(acc) which measures the largest performance drop across protected tasks. At initial stages, the model explores  aggressively with large step, therefore some protected task performance gets better and the other protected task performance gets worse, reflecting a negative DevSafety(acc) overall. As shown in Figure 3 left (revised paper), protected tasks own large adaptive weight at the beginning, leading to negative $\Delta$Acc(Target), presenting a drop-off to the lower left quadrant.
> >
> > It appears to be a misunderstanding that "overcast" has extremely low accuracy. As discussed in Section 6.2, the base model already achieves 73.6% accuracy for overcast task. Since y-axis on Figure 2 reports $\Delta\text{Acc(Target)} = \text{Acc}(\text{Target}, \mathbf w_{\text{new}}) - \text{Acc}(\text{Target}, \mathbf w_{\text{old}})$, improvements are naturally smaller in this case, resulting in a relatively lower$\Delta$Acc(Target) in Figure 2.
> >
> >
> > **Q7:** The paper does not discuss the feasibility or desirability of the MDS constraint.
> >
> > **A:** Thank you for your constructive comment. We've incorporated more discussion on desirability and feasibility of our work in section 7 in the revised paper. While our work focuses on ensuring model developmental safety (i.e., zero forgetting) on essential capabilities in cost-sensitive high-stakes domains via MDS constraints, strict preservation may be not necessary in certain scenarios and a small performance drop can be tolerated in exchange for higher target performance. In such cases, classical continual learning method and metrics (e.g., average accuracy) may be more appropriate. Moreover, since satisfying the MDS constraints while improving target performance is challenging, sufficient model capacity may be required to guarantee zero forgetting, and task-dependent heads can further help mitigate this issue.
> >
> > **Q8:** The theoretical significance of the paper is limited. Besides, the significance of Lemma 2 seems somewhat mischaracterized, as it seems to be a straightforward result of adding more parameters to get better performance.
> >
> > **A:** We politely disagree. The theoretical analysis algorithm 1 is an important contribution to constrained optimization community. To the best of our knowledge, this is the first convergence analysis of a penalty method for solving non-convex inequality constrained optimization, which we highlighted at the beginning of section 5.2.
> >
> >
> > Regarding lemma2, we agree that it's intuitive to explain the benefit of task-dependent heads via that adding more parameters leads to better performance. Note that in our work, task-dependent heads design is motivated by our theoretical analysis which shows that the dependence on $\delta$ slows down the convergence. Our lemma2 provides further clarification for why the intuition holds in our case and how it help with learning by avoid getting trapped at a flat location. Moreover, we also conducted empirical verification of lemma2 in table 3 in section 6.4.2, which shows that minimal singular value of $\nabla \widehat{{h}}(\hat{ \mathbf{w}})$ indeed gets larger than that of  $\nabla{\textbf{h}}(\mathbf{w})$, consistent with lemma2 and providing additional insight about the intuition.

---

> ### Author Response · Authors · 2025-12-11
>
> **Q9:** Evaluate the method on at least one additional architecture to demonstrate the generality of the approach.
>
>
> **A:** To validate applicability of our method to other models, we conduct additional experiments for classification on CheXpert dataset[1] for detecting chest and lung diseases with traditional CNNs. Specifically, we first train a Desnet121 model on detecting four diseases, i.e., *Cardiomegaly, Edema, Atelectasis, Pleural Effusion*, and then continually train the model to detect a new disease (*Consolidation*) while preserving the performance on previously learned four diseases, with 4k samples for each protected task. The AUC score results are summarized below. We can see that our method is still effective for learning traditional CNN models while retaining protected tasks'  performance, which cannot be achieved by RM (0.003 AUC drop, which, different from accuracy, is considered as a significant performance drop as score differences between ranked models are usually less than 0.001 on the Chexpert Competition Leaderboard [2]), demonstrating the effectiveness of our method when applied to other models. We included the results in Appendix A.6 in the revised paper.
>
>
>
>
> | Method | Measures                   | Performance            |
> | ------ | -------------------------- | ---------------------- |
> | Base   | Retention Ratio//DevSafety | 100%//0.00(0.0000)     |
> |        | Target Consolidation (AUC)       | 0.500                  |
> | RM     | Retention Ratio//DevSafety | 0.00%//-0.0031(0.0032)  |
> |        | Target Consolidation (AUC)      | 0.8846(0.0014)         |
> | **Ours**   | Retention Ratio//DevSafety | 100.00%//0.0009(0.0007) |
> |        | Target Consolidation (AUC)      | 0.8852(0.0034)         |
>
> [1] Irvin, Jeremy, et al. "Chexpert: A large chest radiograph dataset with uncertainty labels and expert comparison." 2019.
>
> [2] https://stanfordmlgroup.github.io/competitions/chexpert/
>
>
> **Q10:** evaluate the trade-offs associated with allowing some "forgetting budget". Can we get significantly better performance on the new task if we allow a small drop in accuracy on previous ones by shifting the threshold for zeroing the gradient updates to some value $\epsilon$?
>
> **A:** In real-world settings, the “forgetting budget” is often determined in an ad hoc manner by practitioners. To align with the motivation of this work, we employ strict metrics, DevSafety(acc) and Retention Ratio, to ensure model developmental safety (i.e., zero forgetting) on essential capabilities in high-stakes domains. However, one may still consider adapting the MDS constraints by replace 0 with $\epsilon$ in Eqn.3 to allow a small amount of forgetting. Since the choice of $\epsilon$ is highly application-dependent, we leave it for future exploration.
>
> **Q11:** Discuss how the algorithm can be generalized to other learning protocols.
>
> **A:** We added more discussion in section 7 in the revised paper regrading how to extend our work to other domains. For example, to improve math capability of LLMs, one may instantiate  $F(\mathbf w,\mathcal{D})$ in Eqn. 2 as an SFT (Supervised Finetuning) loss on math task data with $\mathcal {L}_k(\mathbf w, \mathcal {D}_k)$ being SFT loss on other tasks data, like harmlessness task data, then the optimization algorithm 1 is still applicable.
>
>
> **Q12:** It is worth giving a stronger motivation for following the approach outlined in this paper as opposed to simply starting a new training/finetuning run from scratch on all of the new data.
>
> **A:** As discussed in the introduction, learning-enabled systems are iteratively developed to address new challenges, and preserving essential established capabilities is crucial in cost-sensitive applications. For example, autonomous driving systems are continuously developed and validated. Restarting training or fine-tuning from scratch on all of data is often prohibitive, as it discards prior training and verification efforts and not all previously used data may be readily available. These cost-sensitive applications highlight the practical value of our method.

---

### Review · Reviewer_euNm · 2025-09-09

**Summary Of Contributions:**

Authors tackle the problem of continual learning: how to update a pre-trained model with new data/for a new task. If done naively, continual learning degrades the performance of the models on the old tasks and yields the so-called catastrophic forgetting.

Authors propose to avoid catastrophic forgetting by strictly ensuring that the performances of the new/updated models do not decrease of the old tasks: this is done via directly hardcoding in the training loss that the performances on old tasks should not decrease (Equation 2). In order to obtain a simpler optimization problem, the constraint is relaxed (Equation 4).

To empirically support the proposed method, the authors provide experiments on the CLIP representation.

**Audience:**

Yes

**Audience Explanation:**

Catastrophic forgetting is a very important problem in macine learning

**Claims And Evidence:**

Yes

**Claims Explanation:**

The authors want to ensure a very specific property while training on new data.
To this aim, they propose a new algorithm and matching experiments

**Requested Changes:**

- In my (very subjective) opinion, the "convergence analysis" in Section 5.2 hurts a bit the flow of the paper, and does not bring so much to the table. The results (and the 8-page long proof) of Theorem 1 seem convoluted, can't the author directly apply standard convergence optimization theorems? What is the main theoretical challenge here?
- Figure 3. Is the 'retention ratio' a standard metric to evaluate the performances in continual learning? It feels like a very hard metric. Is there a way to show/summarize the target accuracy difference after the update of the model? The average and or the maximal loss in target accuracy difference. Note that I am not asking for SOTA experiments, only for a more 'informative' metric.

---

> ### Author Response · Authors · 2025-12-11
>
> We thank the reviewer for providing constructive comments. Below we would like to answer the raised questions.
>
>
> **Q1:** The "convergence analysis" hurts a bit the flow of the paper and does not bring so much to the table. Can the author directly apply standard convergence optimization theorems for the convergence analysis? What is the main theoretical challenge here?
>
> **A:** Existing standard convergence analysis can’t be directly applied to our problem, due to multiple reasons. (1) Our problem (Eqn. 3) involves a non-convex objective and non-convex constraints, but standard analysis like SGD/Adam can only handle unconstrained problems. (2) Our problem may contain a large number of constraints, so we may need to sample constraints when updating the solution, which is challenging. (3) While existing work [1] proposed convergence analysis for constrained problems with equality constraints and unbiased stochastic gradients, our problem involves inequality constraints and biased stochastic gradients due to **the compositional structure of penalty function**. We highlighted the challenges at the beginning of section 5.1. To the best of our knowledge, this is the first convergence analysis of a penalty method for solving non-convex inequality constrained optimization.
>
> Moreover, the convergence analysis is an important part of the paper. Thanks to the convergence analysis which shows that the dependence on $\delta$ slows down the convergence, we accordingly proposed the task-dependent heads for CLIP models to enable faster learning and better performance.
>
> [1] Ahmet Alacaoglu and Stephen J Wright. Complexity of single loop algorithms for nonlinear programming with stochastic objective and constraints. AISTATS 2024
>
> **Q2:** Is there a way to show/summarize the target accuracy difference after the update of the model? The average and or the maximal loss in target accuracy difference.
>
> **A:** Thank you for your constructive comments. For better understanding of the experimental results, in Table 7, 8, 9 in Appendix A.5, we summarized target performance changes and DevSafety(acc) that measures the largest decrease across protected tasks. We can see that baselines usually lead to 1-14 percent decrease when targeting Tunnel, 0-13 percent decrease when targeting Foggy, 1-30 percent decrease when targeting Overcast. While none of the baselines may achieve both positive DevSafety(acc) and $\Delta$Acc simultaneously, our method is the only one that attains both, highlighting the superiority of the proposed method.

---

> > ### Comment · Reviewer_euNm · 2025-12-19
> >
> > I thank the author for their response.
> > The table 7-9 are actually very hard to parse. Would it be possible just to make the (now) Figure 2 with a "softer" metric?

---

> ### Author Response · Authors · 2025-12-20
> **Comparisons based on target performance change and DevSafety (acc)**
>
> Thank you for the constructive suggestion. In Figure 8 in the revised paper, we provided comparisons with baseline methods based on target performance change ($\Delta$Acc (Target)) and DevSafety (acc). Our method is the only approach that achieves both positive $\Delta$Acc (Target) and DevSafety (acc) simultaneously. In particular, when targeting Tunnel, our method attains target-task improvement comparable to the baselines while ensuring model developmental safety, demonstrating its effectiveness. When targeting Overcast, we also see that baselines may achieve higher target-task gains by sacrificing performance on protected tasks. Such trade-offs are undesirable in cost-sensitive, high-stakes applications, where strictly preserving the model’s core capabilities is essential. We've further discussed the desirability of our method compared with classical continual learning in Section 7.

---

### Review · Reviewer_s7Ax · 2025-11-26

**Summary Of Contributions:**

**The primary question** of this article is how to train a model on a new task while retaining its old abilities strictly. The authors coin the term 'model development safety': a new model strictly retains important capabilities of the old model while improving target-task performance, which is formally defined as for a new set of weights the loss on all tasks cannot be larger than the loss on that task with the old weights.

**The authors propose** retention-centric training which is a constrained optimisation problem that optimises the new model parameters under the constraint that these cannot increase the loss on older tasks, modelled by a set of datapoints from those tasks. A theoretical guarantee is given on the generalisation from this set of examples of the older tasks to how well the model retains its capabilities on these tasks using standard tools.

**The authors propose an efficient algorithm** to optimise the parameters subject to a potentially large set of constraints. This method converts the constraints to quadratic penalties on violations and then solves the unconstrained problem using a variance-reduced stochastic gradient method. Furthermore, the authors demonstrate how to convert the constrained optimisation problem into an unconstrained optimisation problem by overcoming two violated assumptions that prohibit using an existing method to do so, and demonstrate how to efficiently calculate the gradients for their newly obtained unconstrained objective as well as how to make the gradient estimator unbiased. Finally, they show that the algorithm is guaranteed to converge under three assumptions. I have not checked these details as this is too far outside my domain of expertise to be able to comment on in the capacity of a reviewer.
The convergence guarantee result is dependent on some parameter delta, which can slow down convergence and requires the authors to practically use task-dependent heads for the CLIP models for better convergence.

**The authors empirically demonstrate their framework using a CLIP-style image classification model in two domains**: classifying weather condition in several scene types (e.g. foggy, overcast) as well as a scene recognition dataset to experiment with a large number of protected classes / constraints. They demonstrate that their method compared to 6 continual learning baselines gets higher retention rates (which is the percentage of times that none of the protected tasks have a higher loss after training on the target task), and do several ablations to indicate which components of their framework are important (e.g. task-dependent heads are important for convergence).

**Additional Comments:**

Nits:
	- Misspelled OpenAI's model series (e.g. https://openai.com/index/gpt-4-research/).
	- Figure 1 is a bit low-information upon a first sequential paper read. Is this the performance of your method? How does it compare to baselines that allegedly only focus on average performance? Especially given that you compare against 6 relevant baselines, possible to add some more of the information into figure 1?
	- Typo in section 6.2 (Retenntion).
	- Discuss https://arxiv.org/abs/2403.08763 as related work?

**Audience:**

Yes

**Audience Explanation:**

Continual learning is a very relevant topic that is currently unsolved.

**Broader Impact Concerns:**

N.A.

**Claims And Evidence:**

Yes

**Claims Explanation:**

Although the authors only demonstrate their method empirically in a relatively specific setup (CLIP-style image classification where continual learning is about learning new classes), they do so convincingly in this domain, comparing against many relevant baselines and demonstrating clear improvements.

**Requested Changes:**

**Required for acceptance**
- I think the work is strong as it stands. Most of the minimum requirements for acceptance imo are met: the claims are met with sufficient evidence. However, the experimental setup is quite contrived and likely to be similar to only a select few real-world continual learning problems. Although the framework itself is broadly applicable, I'd urge the authors to at least discuss these limitations in the main body of work more up front (the title also seems to hint at a more generally applicable framework than is experimented with empirically). My main point of improvement required therefore is being more explicit about limitations in the text.
- There's no discussion of the cost trade-off of the constraints included. How much less expensive is this optimisation than just retraining the whole model on all the new data? How much more/less expensive is this than the baselines?

**Would strengthen the work**
- The experimental setup is quite contrived (as mentioned above). The way it is setup now, there needs to be no interference between the target classes and the protected classes. The authors mention the framework is applicable to language model finetuning, but here the classes are tokens and what is a task is less well-defined (especially when one tries to retain the capabilities of a base model after finetuning). Experiments on different domains and using a different framework than CLIP would strengthen this work.
- Relatedly, discussion on how this framework could be applied to other domains would be useful.
- If task-dependent parameters (heads) are required for convergence, the method has limited practical applicability. The amount of task-dependent heads can grow very large for real-world continual / lifelong learning. Some discussion about this would be good.
- There seems to be a trade-off between retention rate and target class accuracy (seen from figure 3 and 4 right. It would strengthen the work to discuss this more upfront in the intro and discussion.

---

> ### Author Response · Authors · 2025-12-11
>
> We thank the reviewer for providing a comprehensive review. Below we would like to address raised concerns.
>
> **Q1:** Add discussion regrading limited experimental setup and extension to language model finetuning where a task is less well-defined.
>
>
> **A:** We thank the reviewer for the constructive comment. We added more discussion in section 7 in the revised paper regrading limited experimental validation in this work and how to extend our work to fine-tune LLMs. For example, to improve math capability of LLMs, one may instantiate  $F(\mathbf w,\mathcal{D})$ in Eqn. 2 as an SFT (Supervised Finetuning) loss on math task data with $\mathcal{L}_k(\mathbf w, \mathcal{D}_k)$ being SFT loss on other tasks data, like harmlessness task data, then the optimization algorithm 1 is still applicable.
>
>
>
> **Q2:** What is the cost of the constraints included? How much more/less expensive is the proposed method than the baselines in term of computaion?
>
>
> **A:** In our experiments, all the methods, including baselines and our method, are trained on the same amount of data and iterations. The table below summarizes the runtime for each run when targeting the foggy task. We observe that the cost of our constrained method is similar to that of the regularization method (RM), as both compute gradients using data from the target and protected tasks. This results in a slightly higher cost than the WCLL baseline, which does not involve constraints or regularization.
>
> | Method | Time (min) |
> | ------ | ---------- |
> | WCCL   | 600        |
> | RM     | 713        |
> | Ours   | 714        |
>
>
> **Q3:** Experiments on different domains and using a different framework than CLIP would strengthen this work.
>
> **A:** To validate applicability of our method to other models, we conduct additional experiments for classification on CheXpert dataset[1] for detecting chest and lung diseases with traditional CNNs. Specifically, we first train a Desnet121 model on detecting four diseases, i.e., *Cardiomegaly, Edema, Atelectasis, Pleural Effusion*, and then continually train the model to detect a new disease (*Consolidation*) while preserving the performance on previously learned four diseases, with 4k samples for each protected task. The AUC score results are summarized below. We can see that our method is still effective for learning traditional CNN models while retaining protected tasks'  performance, which cannot be achieved by baseline RM (0.003 AUC drop, which, different from accuracy, is considered as a significant performance drop as score differences between ranked models are usually less than 0.001 on the Chexpert Leaderboard [2]), demonstrating the effectiveness of our method when applied to other models. We included the results in Appendix A.6 in the revised paper.
>
>
> | Method | Measures                   | Performance            |
> | ------ | -------------------------- | ---------------------- |
> | Base   | Retention Ratio//DevSafety | 100%//0.00(0.0000)     |
> |        | Target Consolidation (AUC)       | 0.500                  |
> | RM     | Retention Ratio//DevSafety | 0.00%//-0.0031(0.0032)  |
> |        | Target Consolidation (AUC)      | 0.8846(0.0014)         |
> | **Ours**   | Retention Ratio//DevSafety | 100.00%//0.0009(0.0007) |
> |        | Target Consolidation (AUC)      | 0.8852(0.0034)         |
>
> [1] Irvin, Jeremy, et al. "Chexpert: A large chest radiograph dataset with uncertainty labels and expert comparison." 2019.
>
> [2] https://stanfordmlgroup.github.io/competitions/chexpert/
>
>
> **Q4:** If task-dependent parameters (heads) are required for convergence, the method has limited practical applicability since the amount of task-dependent heads can grow very large for real-world continual / lifelong learning.
>
>
> **A:** We would like to clarify that task-dependent parameters (heads) are not required for convergence but to speed up the convergence, as discussed in section 5.2 and 5.3. Moreover, evidenced by Figure 5, our method without task-dependent heads may also achieve a high retention ratio, which baseline methods fail to attain. With task-dependent heads, our method can achieve even higher retention ratios and higher accuracy on the target task. Therefore, task-dependent heads are recommended to achieve better performance.
>
> Besides, the cost of task-dependent heads is quite small, compared with the model size. Specifically, each task head in our experiments consists of 32k (512\*32\*2) parameters. Even when scaling to 100 heads, this adds only 3.2M parameters, which remains substantially smaller than the 150M parameters of the CLIP ViT-B/16 model. Furthermore, task-dependent head is one method to speed up the convergence. When the number of tasks goes very large, we can also consider more parameter efficient approach to increase $\delta$ to facilitate convergence.

---

> ### Author Response · Authors · 2025-12-11
>
> **Q5:** There seems to be a trade-off between retention rate and target class accuracy.
>
> **A:** From table 1, we can indeed observe that, with the Retention Ratio increasing, the improvement on the targeted task decreases, indicating a tradeoff between enhancing the targeted task’s performance and satisfying the developmental safety requirements. We made it clear in the experimental part in the revision.
>
>
> **Q6:** Figure 1 is a bit low-information upon a first sequential paper read. Is this the performance of your method? What about baseline performance?
>
>
> **A:** We apologize for the confusion. In section 6.2, we compared our method with baselines in one-round continual model development, which shows that none of baselines may achieve MDS while improving target performance. Therefore, in section 6.3, we only include our method here to focus on demonstrating the effectiveness of the proposed retention-centric framework in multiple rounds of model development process. However, to improve the flow of the paper, we move back Figure 1 from the introduction to the experimental part in section 6.3 to avoid confusion.
>
>
> **Q7:** Misspelling, Typo, and discussion about [r1]
>
> **A:** Thank you for pointing these out. As [r1] proposes to combine replay and LR re-scheduling for continually pre-training LLMs, it falls into the category of memory-based approaches. We have revised the paper accordingly.
>
> [r1] https://arxiv.org/abs/2403.08763

---

> ### Comment · Reviewer_s7Ax · 2025-12-18
> **Additional comment on computational trade-offs**
>
> Thanks for the extensive reply! One thing that maybe wasn't clear in my initial review is that I'm also wondering how much less computationally expensive this method is than simply retraining on all the data everytime a new tasks needs to be learned. Could you provide some discussion on this (no need to actually compute the time it would cost, just discuss what you gain from this framework computationally over the simplest potential baseline)

---

> > ### Author Response · Authors · 2025-12-19
> > **Computational cost of retraining on all the data**
> >
> > Thank you for the clarification! This is a very important point. Retraining on all the data will indeed incur substantially higher computational cost compared to our framework. For example, when targeting Tunnel, our method needs only a total of around 4k samples from protected tasks to formulate the constraints to ensure model developmental safety. In contrast, retraining on all the data requires 55.6k samples from protected tasks, which may lead to more than 10 times the computation.
> >
> > Moreover, in real scenarios such as autonomous driving or LLM fintuning, as the model is continuously developed on a large amount of data, it is often impractical and expensive to retrain on all the data every time a new task needs to be learned as not all previously used data is readily available, highlighting the practical value of our method.

---

### Decision · Action_Editor_T3cD · 2026-04-16

**Recommendation:** Reject

**Additional Comments:**

There are mixed opinions among the reviewers, even those that recommend acceptance.

Here is one example of the mixed feedback, "The authors propose a framework for an important problem in our field, namely continual learning, and validate it in one experimental setup before reviews, and another after the reviews+discussion. The paper addresses an important problem in an empirically sound way, and I recommend acceptance for that reason. Nonetheless, the experimental setup is somewhat contrived, even after the addition of the new domain in response to reviewer comments: in both cases, continual learning is about learning a new class on top of existing classes, and whether the proposed framework with task-dependent heads applies to generative modeling remains an open question."

Another reviewer writes, "The authors create their own metric of success: no performance loss on the previous tasks, that IMO is not standard. I do not know the extent to which the community is interested by this metric."

Even the reviewer with the strongest CL background wrote, although recommending acceptance: "The contributions within my field of expertise (continual learning) do not seem sufficient, though it may be possible that someone from the field of constrained optimization would judge theorem 1 to merit inclusion."

In preparing subsequent revisions, the authors are strongly encouraged to incorporate the reviewers' feedback and especially address any lingering concerns mentioned in the reviews.

**Audience:**

Yes

**Audience Explanation:**

This paper is of general interest to CL researchers.

**Claims And Evidence:**

No

**Claims Explanation:**

This article addresses an important problem in CL, proposing a retention-centric framework for preserving prior capabilities while improving performance on a new task. The reviewers generally agree that the paper is relevant and technically complete. Two reviewers weakly support acceptance, based on teh paper's empirical gains, comparison to broad baselines, and subsequent revisions that expanded discussion, softer metrics, computational analysis, and additional experiments that strengthened its scope. However, the feedback remains mixed, even among those reviewers recommending acceptance. Those concerns center around the narrow/contrived aspects of the experimental setup, the use of a retention ratio metric that isn’t widely used, and the strict zero-forgetting objective that may limit generality. There is also disagreement around the contribution of the theoretical contribution, with mixed opinions among the reviewers. Overall, the revision appears to have addressed the initial weaknesses, but not so significantly as to currently justify publication in TMLR without subsequent revision. Consequently, my recommendation must be a weak reject -- it is close to the bar, but does not meet it. This was a difficult decision, as it was a borderline case.

In any subsequent revision, the authors would be advised to 1) expand and better justify the experimental setup, reducing contrived aspects of it, 2) clarify and justify the use of the strict zero-forgetting objective, 3) justify the use of the retention ratio metric, and 4) better position the theoretical contributions of the paper.

**Resubmission Of Major Revision:**

The authors may consider submitting a major revision at a later time.